# Learning Stochastic Multiscale Models

**Andrew F. Ilersich**
Institute for Aerospace Studies
University of Toronto
Toronto, ON M3H 5T6
andrew.ilersich@mail.utoronto.ca

**Prasanth B. Nair**
Institute for Aerospace Studies
University of Toronto
Toronto, ON M3H 5T6
prasanth.nair@utoronto.ca

## Abstract

The physical sciences are replete with dynamical systems that require the resolution of a wide range of length and time scales. This presents significant computational challenges since direct numerical simulation requires discretization at the finest relevant scales, leading to a high-dimensional state space. In this work, we propose an approach to learn stochastic multiscale models in the form of stochastic differential equations directly from observational data. Drawing inspiration from physics-based multiscale modeling approaches, we resolve the macroscale state on a coarse mesh while introducing a microscale latent state to explicitly model unresolved dynamics. We learn the parameters of the multiscale model using a simulator-free amortized variational inference method with a Product of Experts likelihood that enforces scale separation. We present detailed numerical studies to demonstrate that our learned multiscale models achieve superior predictive accuracy compared to under-resolved direct numerical simulation and closure-type models at equivalent resolution, as well as reduced-order modeling approaches.

## 1 Introduction

Multiscale phenomena are ubiquitous in nature, and broadly fall under two categories. The first are systems with large-scale and small-scale regimes, each of which may follow different laws but interact with each other at some interface, so simulation requires one to model both regimes. Such systems are common in materials science [1] and biology [2]. The second, which we focus on in this work, are systems governed by partial differential equations (PDEs) that require one to resolve a wide range of length and time scales to make accurate predictions. A canonical example is weather prediction. Modeling the atmosphere requires resolving phenomena from cloud microphysics ($< 1$m) up to planetary waves (5000 km) [3]. In such scenarios, direct numerical simulation (DNS) of the governing equations on a grid that resolves the smallest scale of interest is computationally infeasible.

The computational intractability of fully-resolved simulations necessitates working with coarsened representations. This introduces a fundamental multiscale separation between resolved and sub-grid-scale dynamics. To "close" the system, the effect of the sub-grid-scale dynamics on the resolved dynamics must be modeled. This approach, known as closure modeling [4, 5], has emerged independently across multiple fields. Large-eddy simulation (LES) in fluid dynamics filters out small-scale phenomena and closes the Navier-Stokes equations with a sub-grid-scale model [6]. Similarly, in coarse-grained molecular dynamics, atoms are grouped into larger "beads" to reduce degrees of freedom [7]. Recently, machine learning techniques have been used to infer closure models for climate models [8], coarse-grained molecular dynamics [9], and turbulence [10, 11].

We propose a novel framework for learning *stochastic multiscale dynamics* directly from observational data. Our approach introduces a latent variable model that explicitly decomposes the system state into macroscale ($\zeta$) and microscale ($\eta$) components, whose dynamics are governed by a learned coupled system of stochastic differential equations (SDEs). This structure captures complex inter-scale

39th Conference on Neural Information Processing Systems (NeurIPS 2025).

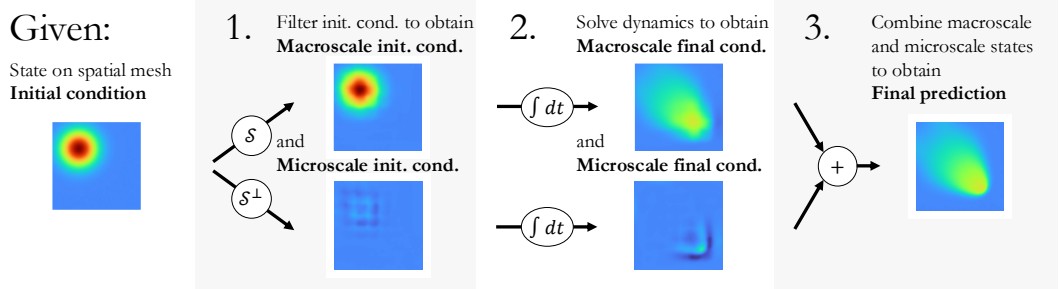

Figure 1: Overview of our multiscale modeling approach.

interactions and propagates uncertainty naturally; see Figure 1 for a graphical overview. Our main contributions are:

- **A novel probabilistic multiscale formulation:** We introduce a latent variable framework for stochastic multiscale modeling, which involves learning a system of coupled SDEs governing distinct, interacting macroscale and microscale latent states directly from data.
- **Multiscale likelihood for enforcing scale separation:** We enforce explicit scale separation by formulating a Product of Experts (PoE) multiscale likelihood that ensures macroscale features dominate predictions while microscale components provide only necessary corrections.
- **Efficient learning and empirical validation:** We enable efficient training of our multiscale stochastic model using an amortized, simulator-free variational inference scheme [12]. Numerical studies on challenging PDE systems demonstrate that explicitly modeling microscale dynamics yields superior predictive accuracy compared to under-resolved DNS and closure-type models at equivalent resolution, as well as reduced-order modeling approaches.

**Related work**   Reduced-order modeling (ROM) methods reduce the computational cost of simulating complex systems through dimensionality reduction of the state space [14]. Traditional ROM methods employ linear projection [15, 16, 17, 18]; however, the inherent limitations of linear subspaces for systems with slowly decaying Kolmogorov $n$-width have led to the development of nonlinear manifold methods [19, 20, 21, 22, 23]. ROM methods struggle with multiscale systems [24], motivating specialized closure models [25, 26]. It is worth noting that standard ROM methods learn a low-dimensional, global latent representations of the dynamics. In contrast, our approach learns a spatially-distributed representation on a coarse grid, positioning it as a stochastic coarse-grained modeling framework more akin to paradigms such as LES.

There is a wide body of literature on data-driven closure modeling; see [4, 5] for a review. The Mori-Zwanzig (MZ) formalism provides a powerful theoretical framework for deriving coarse-grained models of dynamical systems [13, 27, 28], showing that the effects of unresolved scales manifest as memory (non-Markovian) terms in the evolution of the resolved variables. The MZ framework has motivated various approaches for closure modeling [29, 4, 30, 31, 32]. More recently, Boral et al. [11] proposed a neural SDE-based approach for turbulence closure modeling. This approach, motivated by ideal LES, enables probabilistic closure modeling but requires running the coarse solver in the loop during both training and inference.

Our approach fundamentally differs from data-driven closure modeling, which typically augments the macroscale dynamics with a learned correction that implicitly accounts for the sub-grid effects. In contrast, we explicitly model the coupled dynamics of macroscale and microscale latent states, providing a Markovian representation capable of capturing the non-Markovian memory effects arising from coarse-graining (see Appendix B). Moreover, our approach enables prediction at full resolution through a decoder, whereas closure models are inherently limited to coarse-scale predictions.

Recent work in weather forecasting has employed hierarchical graph neural networks operating on multiple mesh resolutions to learn auto-regressive models [33, 34]. In contrast, our multiscale approach learns continuous-time coupled SDEs governing macroscale and microscale dynamics, performing test-time simulations only on a single coarse grid and using high-resolution grids solely as decoding targets.

## 2 Problem setting

We consider dynamical systems governed by PDEs on a spatial domain $\Omega \subset \mathbb{R}^d$, where $d \in \mathbb{N}$ is the spatial dimension. We denote the PDE solution by the spatio-temporal vector field $u \in \mathcal{U}(\Omega \times [t_0, T]; \mathbb{R}^{d_u})$, where $\mathcal{U}$ denotes an appropriate function space (typically a Sobolev space) endowed with a suitable norm.

To learn a multiscale model from data, we work with a discretized representation of $u$ on a spatio-temporal grid. We introduce a spatial projection operator, $\mathcal{P}_n^s : \mathcal{U}(\Omega; \mathbb{R}^{d_u}) \to \mathbb{R}^{nd_u}$, that maps a vector field to its values on a spatial grid with $n$ points. Using this, we define the *fully-resolved* state $y(t) \in \mathbb{R}^{n_y}$ by projecting the PDE solution onto a fine spatial grid with $n_f$ points: $y(t) := \mathcal{P}_{n_f}^s(u(\cdot, t))$, where the state dimension is $n_y = n_f d_u$. This grid is assumed to be sufficiently fine to faithfully represent the dynamics and thus serves as our ground-truth. The training dataset $\mathcal{Y} := \{t_i, y_i\}_{i=1}^{n_t}$ comprises noisy observations of the fully-resolved state at $n_t$ time steps, i.e.,

$$y_i := y(t_i) + \epsilon_i, \quad \epsilon_i \sim \mathcal{N}(\epsilon_i \,|\, 0, \sigma^2 I), \text{ for } i = 1, 2, \ldots, n_t, \tag{1}$$

where $\epsilon_i$ represents zero-mean i.i.d. Gaussian noise with covariance $\sigma^2 I$, and $t_i$ denotes the observation time-stamp. We assume access to this dataset without knowledge of the underlying governing equations.

Our objective is to infer a stochastic, continuous-time, latent variable model from the data. The stochastic framework is essential for capturing both potential inherent randomness in the underlying physics and uncertainty introduced by modeling approximations and data limitations. Next, we present our approach for learning a multiscale SDE model for the dynamics of $y$ by decomposing it into a macroscale state on a coarse grid and a microscale state representing sub-grid-scale features.

## 3 Multiscale framework

We introduce our stochastic multiscale modeling framework that addresses the challenge of learning the dynamics of the fully-resolved state $y(t) \in \mathbb{R}^{n_y}$ from the observational data $\mathcal{Y}$. Our main idea is to represent the high-dimensional state $y$ through a lower-dimensional latent state, $z(t) \in \mathbb{R}^{n_z}$, that explicitly separates the *macroscale state* denoted by $\zeta(t) \in \mathbb{R}^{n_\zeta}$[1] and the *microscale state* denoted by $\eta(t) \in \mathbb{R}^{n_\eta}$. The temporal evolution of $\zeta$ and $\eta$ is governed by a learned system of coupled SDEs, capturing both deterministic dynamics and inherent stochasticity arising from scale separation and model reduction.

Our framework is composed of three core components with learnable parameters that we denote by $\theta$: a probabilistic encoder that maps the observations of $y$ to the latent state $z = (\zeta, \eta)$, a system of coupled SDEs governing the dynamics of $z$, and a probabilistic decoder for mapping $z$ back to the observation space to make predictions. The following sections detail each of these components.

**Probabilistic scale separation (encoder)**  Inspired by classical multiscale methods like LES, we decompose the fully-resolved state $y$ into a large-scale component $\overline{y}$ and a residual small-scale component $\widetilde{y} = y - \overline{y}$. In contrast to classical deterministic approaches, we define a probabilistic encoder $p_\theta(z \,|\, y)$ that explicitly models the uncertainty in identifying and compressing scale-separated features from the high-dimensional state. We parameterize the *mean* of this conditional distribution using the operators defined below that are analogous to traditional multiscale analysis:

1. **Smoothing Operator** $\mathcal{S}_\theta : \mathbb{R}^{n_y} \to \mathbb{R}^{n_y}$ serves as a low-pass filter that extracts the large-scale component, $\overline{y} := \mathcal{S}_\theta(y)$; for example, a fixed Gaussian filter or a learned convolutional network.

2. **Residual Operator** $\mathcal{S}_\theta^\perp$ defines the residual component, $\widetilde{y} := \mathcal{S}_\theta^\perp(y) = y - \overline{y}$, which contains small-scale features.

3. **Restriction Operator (Macroscale Encoder)** $\mathcal{E}_\theta^\zeta : \mathbb{R}^{n_y} \to \mathbb{R}^{n_\zeta}$ maps the smoothed component, $\overline{y}$, to the low-dimensional macroscale latent space, $\zeta$, with $\mathbb{E}[\zeta \,|\, y] := \mathcal{E}_\theta^\zeta(\mathcal{S}_\theta(y))$; for example, coarse sampling or a more complex learned mapping.

---

[1]  While we denote $\zeta$ as a vector for notational simplicity, it represents field variables on a coarse spatial grid. This structure informs the parameterization of the drift and diffusion functions, which can be modeled using spatially-aware architectures such as graph neural networks that are capable of handling both regular and irregular grids.

4. **Microscale Encoder** $\mathcal{E}_\theta^\eta : \mathbb{R}^{n_y} \to \mathbb{R}^{n_\eta}$ compresses the residual into the compact microscale state, $\eta$, with $\mathbb{E}[\eta \,|\, y] := \mathcal{E}_\theta^\eta(\mathcal{S}_\theta^\perp(y))$; for example, a learned nonlinear mapping that efficiently captures complex small-scale structures.

We model the conditional distribution of $z$ given the fully-resolved state as a Gaussian, i.e., $p_\theta(z \,|\, y) = \mathcal{N}(z \,|\, \mu_\theta^z(y), \Sigma_\theta^z)$. The mean $\mu_\theta^z(y) = \mathbb{E}[z \,|\, y] = [\mathcal{E}_\theta^\zeta(\mathcal{S}_\theta(y))^T, \; \mathcal{E}_\theta^\eta(\mathcal{S}_\theta^\perp(y))^T]^T$ is constructed from the outputs of the macroscale and microscale encoders, and the covariance matrix $\Sigma_\theta^z \in \mathbb{R}^{n_z \times n_z}$ is a learned parameter representing encoding uncertainty.

**Latent stochastic dynamics**  We model the evolution of the macroscale state, $\zeta$, and the microscale state, $\eta$, using a coupled system of Itô SDEs:

$$\mathrm{d}\begin{bmatrix} \zeta \\ \eta \end{bmatrix} = \begin{bmatrix} f_\theta(\zeta, \phi(\eta)) \\ g_\theta(\eta, \psi(\zeta)) \end{bmatrix} \mathrm{d}t + \begin{bmatrix} L_{\zeta\zeta}(t) & L_{\zeta\eta}(t) \\ L_{\eta\zeta}(t) & L_{\eta\eta}(t) \end{bmatrix} \begin{bmatrix} \mathrm{d}\beta_\zeta \\ \mathrm{d}\beta_\eta \end{bmatrix}, \tag{2}$$

with initial conditions $\zeta(t_0) \sim p_\theta(\zeta \,|\, y(t_0))$, $\eta(t_0) \sim p_\theta(\eta \,|\, y(t_0))$ defined by the encoder at time $t_0$, i.e., $p_\theta(z(t_0) \,|\, y(t_0))$. The learned drift functions $f_\theta : \mathbb{R}^{n_\zeta} \times \mathbb{R}^{n_\eta^*} \to \mathbb{R}^{n_\zeta}$ and $g_\theta : \mathbb{R}^{n_\eta} \times \mathbb{R}^{n_\zeta^*} \to \mathbb{R}^{n_\eta}$ govern the evolution of the macroscale and microscale states, respectively.[2] These drift functions depend on coupling functions $\phi : \mathbb{R}^{n_\eta} \to \mathbb{R}^{n_\eta^*}$ and $\psi : \mathbb{R}^{n_\zeta} \to \mathbb{R}^{n_\zeta^*}$ that mediate inter-scale interactions. The dispersion block matrices $(L_{\zeta\zeta}, L_{\zeta\eta}, L_{\eta\zeta}, L_{\eta\eta})$ modulate the influence of the $(n_\zeta + n_\eta)$-dimensional Wiener process $\beta$ with components $\beta_\zeta$ and $\beta_\eta$, capturing both inherent randomness and uncertainty from unresolved scales.

More compactly, we can write the coupled multiscale latent SDE as

$$\mathrm{d}z = \gamma_\theta(z)\mathrm{d}t + L_\theta(t)\mathrm{d}\beta, \qquad z(t_0) \sim p_\theta(z(t_0) \,|\, y(t_0)), \tag{3}$$

where $\gamma_\theta(z) = [f_\theta(\zeta, \phi(\eta))^T, \; g_\theta(\eta, \psi(\zeta))^T]^T$ is the concatenated drift, and $L_\theta(t) \in \mathbb{R}^{n_z \times n_z}$ is the dispersion matrix composed of the blocks $L_{\zeta\zeta}, L_{\zeta\eta}, L_{\eta\zeta}, L_{\eta\eta}$.[3] The coupled SDE in (3) defines the prior dynamics on the latent trajectory. The solution to this SDE, initialized from the encoded state at $t_0$, yields the prior predictive distribution $p_\theta(z(t) \,|\, y(t_0))$ for $t > t_0$.

**Probabilistic state reconstruction (decoder)**  In order to make predictions, we require a decoder, $p_\theta(y \,|\, z)$, to map the latent state dynamics governed by the coupled system of SDEs in (3) to the observation space $\mathbb{R}^{n_y}$. Towards this end, we introduce macroscale and microscale decoders below:

1. **Prolongation Operator (Macroscale Decoder)** $\mathcal{D}_\theta^\zeta : \mathbb{R}^{n_\zeta} \to \mathbb{R}^{n_y}$ maps the macroscale state $\zeta$ to the fine grid.

2. **Microscale Decoder** $\mathcal{D}_\theta^\eta : \mathbb{R}^{n_\eta} \to \mathbb{R}^{n_y}$ maps the compressed microscale state $\eta$ to the fine grid.

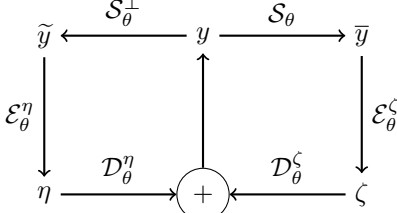

Figure 2: Relationships between multiscale states and operators. $\mathcal{S}_\theta$: smoothing operator; $\mathcal{E}_\theta^\zeta$: macroscale encoder; $\mathcal{D}_\theta^\zeta$: macroscale decoder; $\mathcal{S}_\theta^\perp$: residual operator; $\mathcal{E}_\theta^\eta$: microscale encoder; $\mathcal{D}_\theta^\eta$: microscale decoder.

Later in Section 4.1, we formulate a PoE likelihood to enforce scale separation that will establish the specific functional form of the probabilistic decoder $p_\theta(y \,|\, z)$. The predictive distribution of the learned multiscale model that follows from the proposed multiscale likelihood is presented in Section 4.4. We summarize the relationship between all operators introduced in this section in Figure 2.

**Remarks on closure models and implicit-scale models**  Traditional closure modeling aims to derive equations governing *only* the resolved (macroscale) dynamics by parameterizing the effect of unresolved (microscale) dynamics purely in terms of resolved state variables [4, 5]. At first glance,

---

[2]  More generally, the drift functions can be parametrized as $f_\theta(\zeta, \phi(\eta), \chi(t))$ and $g_\theta(\eta, \psi(\zeta), \chi(t))$ (e.g. neural networks or physics-based parametrizations) with $\chi(t)$ denoting a time encoding. In our implementation, we use a neural network architecture for $f_\theta$ that respects the spatial locality of the coarse grid variables in $\zeta$; see Appendix A for details.

[3]  The latent SDE model implicitly assumes Markovian dynamics; we discuss the validity of this assumption in Appendix B.

it may appear that closure modeling is a special case of our framework obtained by setting $n_\eta = 0$, which reduces the latent state to the macroscale state ($z = \zeta$) and reconstruction involves only the prolongation operator, with $y$ predicted solely from $\zeta$. However, the fundamental objective in closure modeling is to ensure that the predicted coarse state $\zeta$ (or $\mathcal{D}_\theta^\zeta(\zeta)$) matches the filtered state $\overline{y}$, *not* the original observation $y$. In contrast, our framework, even when $n_\eta = 0$, aims to predict the *fully-resolved* state $y$. We therefore refer to the $n_\eta = 0$ case as an **implicit-scale** model to distinguish it from conventional approaches. It implicitly represents all scales solely through $\zeta$ and its associated decoder. Our multiscale framework (with $n_\eta > 0$) explicitly models sub-grid scales via $\eta$, aiming for improved accuracy in representing the dynamics of the fully-resolved state $y$.

## 4 Multiscale variational inference

We now present a simulator-free stochastic variational inference (SVI) approach to learn the parameters of the multiscale SDE model introduced in the previous section. Our approach involves three key ingredients: a multiscale PoE likelihood that enforces scale separation, variational approximations for the latent state $z$ and model parameters $\theta$, and a reparametrized evidence lower bound (ELBO) that can be maximized without an SDE solver in the optimization loop.

### 4.1 Multiscale likelihood

To explicitly enforce separation between macroscale and microscale states, we employ a PoE perspective [35] to define the likelihood of observing $y$ given the latent states $\zeta$ and $\eta$ as:

$$p_\theta(y \,|\, z) = p_\theta(y \,|\, \zeta, \eta) = \frac{1}{Z(\zeta, \eta)} p_1(y \,|\, \zeta) \, p_2(y - \mathcal{D}_\theta^\zeta(\zeta) \,|\, \eta), \qquad (4)$$

where $Z(\zeta, \eta) = \int p_1(y' \,|\, \zeta) \, p_2(y' - \mathcal{D}_\theta^\zeta(\zeta) \,|\, \eta) dy'$ denotes the normalization constant.

The term $p_1(y \,|\, \zeta)$ in (4) can be viewed as the "macroscale expert" that evaluates how well the macroscale state explains the full observation $y$, while $p_2(y - \mathcal{D}_\theta^\zeta(\zeta) \,|\, \eta)$ functions as the "microscale expert" assessing how well the microscale state explains the residual $y - \mathcal{D}_\theta^\zeta(\zeta)$ after accounting for the macroscale prediction. This PoE formulation establishes a natural hierarchy where macroscale features are modeled first, with microscale features accounting only for unexplained residuals. The normalization constant reinforces scale separation by penalizing large microscale contributions, thereby ensuring that the microscale state does not dominate the overall prediction unless necessary.

For Gaussian experts with covariance matrices $\Sigma_\theta^\zeta$ and $\Sigma_\theta^\eta$, i.e., $p_1(y \,|\, \zeta) = \mathcal{N}(y \,|\, \mathcal{D}_\theta^\zeta(\zeta), \Sigma_\theta^\zeta)$ and $p_2(y - \mathcal{D}_\theta^\zeta(\zeta) \,|\, \eta) = \mathcal{N}(y - \mathcal{D}_\theta^\zeta(\zeta) \,|\, \mathcal{D}_\theta^\eta(\eta), \Sigma_\theta^\eta)$, the log-likelihood simplifies to (omitting constants):

$$\log p_\theta(y \,|\, \zeta, \eta) = -\frac{1}{2}\Big( \|r(\zeta)\|_{S_\theta^\zeta}^2 + \|r(\zeta) - \mathcal{D}_\theta^\eta(\eta)\|_{S_\theta^\eta}^2 - \|\mathcal{D}_\theta^\eta(\eta)\|_\Lambda^2 \Big) + \frac{1}{2}\log\Big|S_\theta^\zeta + S_\theta^\eta\Big|, \quad (5)$$

where $r(\zeta) = y - \mathcal{D}_\theta^\zeta(\zeta)$ is the residual after accounting for macroscale prediction, $S_\theta^\zeta = (\Sigma_\theta^\zeta)^{-1}$ and $S_\theta^\eta = (\Sigma_\theta^\eta)^{-1}$ are precision matrices, and $\Lambda = S_\theta^\eta(S_\theta^\eta + S_\theta^\zeta)^{-1}S_\theta^\zeta$. We use the notation $\|x\|_A^2 = x^T A x$ to denote the squared norm induced by the inner product $(x, y)_A = x^T A y$ for $x, y \in \mathbb{R}^n$ and symmetric positive-definite (SPD) matrix $A \in \mathbb{R}^{n \times n}$.

Expanding the terms in (5) and rearranging yields (see Appendix C for details):

$$\log p_\theta(y \,|\, \zeta, \eta) = -\frac{1}{2}\|r(\zeta)\|_{(S_\theta^\zeta + S_\theta^\eta)}^2 + (r(\zeta), \mathcal{D}_\theta^\eta(\eta))_{S_\theta^\eta} - \frac{1}{2}\|\mathcal{D}_\theta^\eta(\eta)\|_{\Lambda'}^2 + \frac{1}{2}\log\Big|S_\theta^\zeta + S_\theta^\eta\Big|, \quad (6)$$

where $\Lambda' = S_\theta^\eta(S_\theta^\eta + S_\theta^\zeta)^{-1}S_\theta^\eta$ is an SPD matrix. It follows from the preceding equation that maximizing the proposed multiscale log-likelihood produces four key effects:

1. Minimizes the $(S_\theta^\zeta + S_\theta^\eta)-$norm of the residual $r(\zeta)$, incentivizing the macroscale state to explain as much of the full observation as possible.

2. Maximizes the $S_\theta^\eta$-inner product between the residual $r(\zeta)$ and the decoded microscale state $\mathcal{D}_\theta^\eta(\eta)$, correlating the microscale prediction with the macroscale residual.

3. Minimizes the $\Lambda'-$norm of the decoded microscale state $\mathcal{D}_\theta^\eta(\eta)$, serving as an adaptive regularizer that keeps microscale contributions small unless necessary to explain the residual.

4. Maximizes the precision matrices $S_\theta^\zeta$ and $S_\theta^\eta$, which is equivalent to minimizing $\Sigma_\theta^\zeta$ and $\Sigma_\theta^\eta$.

In summary, the structure of the proposed multiscale log-likelihood ensures that macroscale features dominate the overall predictions while microscale components provide only necessary corrections. This can be viewed as embodying Occam's razor at the microscale level and maintaining a clear multiscale hierarchy. This regularization effect emerges naturally from the PoE formulation and helps prevent overfitting to high-frequency noise in the observations.

## 4.2 Variational approximation and priors

Our multiscale model requires approximating the posterior distribution of the latent state $z$, whose prior dynamics are governed by (2). We specify a variational approximation for the posterior of the latent state, $q_\varphi(z \mid t)$, whose sample paths follow the linear SDE:

$$\mathrm{d}z = (-A_\varphi(t)z + b_\varphi(t)) \, \mathrm{d}t + L(t)\mathrm{d}\beta, \;\; z(t_0) \sim \mathcal{N}(\mu_{t_0}, \Sigma_{t_0}). \tag{7}$$

where $A_\varphi(t) \in \mathbb{R}^{n_z \times n_z}$ and $b_\varphi(t) \in \mathbb{R}^{n_z}$ are parametrized in terms of the learnable variational parameters denoted by $\varphi$, while $\mu_{t_0} \in \mathbb{R}^{n_z}$ and $\Sigma_{t_0} \in \mathbb{R}^{n_z \times n_z}$ represent the mean and covariance derived from our probabilistic encoder $p_\theta(z \mid y)$ for a given initial condition $y(t_0)$. It is worth noting that this linear SDE is used only during training. At test time, predictions are made by solving the learned *nonlinear* system of SDEs defined in (2).

For model parameters $\theta$, we can either adopt a Bayesian approach or seek point estimates. In the former case, we specify a prior distribution $p(\theta)$, then construct a variational distribution $q_\varphi(\theta)$ to approximate its posterior. Our framework can readily incorporate interpretability constraints – for example, by representing drift functions as linear combinations of basis functions with sparsity-inducing priors on their coefficients.

## 4.3 Evidence lower bound

To train our multiscale model, we maximize the ELBO below, which we derive following [36, 37].

$$\mathcal{L}(\varphi) = \sum_{i=1}^{n_t} \mathop{\mathbb{E}}_{z_i,\theta} \left[ \log p_\theta(y_i \mid z_i) \right] - \frac{1}{2} \int_{t_0}^{T} \mathop{\mathbb{E}}_{z(t),\theta} \| r_{\theta,\varphi}(z(t),t) \|_{C(t)}^2 \, \mathrm{d}t - D_{\mathrm{KL}}(q_\varphi(\theta) \parallel p(\theta)), \tag{8}$$

where $\theta \sim q_\varphi(\theta)$, $z_i \sim q_\varphi(z \mid t_i)$ and $z(t) \sim q_\varphi(z \mid t)$. The first term is the expected log-likelihood of the dataset $\mathcal{Y}$ given the latent states, incorporating the multiscale likelihood from Section 4.1. The second term is the KL divergence between the solutions of the nonlinear SDE in Equation (2) and our linear variational SDE (7), with $r_{\theta,\varphi}(z(t),t) = -A_\varphi(t)z(t) + b_\varphi(t) - \gamma_\theta(z(t))$ denoting the drift residual and $C(t) = (L_\theta(t)L_\theta^T(t))^{-1}$. The third term is the KL divergence between our variational distribution over model parameters and their prior. For conciseness, we present the ELBO for a single time-series trajectory; the extension to multiple-trajectory datasets is discussed in Appendix D.

To efficiently maximize the ELBO without requiring a forward SDE solver in the training loop, we leverage the reparameterization trick for SDEs from [37] (details in Appendix E).

## 4.4 Computing the posterior predictive distribution

After training our multiscale model, we generate probabilistic predictions by computing the posterior predictive distribution $p_\theta(y(t) \mid y(t_0))$ given an initial observation $y(t_0)$. This distribution accounts for all sources of uncertainty captured by our model.

The prediction process involves three stages. In the first stage, we map the initial observation to a distribution over possible latent states, i.e., $p_\theta(z(t_0) \mid y(t_0)) = \mathcal{N}(z(t_0) \mid \mu_{t_0}, \Sigma_{t_0})$, where $\mu_{t_0} = [\mathcal{E}_\theta^\zeta(\mathcal{S}_\theta(y(t_0)))^T, \mathcal{E}_\theta^\eta(\mathcal{S}_\theta^\perp(y(t_0)))^T]^T$ and $\Sigma_{t_0} = \Sigma_\theta^z$. In the second stage, we propagate this initial distribution through our learned latent SDE, $\mathrm{d}z = \gamma_\theta(z(s))\mathrm{d}s + L_\theta(s)\mathrm{d}\beta(s)$, $s \in [t_0, t]$, yielding the non-Gaussian distribution $p_\theta(z(t) \mid y(t_0))$. Finally, we map the latent distribution to the observation space using our PoE likelihood. This step yields a Gaussian conditional distribution for $y$ given $z$, whose mean depends on an adaptively weighted contribution of the decoded microscale state (see Appendix F for a detailed discussion):

$$p_\theta(y \mid z) = \mathcal{N}(y \mid \mu_y(z), \Sigma_y), \;\; \text{where } \mu_y(z) = \mathcal{D}_\theta^\zeta(\zeta) + \Sigma_y S_\theta^\eta \mathcal{D}_\theta^\eta(\eta) \text{ and } \Sigma_y = (S_\theta^\zeta + S_\theta^\eta)^{-1}. \tag{9}$$

Table 1: Comparison of error statistics obtained using different models for all test cases.

| Model type | Error mean $\pm$ std. dev. on test set | | | |
| --- | --- | --- | --- | --- |
| | Wave 1D $(n_\zeta = 20)$ | KdV 1D $(n_\zeta = 20)$ | Cylinder 2D $(n_\zeta = 512)$ | SWE 2D $(n_\zeta = 64)$ |
| Coarse DNS | $1.381 \pm 0.382$ | $0.523 \pm 0.196$ | $0.130 \pm 0.040$ | $0.861 \pm 0.190$ |
| DMD | $0.492 \pm 0.066$ | $0.163 \pm 0.046$ | $0.016 \pm 0.006$ | $0.531 \pm 0.227$ |
| POD-SINDy | $0.469 \pm 0.054$ | $0.057 \pm 0.028$ | $0.051 \pm 0.024$ | $0.535 \pm 0.210$ |
| Implicit scale | $0.565 \pm 0.116$ | $0.324 \pm 0.078$ | $0.063 \pm 0.005$ | $0.426 \pm 0.099$ |
| Our approach | | | | |
| $n_\eta = 1$ | $0.517 \pm 0.095$ | $0.197 \pm 0.146$ | $0.038 \pm 0.004$ | $0.543 \pm 0.360$ |
| $n_\eta = 2$ | $0.210 \pm 0.095$ | $0.077 \pm 0.038$ | $0.015 \pm 0.003$ | $0.202 \pm 0.084$ |
| $n_\eta = 3$ | $0.105 \pm 0.047$ | $0.059 \pm 0.030$ | $0.011 \pm 0.001$ | $0.190 \pm 0.075$ |
| $n_\eta = 4$ | $0.089 \pm 0.035$ | $0.062 \pm 0.029$ | $0.011 \pm 0.002$ | $0.163 \pm 0.056$ |
| $n_\eta = 5$ | $\mathbf{0.079 \pm 0.031}$ | $\mathbf{0.042 \pm 0.017}$ | $\mathbf{0.009 \pm 0.001}$ | $\mathbf{0.152 \pm 0.047}$ |

The posterior predictive distribution is then obtained by marginalization:

$$p_\theta(y(t) \,|\, y(t_0)) = \int \mathcal{N}(y(t) \,|\, \mu_y(z(t)), \Sigma_y) \, p_\theta(z(t) \,|\, y(t_0)) \, \mathrm{d}z(t). \tag{10}$$

Since this integral is analytically intractable, we use Monte Carlo sampling leading to the Gaussian mixture approximation: $p_\theta(y(t) \,|\, y(t_0)) \approx (1/N) \sum_{i=1}^{N} \mathcal{N}(y(t) \,|\, \mu_y(z_i(t)), \Sigma_y)$, where $z_i(t)$ denotes the state at time $t$ of the $i$th sample trajectory, generated by integrating the learned SDE (3) starting from an initial condition $z_i(t_0) \sim \mathcal{N}(\mu_{t_0}, \Sigma_{t_0})$. We provide further details on the prediction procedure in Appendix F.

## 5 Results and discussion

We evaluate our multiscale framework on four systems exhibiting slowly-decaying Kolmogorov $n$-width and energy cascade between scales: (1) the 1D advecting wave with varying initial conditions, (2) the 1D Korteweg-de Vries (KdV) equation with varying initial conditions, (3) a 2D Von Kármán vortex street, and (4) a radial dam break modeled with the shallow water equations (SWE) with varying initial conditions.[4] Setup/training details including an extra test case are provided in Appendix H.

We adopt a hierarchical training strategy for our multiscale models, varying $n_\eta \in \{0, \ldots, 5\}$. The implicit scale model was trained first. Its learned parameters were then used to initialize the training of the multiscale model with $n_\eta = 1$. This process was repeated, with each trained model initializing the next in the sequence, incrementing $n_\eta$ by one. For implicit scale and multiscale models, offline training cost therefore scales linearly with $n_\eta$; for the four test cases, the training time in minutes is (1) $46 + 51n_\eta$, (2) $46 + 50n_\eta$, (3) $53 + 64n_\eta$, and (4) $67 + 79n_\eta$.

We compare our multiscale models against four baselines: a coarse DNS, a DMD linear system, a reduced-order model formed over a POD basis with latent dynamics learned by SINDy, and an implicit scale model ($n_\eta = 0$). The number of gridpoints in the coarse DNS, the rank of the DMD model, and the latent state dimension in the POD-SINDy model are chosen to be equal to the total latent state dimension of our best-performing multiscale model (i.e., $n_\zeta + 5$). Further details are provided in Appendix G.

Performance is evaluated using the normalized error metric $\epsilon(t_i) = \|y_i - \hat{y}(t_i)\| / \|y_i\|$, where $y_i$ is the observation at time $t_i$ and $\hat{y}(t_i)$ is the mean prediction. All experiments were conducted on a system with 24 CPU cores, 128GB RAM, and an NVIDIA RTX4090 GPU. Results across all test cases are summarized in Table 1, showing the mean and standard deviation of the error metric $\epsilon$ on each test set. Additional visualizations of the results are provided in Appendix I.

---

[4]  Code for reproducing our results is available: `https://github.com/ailersic/multiscale-visde`.

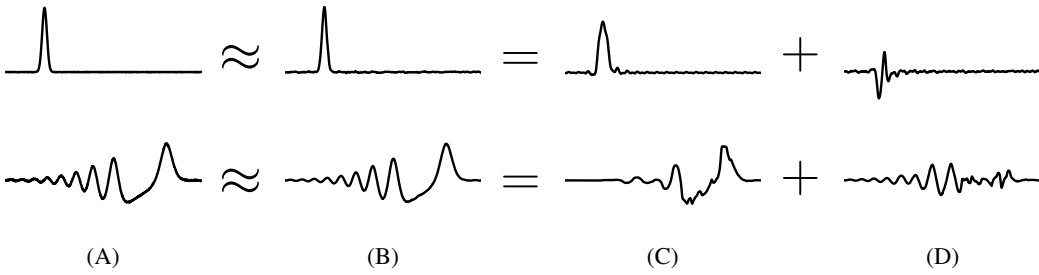

(A)          (B)          (C)          (D)

Figure 3: Advecting wave (top row) and KdV equation (bottom row): scale separation with $n_\eta = 5$, shown at $t = 0.2$ (wave) and $t = 1$ (KdV). The columns are (A) observation from dataset, (B) reconstructed full state, (C) prolonged macroscale state, and (D) decoded microscale state.

**One-dimensional advecting wave**      We test on the one-dimensional advection equation with periodic boundary conditions, i.e., $u_t = -cu_x$, where $c = 1$, $x \in [0, 1]$, and $t \in [0, 1]$ . The initial condition is $\exp(-(x/w)^2)$, where $w \sim \mathcal{U}[0.01, 0.02]$. We generate training trajectories on a fine spatial grid with spacing 0.001 and time-step 0.001 ($n_t = 1001$ and $n_y = 1000$), which are then corrupted by zero-mean Gaussian noise with standard deviation 0.001. The test set covers $t \in [0, 0.2]$. The training dataset comprises 20 trajectories, with 5 each for validation and testing. For our multiscale models, we use a coarse macroscale mesh with only $n_\zeta = 20$ points.

On this classic problem, where linear projection methods struggle, the ability to explicitly model sub-grid features proves crucial. As shown in Table 1, our multiscale model with $n_\eta = 5$ achieves an 83% error reduction compared to the best baseline (POD-SINDy), with mean error dropping from 0.469 to 0.079. The multiscale models outperform all baselines on this advection-dominated problem. Figure 3 illustrates the learned scale separation, showing how the microscale component $\eta$ captures sub-grid features that are lost in the coarse macroscale representation.

**One-dimensional Korteweg-de Vries equation**      The second test problem is the one-dimensional Korteweg-de Vries (KdV) equation with periodic boundary conditions: $u_t + uu_x + \nu u_{xxx} = 0$, where $x \in [0, 10]$ and $\nu = 0.02$. The initial condition is $a \cos(\pi x) \exp(-(x - 7.5)^2 / s^2)$, where $a \sim \mathcal{N}(2, 0.01)$ and $s \sim \mathcal{N}(1, 0.01)$. We generate training trajectories on a spatial grid with spacing 0.01 and time-step 0.001 ($n_t = 1001$, $n_y = 1000$), which are then corrupted by Gaussian noise with standard deviation 0.01. The training dataset comprises 10 trajectories, with 5 each for validation and testing. Our multiscale models use a coarse macroscale mesh with only $n_\zeta = 20$ points.

Table 1 shows that our multiscale models dramatically outperform all baselines. The best multiscale model ($n_\eta = 5$) achieves 87% error reduction compared to the implicit-scale baseline, with mean error dropping from 0.324 to 0.042. Among reduced-order methods, POD-SINDy performs best but still has 36% higher error than our approach.

Figure 3 demonstrates scale separation in the KdV system. The coarse macroscale mesh cannot resolve the wave structure, making the microscale contribution critical. Interestingly, the macroscale state contains apparent small-scale features beyond what simple smoothing would produce. This arises from our learned convolution kernel, which differs from a standard Gaussian filter as it attempts to approximate sub-grid-scale features (see Appendix J for details).

**Two-dimensional flow over a cylinder**      Our next test case examines a 2D Von Kármán vortex street at $Re = 160$ from Günther et al. [38]. This dataset consists of a nondimensionalized velocity field ($d_u = 2$) developing from a zero initial condition. The spatial domain is truncated to $[-0.5, 3.5] \times [-0.5, 0.5]$ and sampled on a fine $320 \times 80$ grid, resulting in a high-dimensional state ($n_y = 2 \cdot 320 \cdot 80 = 51200$). The cylinder is modeled as a region of zero velocity. The time domain spans $[0, 15]$ with 1501 time steps, which we partition into training $[0, 13]$, validation $(13, 14]$, and test $(14, 15]$ intervals. Our multiscale models use a coarse $32 \times 8$ macroscale grid ($n_\zeta = 2 \cdot 32 \cdot 8 = 512$).

Table 1 demonstrates that our multiscale models significantly outperform the implicit-scale baseline, with the best model ($n_\eta = 5$) reducing error by 86%. Figure 4 reveals how the implicit-scale model fails to resolve the intricate vortex structures in the cylinder wake, whereas the multiscale model captures these flow patterns accurately. Figure 5 provides further insight into the scale separation

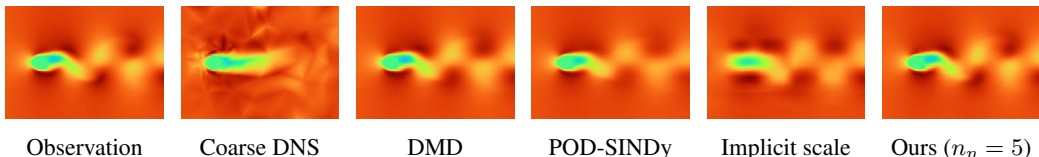

Observation    Coarse DNS    DMD    POD-SINDy    Implicit scale    Ours ($n_\eta = 5$)

Figure 4: Cylinder flow: prediction comparison at $t = 15$ in the test interval. The $x$-component of velocity is shown here, zoomed in on the cylinder.

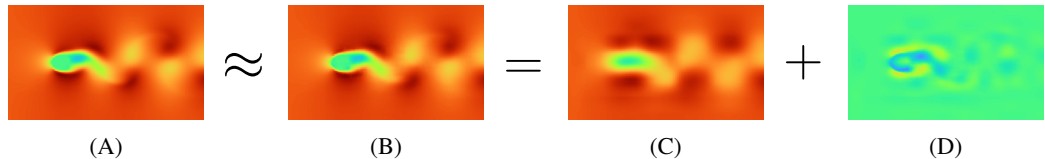

(A)    (B)    (C)    (D)

Figure 5: Cylinder flow: scale separation with $n_\eta = 5$, shown at $t = 15$, zoomed in on the cylinder. The columns are (A) observation from dataset, (B) reconstructed full state, (C) prolonged macroscale state, and (D) decoded microscale state.

mechanism, showing how the microscale state encodes the unresolved vortex structures. Notably, as vortices grow larger toward the right of the domain, the microscale contribution diminishes, demonstrating our model's ability to adaptively allocate representational capacity across scales based on the local physics. The periodic dynamics and relatively fast-decaying Kolmogorov $n$-width in this problem allows DMD to achieve competitive performance (error 0.016), though our multiscale model still achieves 44% lower error.

**Two-dimensional shallow water equations**    Our final test case models a radial dam break using the shallow water equations, from the PDEBench dataset [39]. The spatial domain is $[-2.5, 2.5]^2$ discretized using a $128 \times 128$ grid, resulting in a high-dimensional state ($n_y = 16384$). The fluid elevation field evolves from an initial condition of height 2 within radius $r \sim \mathcal{U}[0.3, 0.7]$ of the origin and height 1 elsewhere. The time domain spans $[0, 1]$ with 101 time steps. We corrupt observations with Gaussian noise (standard deviation 0.01) and partition 1000 trajectories into 900 training, 50 validation, and 50 test samples. Our multiscale models use a coarse $8 \times 8$ macroscale grid ($n_\zeta = 64$).

Table 1 shows our multiscale model ($n_\eta = 5$) achieves 64% error reduction compared to the best baseline (implicit scale), with mean error dropping from 0.426 to 0.152. The coarse DNS exhibits particularly high error (0.861), while reduced-order methods (DMD and POD-SINDy) achieve intermediate performance with error of around 0.53. This problem features slowly-decaying Kolmogorov $n$-width due to significant advection and the diverse ensemble of initial conditions, explaining the challenges faced by linear manifold methods. Figures 6 and 7 illustrate how the microscale representation captures the complex flow patterns that enable this performance improvement.

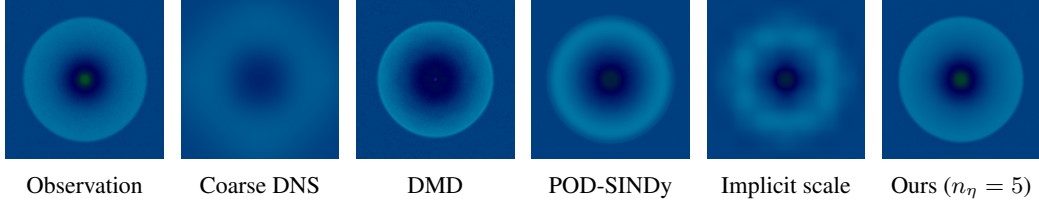

Observation    Coarse DNS    DMD    POD-SINDy    Implicit scale    Ours ($n_\eta = 5$)

Figure 6: Shallow water: prediction comparison on example trajectory from test set at $t = 1$.

**Computational cost studies**    Figure 8 shows the error-cost tradeoffs for online prediction across all test cases. We use the product of number of drift function evaluations (NFE) and state dimension as a measure of computational cost. Our multiscale models achieve favorable error-cost tradeoffs compared to the baselines. Interestingly, increasing $n_\eta$ from 1 to 5 yields significant accuracy gains without proportionally increasing inference cost. This highlights the computational efficiency of our hierarchical structure, which adaptively engages the microscale model only as needed.

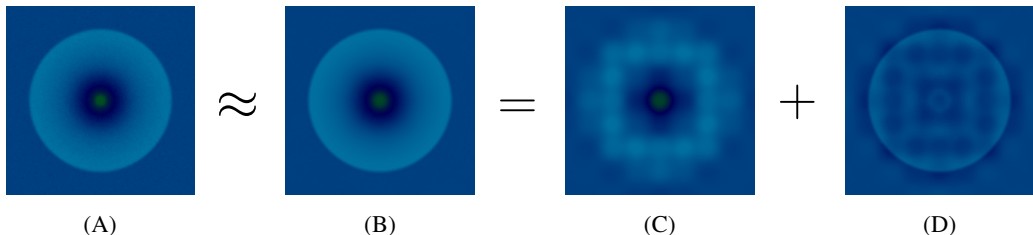

Figure 7: Shallow water: scale separation with $n_\eta = 5$, shown at $t = 1$. The columns are (A) observation from dataset, (B) reconstructed full state, (C) prolonged macroscale state, and (D) decoded microscale state.

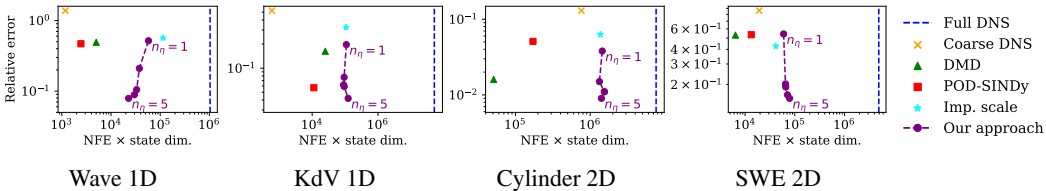

Figure 8: Mean relative error for each model type vs. NFE $\times$ latent state dimension (just state dimension for DNS), as a proxy for online inference cost. The cost is for a single test trajectory.

# 6   Concluding remarks

We have introduced a framework for learning stochastic multiscale models directly from data, where macroscale and microscale states evolve as distinct but interacting dynamical systems. This framing represents a fundamental departure from physics-based and data-driven closure modeling paradigms. Instead of parameterizing the effects of unresolved scales as a corrective term dependent only on the coarse state, we introduce a distinct, dynamical latent state for the microscale itself. This allows our model to capture the complex, state-dependent interplay between scales that characterizes many challenging physical systems.

The performance improvements are substantial: across all test cases, our multiscale models achieved approximately an order of magnitude reduction in error compared to implicit-scale baselines. Our approach also consistently outperformed reduced-order modeling methods (DMD and POD-SINDy), with the performance gap most pronounced on problems with slowly-decaying Kolmogorov $n$-width. Crucially, unlike global ROM approaches that sacrifice spatial structure, our framework maintains a physically interpretable representation on a coarse grid while achieving superior accuracy through the learned microscale corrections.

While our framework shows strong performance on the test problems presented, scaling to more complex systems presents opportunities for methodological advances. While the current global compression of $\eta$ is effective, it may not be optimal for problems like 3D turbulence where sub-grid dynamics are dominated by local structures. Future work could explore spatially-aware representations for the microscale state. Furthermore, extending the framework to chaotic systems will require careful consideration of long-term stability in the learned dynamics. For systems with distinct scale hierarchies, one could envision incorporating multiple nested coarse grids, creating a sequence of latent states corresponding to different scales. While this hierarchical generalization presents non-trivial challenges, it represents a compelling direction for future work.

Looking forward, we believe the potential applications extend to numerous fields where multiscale phenomena present fundamental modeling challenges, including climate modeling [40], astrophysical simulations [41], biological pattern formation [42], and neural dynamics [43]. Ultimately, our work demonstrates a path toward building models that combine the flexibility of data-driven methods with the physically-grounded structure of multiscale formalisms. This synthesis of machine learning and physics is particularly promising for complex systems where first-principles closures remain elusive.

## Acknowledgments

This research is supported by an NSERC Discovery Grant, a Queen Elizabeth II Graduate Scholarship in Science and Technology, and a grant from the Data Sciences Institute at the University of Toronto.

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

## A Macroscale drift parametrization

We parametrize the macroscale drift $f_\theta(\zeta, \phi(\eta))$ to respect the underlying spatial structure of the coarse grid. This is achieved by defining the drift at each grid point as the output of a single, shared learnable function, $\widehat{f}_\theta$, which takes as input the state values from a local neighborhood or "stencil" around that point. This approach ensures that the learned dynamics are translation-equivariant, meaning the same physical rule is applied everywhere on the grid.

For a one-dimensional system (with $d_u = 1$), the drift component at grid point $j$ is defined as

$$[f_\theta(\zeta, \phi(\eta))]_j = \widehat{f}_\theta\left(\zeta_{j-q}, \ldots, \zeta_{j+q}, \phi(\eta)\right) \text{ for } j = 1, \ldots, n_\zeta, \tag{11}$$

where $q \in \mathbb{N}$ defines the stencil half-width, and $\widehat{f}_\theta$ is a neural network with weights that are shared across all positions $j$. Boundary conditions are handled as appropriate for the specific test case. This structure is a learnable, nonlinear generalization of a finite difference scheme. While this stencil-based MLP is highly effective for regular grids, the principle of a shared, local operator can be extended to irregular grids using architectures like graph neural networks.

It is worth noting that if the governing equations are known, $f_\theta$ can be derived using a spatial discretization method, such as a finite difference (FD) scheme, on a coarse macroscale spatial mesh. We do not adopt this approach for two primary reasons.

Firstly, even when an FD scheme is derived directly for the coarse grid spacing $(\Delta x)_{\text{coarse}}$, from the original PDE, its accuracy is limited by truncation errors inherent to coarse discretizations. Standard low-order FD stencils may provide a poor numerical approximation of the original PDE operator on such a grid. Secondly, and more crucially, the effective governing dynamics on a coarse grid often differ significantly in their functional form from the original PDE. The process of coarse-graining means that unresolved subgrid-scale physics exert an influence on the resolved macroscale dynamics. This influence can manifest as complex, state-dependent modifications to the original PDE terms or even as entirely new effective terms (often referred to as closure terms or subgrid-scale parametrizations). A simple FD discretization of the original PDE, regardless of the care taken in choosing stencil coefficients for $(\Delta x)_{\text{coarse}}$, is generally incapable of representing these emergent subgrid-scale effects.

In contrast, by learning the drift function directly from observed trajectories, our approach allows the model to capture these effective coarse-grid dynamics, including any implicit subgrid-scale contributions, without being restricted to the structure of the original PDE. This flexibility is key to accurately modeling multiscale systems on a computationally tractable coarse grid.

To demonstrate that our representation enables interpretable models of macroscale dynamics to be learned, we perform a numerical study on the KdV test problem from Section 5 of the main paper. Figure 9 presents a sensitivity study of the localized drift $\widehat{f}_\theta$ (from Eq. (11) with $q = 2$) for an implicit-scale model. This model, corresponding to our full framework with microscale dimension $n_\eta = 0$ (see "Remarks on closure models and implicit-scale models" in Section 4 of the main paper), was trained on the KdV dataset. For each subplot in Figure 9, corresponding to an input stencil location $k \in \{-2, \ldots, 2\}$ (since $q = 2$ here), the macroscale state component $\zeta_{j+k}$ is varied over $[-1, 1]$, while the other $\zeta$ components within the stencil are held at zero. The resulting learned drift is compared to the drift term derived from a standard FD approximation of the KdV equation (details of the FD scheme are in Appendix H.2).

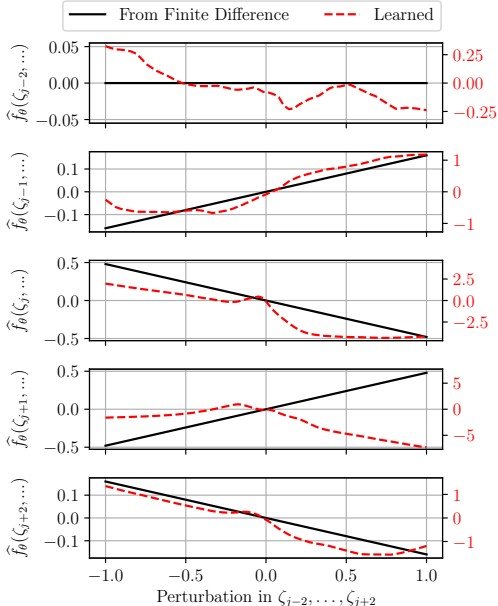

Figure 9: Sensitivity of the localized macroscale drift $\widehat{f}_\theta$ to perturbations of its stencil inputs $\zeta_{j-2}, \ldots, \zeta_{j+2}$. The drift $\widehat{f}_\theta$ is from an implicit-scale model trained on the KdV dataset.

Since the learned macroscale state $\zeta$ may have different magnitudes or units than the original physical state due to the learned encoder, the outputs of both the learned drift and the FD-derived drift are rescaled to a common range in Figure 9 to enable a direct visual comparison.

It can be seen from Figure 9 that the drift derived using a FD scheme, not being optimized for the coarse grid, indeed differs from the learned drift function. The unresolved small-scale features introduce complex effects into the coarse-grid macroscale dynamics (such as apparent nonlinearities or memory effects) which a standard FD scheme cannot capture. These results motivate the present approach where the drift functions are learned directly from data.

## B Validity of Markovian latent dynamics

Our stochastic multiscale modeling framework models the latent dynamics $z = (\zeta, \eta)$ with an Itô SDE, which assumes the process is Markovian. However, when coarse-graining complex physical systems, the dynamics of the resolved variables often exhibit memory effects, rendering them non-Markovian. As mentioned in the Introduction, the Mori-Zwanzig formalism [13] provides a theoretical basis for understanding these memory effects in the context of closure modeling.

The Mori-Zwanzig formalism shows that the *exact* evolution of a set of resolved variables $\overline{y}$ (analogous to our macroscale state $\zeta$) interacting with unresolved variables $\widetilde{y}$ can be described by a generalized Langevin equation:

$$\frac{\mathrm{d}}{\mathrm{d}t}\overline{y}(t) \;=\; \underbrace{\mathcal{M}\overline{y}(t)}_{\text{Markov}} \;-\; \underbrace{\int_0^t K(s)\overline{y}(t-s)\mathrm{d}s}_{\text{Memory}} \;+\; \underbrace{F(t)}_{\text{Noise}}, \tag{12}$$

where $\mathcal{M}\overline{y}$ is a Markovian term that captures the resolved dynamics, $F$ is a noise term arising from the unresolved dynamics associated with $\widetilde{y}$, and the convolution integral involving the memory kernel $K(s)$ captures the non-Markovian feedback from the history of $\overline{y}(t)$ due to its coupling with $\widetilde{y}(t)$.

To obtain a finite-dimensional Markovian representation suitable for learning SDE models, the memory integral in (12) must be approximated. One common approach (e.g., [44]) is to introduce a hierarchy of auxiliary variables $w_0, w_1, \ldots, w_m$ to represent the memory term and its derivatives. For instance, defining $w_0(t) = \int_0^t K(s)\overline{y}(t-s)\mathrm{d}s$, the generalized Langevin equation can be recast as part of an extended, Markovian system:

$$\frac{\mathrm{d}\overline{y}(t)}{\mathrm{d}t} = \mathcal{M}\overline{y}(t) - w_0(t) + F(t) \tag{13}$$

$$\frac{\mathrm{d}w_0(t)}{\mathrm{d}t} = K(0)\overline{y}(t) + \int_0^t K'(s)\overline{y}(t-s)\mathrm{d}s, \tag{14}$$

where $K'$ denotes the time derivative of the kernel. This procedure can be iterated by defining $w_1(t)$ for the integral in (14) and accounting for the dynamics of $w_1$, which leads to

$$\frac{\mathrm{d}}{\mathrm{d}t}\overline{y}(t) = \mathcal{M}\overline{y}(t) - w_0(t) + F(t)$$
$$\frac{\mathrm{d}}{\mathrm{d}t}w_0(t) = K(0)\overline{y}(t) + w_1(t) \tag{15}$$
$$\frac{\mathrm{d}}{\mathrm{d}t}w_1(t) = K'(0)\overline{y}(t) + \int_0^t K''(s)\overline{y}(t-s)\mathrm{d}s,$$

This iterative process generates a hierarchy of auxiliary variables $w_0, w_1, \ldots, w_m$ that collectively represent the memory effects. The augmented state $(\overline{y}, w_0, w_1, \ldots, w_m)$ becomes approximately Markovian if this hierarchy is truncated at a level $m$ where the remaining integral term is negligible. This truncation is an exact representation if the kernel is a polynomial of degree $p-1$, in which case the hierarchy terminates after $p$ auxiliary variables.

In our multiscale modeling framework, we do not explicitly construct these Mori-Zwanzig auxiliary variables. Instead, we posit that our learned latent SDE system for $z = (\zeta, \eta)$ addresses the memory challenge in two ways. Firstly, the explicit microscale state $\eta$ aims to capture some of the fast-evolving components that would otherwise contribute to the memory kernel $K(t)$ in a model for $\zeta$

alone. By conditioning the evolution of $\zeta$ on $\eta$ (and vice-versa) within a coupled Markovian SDE, the effective memory in the $\zeta$ dynamics may be significantly reduced. Secondly, the flexibility of the universal approximators used for the learned drift and diffusion functions of our SDE, operating on the combined state $z$, are sufficiently flexible to implicitly represent any residual short-term memory effects within the chosen finite-dimensional latent space.

In summary, while the Markovian assumption for latent dynamics is an approximation for most physical systems, it can be justified if the learned latent space $z$ is structured and expressive enough to capture the dominant interactions and short-term memory. The success of our model in practice (see Section 5 of the main paper) lends empirical support to this assumption for the systems studied.

## C  Multiscale likelihood derivation

We summarize below the algebraic manipulations used to derive the multiscale log-likelihood in Section 4.1 for the case of product of Gaussian experts. Adopting the PoE perspective [35], we define the following multiscale likelihood:

$$p_\theta(y \mid \zeta, \eta) = \frac{1}{Z(\zeta, \eta)} p_1(y \mid \zeta)\, p_2(y - \mathcal{D}_\theta^\zeta(\zeta) \mid \eta). \tag{16}$$

For the case of Gaussian experts, we have $p_1(y \mid \zeta) = \mathcal{N}(y \mid \mathcal{D}_\theta^\zeta(\zeta), \Sigma_\theta^\zeta)$ and $p_2(y - \mathcal{D}_\theta^\zeta(\zeta) \mid \eta) = \mathcal{N}((y - \mathcal{D}_\theta^\zeta(\zeta)) \mid \mathcal{D}_\theta^\eta(\eta), \Sigma_\theta^\eta)$, i.e.,

$$p_1(y \mid \zeta) = \frac{1}{Z_1} \exp\left(-\frac{1}{2}(y - \mathcal{D}_\theta^\zeta(\zeta))^T (\Sigma_\theta^\zeta)^{-1}(y - \mathcal{D}_\theta^\zeta(\zeta))\right), \tag{17}$$

$$p_2(y - \mathcal{D}_\theta^\zeta(\zeta) \mid \eta) = \frac{1}{Z_2} \exp\left(-\frac{1}{2}((y - \mathcal{D}_\theta^\zeta(\zeta)) - \mathcal{D}_\theta^\eta(\eta))^T (\Sigma_\theta^\eta)^{-1}((y - \mathcal{D}_\theta^\zeta(\zeta)) - \mathcal{D}_\theta^\eta(\eta))\right). \tag{18}$$

where $Z_1 = (2\pi)^{n_y/2}|\Sigma_\theta^\zeta|^{1/2}$ and $Z_2 = (2\pi)^{n_y/2}|\Sigma_\theta^\eta|^{1/2}$ are the normalization constants of the two Gaussian PDFs.

We define the precision matrices as $S_\theta^\zeta = (\Sigma_\theta^\zeta)^{-1}$ and $S_\theta^\eta = (\Sigma_\theta^\eta)^{-1}$. For compactness in the subsequent derivation of the product, we introduce the temporary notational shorthands: $\mu_1 = \mathcal{D}_\theta^\zeta(\zeta)$ and $\mu_2 = \mathcal{D}_\theta^\zeta(\zeta) + \mathcal{D}_\theta^\eta(\eta)$. Using these definitions, the product of the two Gaussian PDFs can be written as

$$p_1(y \mid \zeta) \cdot p_2(y - \mathcal{D}_\theta^\zeta(\zeta) \mid \eta) = \frac{1}{K} \exp\left(-\frac{1}{2}(y - \mu_1)^T S_\theta^\zeta(y - \mu_1) - \frac{1}{2}(y - \mu_2)^T S_\theta^\eta(y - \mu_2)\right)$$

$$= \frac{e^C}{K} \exp\left(-\frac{1}{2}(y - \mu_y)^T S_y(y - \mu_y)\right), \tag{19}$$

where $S_y = S_\theta^\zeta + S_\theta^\eta$, $\mu_y = S_y^{-1}(S_\theta^\zeta \mu_1 + S_\theta^\eta \mu_2)$, $C = \frac{1}{2}\mu_y^T S_y \mu_y - \frac{1}{2}(\mu_1^T S_\theta^\zeta \mu_1 + \mu_2^T S_\theta^\eta \mu_2)$, and $K = (2\pi)^{n_y}|\Sigma_\theta^\zeta|^{1/2}|\Sigma_\theta^\eta|^{1/2}$.

The normalization constant of the product of Gaussians is given by

$$Z(\zeta, \eta) = \frac{e^C}{K} \int \exp\left(-\frac{1}{2}(y - \mu_y)^T S_y(y - \mu_y)\right) dy = \frac{e^C}{K}(2\pi)^{n_y/2}|S_y^{-1}|^{1/2}, \tag{20}$$

which yields the following expression for the normalized PoE likelihood

$$p_\theta(y \mid \zeta, \eta) = (2\pi)^{-n_y/2}|S_y^{-1}|^{-1/2} \exp\left(-\frac{1}{2}(y - \mu_y)^T S_y(y - \mu_y)\right). \tag{21}$$

Recalling our definitions $\mu_1 = \mathcal{D}_\theta^\zeta(\zeta)$, $\mu_2 = \mathcal{D}_\theta^\zeta(\zeta) + \mathcal{D}_\theta^\eta(\eta)$, and using the resulting expression for $\mu_y = S_y^{-1}(S_\theta^\zeta \mu_1 + S_\theta^\eta \mu_2)$, the log-likelihood can be written as

$$
\begin{aligned}
\log p_\theta(y \,|\, \zeta, \eta) = & -\frac{1}{2}(y - \mu_y)^T S_y (y - \mu_y) - \frac{n_y}{2}\log(2\pi) + \frac{1}{2}\log|S_y| \\
= & -\frac{1}{2}\|r(\zeta)\|_{S_\theta^\zeta}^2 - \frac{1}{2}\|r(\zeta) - \mathcal{D}_\theta^\eta(\eta)\|_{S_\theta^\eta}^2 + \frac{1}{2}\|\mathcal{D}_\theta^\eta(\eta)\|_\Lambda^2 \\
& -\frac{n_y}{2}\log(2\pi) + \frac{1}{2}\log|S_\theta^\zeta + S_\theta^\eta|,
\end{aligned}
\tag{22}
$$

where $\Lambda = S_\theta^\eta(S_\theta^\eta + S_\theta^\zeta)^{-1}S_\theta^\zeta$ and $r(\zeta) = y - \mathcal{D}_\theta^\zeta(\zeta)$ denotes the residual between the full observation and the macroscale model prediction.

To derive the expression for the log-likelihood in (6) which provides valuable insights into the regularization terms arising from the PoE likelihood, we first simplify the sum of the second and third terms in (22) as follows:

$$
\begin{aligned}
-\frac{1}{2}\|r(\zeta) - \mathcal{D}_\theta^\eta(\eta)\|_{S_\theta^\eta}^2 + \frac{1}{2}\|\mathcal{D}_\theta^\eta(\eta)\|_\Lambda^2 = & -\frac{1}{2}r(\zeta)^T S_\theta^\eta r(\zeta) + r(\zeta)^T S_\theta^\eta \mathcal{D}_\theta^\eta(\eta) \\
& -\frac{1}{2}\mathcal{D}_\theta^\eta(\eta)^T S_\theta^\eta \mathcal{D}_\theta^\eta(\eta) + \frac{1}{2}\mathcal{D}_\theta^\eta(\eta)^T \Lambda \mathcal{D}_\theta^\eta(\eta) \\
= & -\frac{1}{2}r(\zeta)^T S_\theta^\eta r(\zeta) + r(\zeta)^T S_\theta^\eta \mathcal{D}_\theta^\eta(\eta) \\
& -\frac{1}{2}\mathcal{D}_\theta^\eta(\eta)^T (S_\theta^\eta - \Lambda)\mathcal{D}_\theta^\eta(\eta).
\end{aligned}
\tag{23}
$$

Noting that

$$
\begin{aligned}
\Lambda' = S_\theta^\eta - \Lambda = & \, S_\theta^\eta - S_\theta^\eta(S_\theta^\eta + S_\theta^\zeta)^{-1}S_\theta^\zeta = S_\theta^\eta\left[I - (S_\theta^\eta + S_\theta^\zeta)^{-1}S_\theta^\zeta\right] \\
= & \, S_\theta^\eta\left[(S_\theta^\eta + S_\theta^\zeta)^{-1}(S_\theta^\eta + S_\theta^\zeta) - (S_\theta^\eta + S_\theta^\zeta)^{-1}S_\theta^\zeta\right] = S_\theta^\eta(S_\theta^\eta + S_\theta^\zeta)^{-1}S_\theta^\eta
\end{aligned}
\tag{24}
$$

is an SPD matrix, the last term in (23) can written as $-(1/2)\|\mathcal{D}_\theta^\eta(\eta)\|_{\Lambda'}^2$ which is non-positive by definition. Substituting (23) into (22) and combining the first two terms yields the expression for the log-likelihood in (6) that we reproduce here for clarity:

$$
\begin{aligned}
\log p_\theta(y \,|\, \zeta, \eta) = & -\frac{1}{2}\|r(\zeta)\|_{(S_\theta^\zeta + S_\theta^\eta)}^2 + (r(\zeta), \mathcal{D}_\theta^\eta(\eta))_{S_\theta^\eta} - \frac{1}{2}\|\mathcal{D}_\theta^\eta(\eta)\|_{\Lambda'}^2 \\
& -\frac{n_y}{2}\log(2\pi) + \frac{1}{2}\log|S_\theta^\zeta + S_\theta^\eta|.
\end{aligned}
$$

The term $(1/2)\|\mathcal{D}_\theta^\eta(\eta)\|_{\Lambda'}^2$ implements an automatic scale separation mechanism by strongly penalizing microscale components in directions already well-represented by the macroscale model and allowing the microscale state more freedom in directions poorly captured by the macroscale model. This creates an "adaptive regularization" effect, wherein the regularization strength for each component of the microscale state is automatically calibrated based on the relative confidence in macroscale versus microscale predictions for that component, as encoded in the precision matrices $S_\theta^\zeta$ and $S_\theta^\eta$. This is performed automatically based on the learned precision matrices without requiring manual specification of scale boundaries.

## D  Multiple trajectory training

The SVI approach presented in Section 4 is formulated for a dataset with a single time series. In this section, we consider the case of multiple trajectories with potentially different initial conditions. We assume that the multiscale model parameters are shared across all time series. We denote the multiple-trajectory dataset as $\mathcal{Y} = \{\mathcal{Y}_k\}_{k=1}^{n_{\mathrm{tr}}}$, where $\mathcal{Y}_k$ is the $k$-th trajectory $\{t_{ik}, y_{ik}\}_{i=1}^{n_{t,k}}$, and $n_{\mathrm{tr}}$ is the number of trajectories. The $i$-th time step of the $k$-th trajectory is denoted $t_{ik}$, and the corresponding observation is $y_{ik}$. The multiple-trajectory ELBO is then given by simply including an extra summation in (8),

$$
\begin{aligned}
\mathcal{L}(\varphi) = & \sum_{k=1}^{n_{\mathrm{tr}}} \sum_{i=1}^{n_{t,k}} \mathbb{E}_{z_{ik},\theta}[\log p_\theta(y_{ik} \,|\, z_{ik})] - \frac{1}{2}\sum_{k=1}^{n_{\mathrm{tr}}} \int_{t_0}^{T_k} \mathbb{E}_{z_k(t),\theta}\|r_{\theta,\varphi}(z_k(t), t)\|_{C(t)}^2 \, dt \\
& - D_{\mathrm{KL}}\left(q_\varphi(\theta) \,\|\, p(\theta)\right),
\end{aligned}
\tag{25}
$$

where $T_k$ is the final time of the $k$-th trajectory, $\theta \sim q_\varphi(\theta)$, $z_{ik} \sim q_\varphi(z_k \,|\, t_{ik})$, and $z_k(t) \sim q_\varphi(z_k \,|\, t)$. The log-likelihood term is evaluated at each observation time $t_{ik}$ of each trajectory $k$ and summed over all observations. The drift residual term is evaluated over the entire time span of each trajectory.

## E  Reparametrized, amortized ELBO

In this section, we outline how our multiscale SVI scheme, presented in Section 4 of the main paper, leverages the reparametrization trick and amortized SVI scheme proposed by Course and Nair [37, 12] for efficient training.

A key challenge in evaluating the ELBO defined in (8) of the main paper is that we require a SDE solver to generate sample trajectories of the linear variational SDE (7). The reparametrization trick, proposed in [37], allows us to circumvent the need for an SDE solver in the training loop. The core idea is that the marginal distribution of the variational linear SDE takes the form $q_\varphi(z \,|\, t) = \mathcal{N}(\mu_\varphi(t),\, \Sigma_\varphi(t))$, where the dynamics of $\mu_\varphi$ and $\Sigma_\varphi$ are governed by the following ODEs[5]

$$\dot{\mu}_\varphi(t) = -A_\varphi(t)\mu_\varphi(t) + b_\varphi(t), \quad \mu_\varphi(0) = \mu_0,$$
$$\dot{\Sigma}_\varphi(t) = -A_\varphi(t)\Sigma_\varphi(t) - \Sigma_\varphi(t)A_\varphi(t)^T + L_\theta(t)L_\theta(t)^T, \quad \Sigma_\varphi(0) = \Sigma_0. \tag{26}$$

The reparametrization trick in [37] leverages (26) to express $A_\varphi$ and $b_\varphi$, in terms of $\mu_\varphi, \Sigma_\varphi$ and their time derivatives. This, in turn, allows the drift residual to be rewritten as $r_{\theta,\varphi}(z(t),t) = \dot{\mu}_\varphi(t) - B(t)(z(t) - \mu_\varphi(t)) - \gamma_\theta(z(t))$, where $B(t) = \mathcal{V}^{-1}((\Sigma_\varphi(t) \oplus \Sigma_\varphi(t))^{-1}\mathcal{V}(L_\theta(t)L_\theta(t)^T - \dot{\Sigma}_\varphi(t)))$, $\oplus$ denotes the Kronecker sum, the operator $\mathcal{V} : \mathbb{R}^{n \times n} \to \mathbb{R}^{n^2}$ stacks the columns of a matrix into a vector, and $\mathcal{V}^{-1} : \mathbb{R}^{n^2} \to \mathbb{R}^{n \times n}$ does the opposite. Consequently, by parametrizing the variational distribution $q_\varphi(z \,|\, t)$ directly via $\mu_\varphi$ and $\Sigma_\varphi$, all the ELBO terms can be evaluated by drawing samples from the multivariate Gaussian $\mathcal{N}(\mu_\varphi(t),\, \Sigma_\varphi(t))$. This allows the ELBO to be maximized without an SDE solver in the training loop, thereby decoupling the computational cost of training from the stiffness of the variational SDE.

To address the challenge of learning the variational distribution $q_\varphi(z \,|\, t)$ (i.e., $\mu_\varphi(t), \Sigma_\varphi(t)$) over long time horizons, we adopt the amortization strategy from [12]. This involves partitioning the time series into shorter segments and learning the variational distribution locally for each segment. To illustrate this approach, consider a trajectory with observation times $\{t_1, \ldots, t_m, t_{m+1}, \ldots, t_{n_t}\}$, where $t_1 = t_0$ and $t_{n_t} = T$. We partition this sequence into $n_m = \lceil n_t/m \rceil$ non-overlapping segments (sub-intervals) as follows[6]

$$\left\{ \mathcal{I}_k = [t_1^{(k)}, t_2^{(k)}, \ldots, t_{m_k}^{(k)}] \right\}_{k=1}^{n_m}, \tag{27}$$

where $t_1^{(k)} = t_{m_{k-1}}^{(k-1)}$ (for $k > 1$), $m_k = m$ for $k < n_m$, and $m_{n_m} \leq m$. The variational distribution is then amortized over these $n_m$ segments, leading to the reformulated ELBO:

$$\mathcal{L}(\varphi) = \sum_{k=1}^{n_m} \left( \sum_{j=1}^{m_k} \mathbb{E}_{z_j^{(k)},\theta} \left[ \log p_\theta \left( y_j^{(k)} \,|\, z_j^{(k)} \right) \right] - \frac{1}{2} \int_{t_1^{(k)}}^{t_{m_k}^{(k)}} \mathbb{E}_{z(t),\theta} \| r_{\theta,\varphi}(z(t),t) \|_{C(t)}^2 \, \mathrm{d}t \right)$$
$$- D_{\mathrm{KL}}\left( q_\varphi(\theta) \,\|\, p(\theta) \right), \tag{28}$$

where $\theta \sim q_\varphi(\theta)$, $z_j^{(k)} \sim q_\varphi(z \,|\, t_j^{(k)})$, $z(t) \sim q_\varphi(z \,|\, t)$, and $y_j^{(k)}$ is the $j$-th observation in the $k$-th segment, corresponding to time $t_j^{(k)}$.

Within each segment $\mathcal{I}_k$, the parameters of the variational distribution, $\mu_\varphi$ and $\Sigma_\varphi$, are defined as follows: At the discrete observation times $t_j^{(k)}$ within the segment, $\mu_\varphi(t_j^{(k)})$ and $\Sigma_\varphi(t_j^{(k)})$ are given by applying the probabilistic encoder (defined in Section 3) to the observation $y_j^{(k)}$. To obtain a continuous-time representation for $\mu_\varphi$ and $\Sigma_\varphi$ (and their derivatives $\dot{\mu}_\varphi, \dot{\Sigma}_\varphi$) for $t \in [t_1^{(k)}, t_{m_k}^{(k)}]$, which is necessary for evaluating the integral in (28), we employ a deep kernel interpolation scheme that operates on these encoder outputs, following [12]. The trajectory partitioning parameter $m \in \mathbb{N}$ is a hyperparameter of the training process.

---

[5]  Recall that the initial conditions $\mu_{t_0}, \Sigma_{t_0}$ are given by the probabilistic encoder $p_\theta(z \,|\, y(t_0))$, as discussed in Section 4.2 of the main paper.

[6]  Note that when $n_t$ is not divisible by $m$, the last interval has fewer than $m$ elements.

# F   Computing the posterior predictive distribution

After the model parameters are estimated using the multiscale variational inference framework, the primary objective during inference is to generate probabilistic predictions over a given time horizon. Specifically, given an initial observation of the full state $y(t_0) \in \mathbb{R}^{n_y}$ at time $t_0$, we seek the *posterior predictive distribution* $p_\theta(y(t) \,|\, y(t_0))$ for any $t > t_0$. This distribution quantifies our prediction for $y(t)$ and accounts for all sources of uncertainty captured by the model, including the initial state decomposition, the stochastic evolution of the latent variables over the interval $[t_0, t]$, and the final state reconstruction. The prediction process involves three fundamental stages that bridge the fully-resolved observation ($y$) and the latent state $z = [\zeta^T, \eta^T]^T \in \mathbb{R}^{n_z}$.

**Stage 1: Probabilistic encoding of the initial state**   The inference process commences by mapping the initial observation $y(t_0)$ into the latent space defined by the macroscale state $\zeta \in \mathbb{R}^{n_\zeta}$ and the microscale state $\eta \in \mathbb{R}^{n_\eta}$. Since our model posits a latent representation, the true initial latent state $z(t_0)$ corresponding to $y(t_0)$ is inherently uncertain. The learned probabilistic encoder $p_\theta(z \,|\, y)$ provides the distribution over possible initial latent states at time $t_0$:

$$p_\theta(z(t_0) \,|\, y(t_0)) = \mathcal{N}(z(t_0) \,|\, \mu_{t_0}, \Sigma_{t_0}), \tag{29}$$

where $\mu_{t_0} \in \mathbb{R}^{n_z}$ and $\Sigma_{t_0} \in \mathbb{R}^{n_z \times n_z}$ are determined by the learned encoder applied to $y(t_0)$:

$$\mu_{t_0} = \mathbb{E}[z(t_0) \,|\, y(t_0)] = \begin{bmatrix} \mathcal{E}_\theta^\zeta(\mathcal{S}_\theta(y(t_0))) \\ \mathcal{E}_\theta^\eta(\mathcal{S}_\theta^\perp(y(t_0))) \end{bmatrix}, \quad \Sigma_{t_0} = \mathrm{Cov}(z(t_0) \,|\, y(t_0)) = \Sigma_\theta^z.$$

Here, $\Sigma_\theta^z$ represents the learned covariance matrix associated with the encoder, capturing the uncertainty introduced by the initial scale separation ($\mathcal{S}_\theta, \mathcal{E}_\theta^\zeta$) and microscale compression ($\mathcal{E}_\theta^\eta$). This initial distribution $\mathcal{N}(z(t_0) \,|\, \mu_{t_0}, \Sigma_{t_0})$ serves as the starting point for evolving the latent dynamics over the time interval $(t_0, t]$.

**Stage 2: Evolving latent state uncertainty via the learned multiscale SDE**   The core of the model lies in the learned coupled system of multiscale Itô SDEs governing the evolution of the latent macroscale ($\zeta$) and microscale ($\eta$) states, represented compactly as:

$$dz = \gamma_\theta(z(s))ds + L_\theta(s)d\beta(s), \quad s \in [t_0, t], \tag{30}$$

where $\gamma_\theta(z) \in \mathbb{R}^{n_z}$ is the learned drift function, $L_\theta(s) \in \mathbb{R}^{n_z \times n_z}$ is the learned time-dependent diffusion matrix, and $\beta \in \mathbb{R}^{n_z}$ is a standard Wiener process. Propagating the initial distribution $p_\theta(z(t_0) \,|\, y(t_0))$ through this SDE from time $t_0$ to $t$ yields the distribution of the latent state at the prediction time $t$, denoted by $p_\theta(z(t) \,|\, y(t_0))$. It is worth noting that even if the initial distribution $p_\theta(z(t_0) \,|\, y(t_0))$ is Gaussian, the distribution $p_\theta(z(t) \,|\, y(t_0))$ after evolution through the nonlinear SDE (30) is *non-Gaussian*. Its probability density function, formally governed by the Fokker-Planck equation associated with (30), rarely admits a closed-form analytical solution. We will show next how this challenge can be addressed using Monte Carlo sampling.

**Stage 3: Probabilistic reconstruction using the PoE likelihood**   The final stage connects the distribution of the latent state at time $t$, i.e., $p_\theta(z(t) \,|\, y(t_0))$, back to the observable state space $y(t) \in \mathbb{R}^{n_y}$. This mapping is defined by the likelihood model $p_\theta(y \,|\, z)$ learned during training. As described in Section 4.1 and Appendix C, this likelihood leverages the PoE perspective to enforce scale separation. In brief, the combination of the macroscale expert $p_1(y \,|\, \zeta) = \mathcal{N}(y \,|\, \mathcal{D}_\theta^\zeta(\zeta), \Sigma_\theta^\zeta)$ and the microscale expert $p_2(y - \mathcal{D}_\theta^\zeta(\zeta)|\eta) = \mathcal{N}(y - \mathcal{D}_\theta^\zeta(\zeta) \,|\, \mathcal{D}_\theta^\eta(\eta), \Sigma_\theta^\eta)$ results in a single Gaussian conditional distribution for $y$ given a specific latent state $z$:

$$p_\theta(y \,|\, z) = \mathcal{N}(y \,|\, \mu_y(z), \Sigma_y), \tag{31}$$

where the conditional mean $\mu_y(z) \in \mathbb{R}^{n_y}$ and conditional covariance $\Sigma_y \in \mathbb{R}^{n_y \times n_y}$ are

$$\mu_y(z) = \mathbb{E}[y \,|\, z] = \mathcal{D}_\theta^\zeta(\zeta) + \Sigma_y S_\theta^\eta \, \mathcal{D}_\theta^\eta(\eta) \text{ and } \Sigma_y = \mathrm{Cov}(y \,|\, z) = (S_\theta^\zeta + S_\theta^\eta)^{-1}. \tag{32}$$

Here, $S_\theta^\zeta = (\Sigma_\theta^\zeta)^{-1}$ and $S_\theta^\eta = (\Sigma_\theta^\eta)^{-1}$ are the precision matrices (parametrized by $\theta$) associated with the Gaussian macroscale and microscale experts, respectively, $\mathcal{D}_\theta^\zeta : \mathbb{R}^{n_\zeta} \to \mathbb{R}^{n_y}$ is the macroscale decoder, and $\mathcal{D}_\theta^\eta : \mathbb{R}^{n_\eta} \to \mathbb{R}^{n_y}$ denotes the microscale decoder.

The PoE likelihood introduces two key statistical nuances. Firstly, the conditional mean $\mu_y(z)$ in (32) is *not* simply the sum of the decoded macrostate $\mathcal{D}_\theta^\zeta(\zeta)$ and the decoded microstate $\mathcal{D}_\theta^\eta(\eta)$. The contribution from the microscale decoder $\mathcal{D}_\theta^\eta(\eta)$ is adaptively weighted by the matrix $W = \Sigma_y S_\theta^\eta = (S_\theta^\zeta + S_\theta^\eta)^{-1} S_\theta^\eta$. This weighting factor emerges naturally from combining the Gaussian experts in the PoE framework and depends on the learned relative precisions $(S_\theta^\zeta, S_\theta^\eta)$. This ensures that the microscale state primarily contributes to explaining aspects of $y$ where the macroscale expert has low precision. Secondly, the conditional covariance $\Sigma_y$ in (32), which is independent of $z$, quantifies the inherent uncertainty in reconstructing $y$ even if $z$ were known perfectly. This uncertainty stems from the combination and potential disagreement between the macroscale and microscale experts, as encoded in their respective covariance matrices $\Sigma_\theta^\zeta$ and $\Sigma_\theta^\eta$.

The final predictive distribution $p_\theta(y(t) \mid y(t_0))$ is obtained by marginalizing the intermediate latent state $z(t)$, which yields

$$p_\theta(y(t) \mid y(t_0)) = \int p_\theta(y(t) \mid z(t)) \, p_\theta(z(t) \mid y(t_0)) \, \mathrm{d}z(t)$$

$$= \int \mathcal{N}(y(t) \mid \mu_y(z(t)), \Sigma_y) \, p_\theta(z(t) \mid y(t_0)) \, \mathrm{d}z(t), \tag{33}$$

where in the second step we used the PoE likelihood (31). The preceding integral is the expectation of the conditional likelihood $p_\theta(y(t) \mid z(t))$ with respect to the distribution of the latent state $p_\theta(z(t) \mid y(t_0))$ obtained by propagating the initial condition provided at $t_0$ through the latent SDE (30). Unfortunately, this integral is analytically intractable due to the nonlinearity of the conditional mean $\mu_y(z(t))$ and the non-Gaussian nature of the distribution $p_\theta(z(t) \mid y(t_0))$. In the present work, we use Monte Carlo sampling to approximate this integral and characterize the posterior predictive distribution as follows:

1. *Sample initial latent states:* Draw $N > 1$ independent samples from the initial latent distribution defined in Eq. (29), i.e., $z_i(t_0) \sim p_\theta(z(t_0) \mid y(t_0)) = \mathcal{N}(\mu_{t_0}, \Sigma_{t_0})$, for $i = 1, \ldots, N$.

2. *Propagate samples via SDE:* Numerically integrate the learned SDE (30) forward in time from $t_0$ to $t$ for each initial sample $z_i(t_0)$. This yields $N$ samples of the latent state $\{z_i(t)\}_{i=1}^N$, which collectively form an empirical approximation of the intractable distribution $p_\theta(z(t) \mid y(t_0))$ at time $t$.

3. *Calculate conditional means:* For each latent sample $z_i(t)$, compute the corresponding conditional mean of the observed state $y(t)$ using (32), i.e., $\mu_{y,i} = \mu_y(z_i(t)) = \mathcal{D}_\theta^\zeta(\zeta_i(t)) + \Sigma_y S_\theta^\eta \mathcal{D}_\theta^\eta(\eta_i(t))$.

4. *Approximate predictive distribution:* The posterior predictive distribution $p_\theta(y(t) \mid y(t_0))$ is approximated by the empirical distribution derived from these samples. Specifically, it can be viewed as a Gaussian mixture model with $N$ components: $p_\theta(y(t) \mid y(t_0)) \approx \frac{1}{N} \sum_{i=1}^N \mathcal{N}(y(t) \mid \mu_{y,i}, \Sigma_y)$, where each component $\mathcal{N}(y(t) \mid \mu_{y,i}, \Sigma_y)$ in this mixture represents the distribution of $y(t)$ conditioned on a specific realization $z_i(t)$ of the latent state evolution. All components share the same conditional covariance $\Sigma_y$ (the reconstruction uncertainty from the PoE), but are centered at potentially different means $\mu_{y,i}$ reflecting the uncertainty propagated through the latent dynamics.

**Predictive mean and covariance** The overall moments of the posterior predictive distribution can be estimated from the Gaussian mixture approximation. The predictive mean is the expected value of $y(t)$ given the initial observation $y(t_0)$, which can be estimated by taking the sample average of the conditional means $\mu_{y,i}$ obtained from each propagated latent state sample $z_i(t)$, i.e.,

$$\widehat{\mu}_{pred}(t) = \mathbb{E}[y(t) \mid y(t_0)] \approx \frac{1}{N} \sum_{i=1}^N \mathbb{E}[y(t) \mid z_i(t)] = \frac{1}{N} \sum_{i=1}^N \mu_{y,i}. \tag{34}$$

The predictive covariance which is the total covariance matrix of $y(t)$ given $y(t_0)$, capturing the overall prediction uncertainty at time $t$ can similarly be estimated as:

$$\widehat{\Sigma}_{pred}(t) = \mathrm{Cov}(y(t) \mid y(t_0)) = \mathbb{E}_{z(t)}[\mathrm{Cov}(y(t) \mid z(t))] + \mathrm{Cov}_{z(t)}[\mathbb{E}(y(t) \mid z(t))]$$

$$\approx \frac{1}{N} \sum_{i=1}^N \Sigma_y + \frac{1}{N-1} \sum_{i=1}^N (\mu_{y,i} - \widehat{\mu}_{pred}(t))(\mu_{y,i} - \widehat{\mu}_{pred}(t))^T. \tag{35}$$

$$= \Sigma_y + \mathrm{SampleCov}(\{\mu_{y,i}\}_{i=1}^N).$$

Table 2: POD-SINDy hyperparameters.

| Hyperparameter | Wave 1D ($n_z = 25$) | KdV 1D ($n_z = 25$) | Burgers 2D ($n_z = 69$) | Cylinder 2D ($n_z = 517$) | SWE 2D ($n_z = 69$) |
|---|---|---|---|---|---|
| Polynomial order | 1 | 2 | 1 | 1 | 1 |
| Threshold | 0.1 | 0.01 | 0.03 | 0.1 | 0.01 |

The total predictive covariance is given by the sum of two distinct terms: (i) the average *reconstruction uncertainty*, $\Sigma_y = (S_\theta^\zeta + S_\theta^\eta)^{-1}$, which is independent of the latent state propagation reflecting the precision of the combined experts, and (ii) the *propagated uncertainty*, $\mathrm{SampleCov}(\{\mu_{y,i}\})$, which arises from the variability of the latent state samples $\{z_i(t)\}$ incorporating both the initial uncertainty encoded in $\Sigma_{t_0}$ and the stochasticity introduced by the SDE dynamics over $(t_0, t]$. The proposed Monte Carlo sampling procedure provides a practical and statistically principled method for generating predictions and quantifying their uncertainty at any future time $t > t_0$. It fully leverages the structure of the learned stochastic multiscale model and the scale-separating PoE likelihood, without requiring further analytical approximations beyond the finite number of samples $N$.

## G Coarse DNS, POD-SINDy, and DMD baselines

In this section, we discuss the setup for three of our baseline methods in each test case: coarse DNS, POD-SINDy, and DMD. The coarse DNS baseline involves simply coarsening the computational mesh used to evaluate the full-order model that generated each dataset. This is straightforward for the wave, KdV, and Burgers test cases, as we created those test cases, and for the shallow water case, as the model that generated the data is publicly available in the associated code repository [39]. As the original solver for the cylinder flow dataset was unavailable, we implemented the full-order model using FEniCS[7] based on the description given by Günther et al. [38]; this code is available in the git repository of this paper. A cubic interpolation scheme was used to map the coarse grid DNS predictions to the fine mesh for computing error metrics.

For the POD-SINDy baseline, implemented using PySINDy [47], we first projected the training data onto a POD basis. The latent dimension was set equal to the total latent state dimension of our model to ensure a fair comparison of dynamics at the same level of compression. We use a polynomial basis with thresholded least-squares to learn the latent dynamics. We performed a hyperparameter search over the polynomial order $\in \{1, 2\}$ and the sparsity threshold $\in \{0.1, 0.03, 0.01, 0.003, 0.001\}$, selecting the combination that minimized validation error. Due to the high-dimensional latent space for the 2D test cases, the number of basis functions becomes impractically large when using quadratic polynomials. We therefore only consider linear latent dynamics for the 2D test problems. The final hyperparameters for each case are reported in Table 2.

For DMD, we use the package PyDMD[8] to learn a discrete-time linear model. The rank of the learned operator is chosen to be $n_z$, and the Tikhonov regularization parameter is fixed at $0.01$ for all cases.

## H Detailed setups for test cases

In this section, we provide detailed setups for the test cases, as well as their respective multiscale model architectures and training procedures.

---

[7] Igor A. Baratta, Joseph P. Dean, Jørgen S. Dokken, Michal Habera, Jack S. Hale, Chris N. Richardson, Marie E. Rognes, Matthew W. Scroggs, Nathan Sime, and Garth N. Wells. DOLFINx: The next generation FEniCS problem solving environment, December 2023

[8] Sara M. Ichinaga, Francesco Andreuzzi, Nicola Demo, Marco Tezzele, Karl Lapo, Gianluigi Rozza, Steven L. Brunton, and J. Nathan Kutz. PyDMD: A python package for robust dynamic mode decomposition. *Journal of Machine Learning Research*, 25(417):1–9, 2024

## H.1 One-dimensional advecting wave

The data for this test case is generated from the 1D advection equation defined over the domain $x \in [0, 1]$ with periodic boundary conditions:

$$\frac{\partial u}{\partial t} = -c \frac{\partial u}{\partial x}, \tag{36}$$

where $c = 1$. The initial condition is $u(x, 0) = \exp(-(x/w)^2)$, where $w \sim \mathcal{U}[0.01, 0.02]$.

To generate the training trajectories, we use a spatial grid with $n_y = 1000$ points ($\Delta x = 0.001$) and approximate the spatial derivative with a centered finite-difference scheme resulting in the system of ODEs

$$\frac{\mathrm{d}u_j}{\mathrm{d}t} = -c \frac{u_{j+1} - u_{j-1}}{2\Delta x}, \tag{37}$$

for $j = 1, 2, \ldots, n_y$, where $u_j$ denotes the solution at the $j$-th spatial gridpoint. The system of ODEs is integrated over the interval $[0, 1]$ using a time step of $\Delta t = 0.001$ ($n_t = 1001$). The final observational data is created by corrupting these ground-truth trajectories with i.i.d. zero-mean Gaussian noise ($\sigma = 0.001$). The training dataset comprises 20 trajectories, with 5 each for validation and testing. The test set covers a shorter time domain of $t \in [0, 0.2]$. For our multiscale models, the macroscale state is defined on a coarse mesh with $n_\zeta = 20$ points.

## H.2 Korteweg-de Vries equation

The Korteweg-de Vries (KdV) equation in 1D is given by

$$\frac{\partial u}{\partial t} + u \frac{\partial u}{\partial x} + \nu \frac{\partial^3 u}{\partial x^3} = 0, \tag{38}$$

where $\nu = 0.02$ and the domain is $x \in [0, 10]$ with periodic boundary conditions. The initial condition is $u(x, 0) = a \cos(\pi x) \exp\left(-\frac{(x-7.5)^2}{s^2}\right)$, where $a \sim \mathcal{N}(2, 0.01)$ and $s \sim \mathcal{N}(1, 0.01)$.

We generate the observational data using a spatial grid with $n_y = 1000$ points ($\Delta x = 0.01$) and approximate the spatial derivatives using a finite-difference scheme[9] leading to the system of ODEs

$$\frac{\mathrm{d}u_j}{\mathrm{d}t} = -\frac{u_{j+1} - u_{j-1}}{2\Delta x} u_j - \nu \frac{u_{j+2} - 3u_{j+1} + 3u_j - u_{j-1}}{\Delta x^3}, \tag{39}$$

for $j = 1, \ldots, n_y$, where $u_j$ denotes the solution at the $j$-th spatial gridpoint. The system of ODEs is integrated over the time interval $t \in [0, 1]$ using a time step of $\Delta t = 0.001$ ($n_t = 1001$). The observations are corrupted by i.i.d. zero-mean Gaussian noise with standard deviation 0.01. The training dataset is formed using 10 samples of initial conditions and their corresponding trajectories, and the validation and test sets are formed using 5 samples each. For our multiscale models, we use a coarse mesh with $n_\zeta = 20$ points for the macroscale state.

## H.3 Burgers' equation in 2D

As an additional test case not included in the main body for brevity, we evaluate the models on the 2D viscous Burgers' equation:

$$\frac{\partial u}{\partial t} + u \frac{\partial u}{\partial x_1} + u \frac{\partial u}{\partial x_2} = \nu \nabla^2 u. \tag{40}$$

The domain is $[0, 1]^2$ with zero Dirichlet boundary conditions and viscosity $\nu = 0.005$. The initial condition is given by $u(x_1, x_2, 0) = a \exp\left(-\frac{(x_1-0.3)^2 + (x_2-0.3)^2}{s^2}\right)$, where $a \sim \mathcal{N}(1, 0.01)$ and $s \sim \mathcal{N}(0.2, 0.0001)$.

We use a uniform Cartesian spatial grid with mesh spacing $\Delta x = 1/127$ in each direction ($n_y = 128^2 = 16384$) and approximate the spatial derivatives using a finite-difference scheme leading to

---

[9] Bengt Fornberg. Generation of finite difference formulas on arbitrarily spaced grids. *Mathematics of Computation*, 51:699–706, 1988

Table 3: Error statistics for each model type in Burgers 2D test case.

| Model type | Error mean ± std. dev. on test set |
|---|---|
| Coarse DNS | $0.560 \pm 0.273$ |
| DMD | $0.097 \pm 0.039$ |
| POD-SINDy | $0.057 \pm 0.045$ |
| Implicit scale | $0.141 \pm 0.040$ |
| Our approach | |
| $n_\eta = 1$ | $0.108 \pm 0.056$ |
| $n_\eta = 2$ | $0.035 \pm 0.022$ |
| $n_\eta = 3$ | $0.037 \pm 0.029$ |
| $n_\eta = 4$ | $0.047 \pm 0.039$ |
| $n_\eta = 5$ | $0.028 \pm 0.019$ |

the system of ODEs

$$
\begin{aligned}
\frac{\mathrm{d}u_{i,j}}{\mathrm{d}t} = & - u_{i,j} \left( \frac{u_{i+1,j} - u_{i-1,j}}{2\Delta x} + \frac{u_{i,j+1} - u_{i,j-1}}{2\Delta x} \right) \\
& + \nu \left( \frac{u_{i+1,j} - 2u_{i,j} + u_{i-1,j}}{\Delta x^2} + \frac{u_{i,j+1} - 2u_{i,j} + u_{i,j-1}}{\Delta x^2} \right),
\end{aligned}
\tag{41}
$$

for $i = 1, \ldots, 128$, $j = 1, \ldots, 128$, where $u_{i,j}$ denotes the solution at the $i,j$-th spatial gridpoint. We generate ground-truth trajectories by integrating the system of ODEs over the time interval $t \in [0, 1]$ with a time step of $\Delta t = 0.001$ ($n_t = 1001$). The observations are corrupted by Gaussian noise with standard deviation 0.001. The training dataset is formed using 20 samples of initial conditions and their corresponding trajectories, and the validation and test sets are formed using 5 samples each. For our multiscale models, the macroscale state is defined on a coarse $8 \times 8$ grid.

As shown in Table 3, our multiscale model with $n_\eta = 5$ achieves the lowest error, surpassing the strongest baseline, POD-SINDy, and reducing the mean prediction error by approximately 51% (from 0.057 to 0.028). The improvement over the implicit-scale model is even more pronounced (an 80% reduction), demonstrating the critical role of the explicit microscale state in capturing sub-grid phenomena. Figure 10 visually confirms this advantage; while even the best baselines struggle to resolve the sharp wavefront, our multiscale model captures it accurately. As shown in Figure 11, this is because the microscale state successfully encodes these fine-scale features, complementing the smooth macroscale state to yield a superior prediction.

### H.4 Cylinder flow in 2D

The data for this test case is a simulation of a 2D Von Kármán vortex street at a Reynolds number of $Re = 160$ generated using the Gerris flow solver [50] by Günther et al. [38]. The dataset provides the nondimensionalized, two-component velocity field. The original data is defined on the spatial domain $[-0.5, 7.5] \times [-0.5, 0.5]$ with a $640 \times 80$ spatial grid. For our study, we truncate the domain to $[-0.5, 3.5] \times [-0.5, 0.5]$ resulting in a $320 \times 80$ spatial grid.

The data consists of a single trajectory with 1501 snapshots over the time interval $t \in [0, 15]$, starting from a zero-velocity initial condition. We partitioned this trajectory temporally for training, validation, and testing, using the intervals $[0, 13]$, $(13, 14]$, and $(14, 15]$, respectively. No noise was added to this dataset. For our multiscale models, the macroscale state is defined on a coarse $32 \times 8$ grid, resulting in a latent state dimension of $n_\zeta = 2 \times 32 \times 8 = 512$ for the two velocity components.

### H.5 Shallow water equations in 2D

This test case uses the radial dam break problem from the PDEBench dataset [39], which simulates the 2D shallow water equations. The dataset provides the scalar fluid elevation field on a $[-2.5, 2.5]^2$

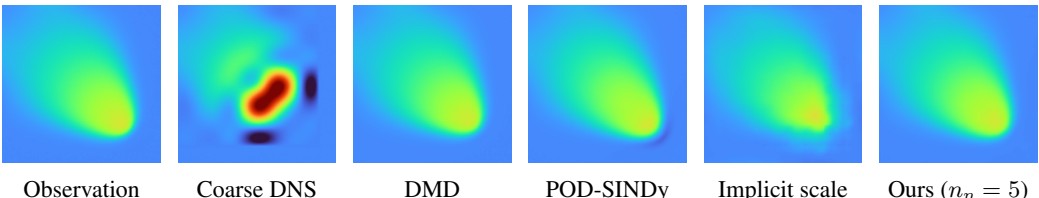

| Observation | Coarse DNS | DMD | POD-SINDy | Implicit scale | Ours ($n_\eta = 5$) |

Figure 10: Burgers' equation in 2D: prediction comparison on test trajectory at $t = 1$.



| Observation | | Approx. full state | | Prolonged macro | | Decoded micro |

Figure 11: Burgers' equation in 2D: scale separation with $n_\eta = 5$, shown at $t = 1$.

spatial domain discretized using a uniform $128 \times 128$ grid. The initial condition is a binary field: $u(x, 0) = 2 \, \forall \|x\| < r$, otherwise $u(x, 0) = 1$, where $r \sim \mathcal{U}[0.3, 0.7]$.

The dataset comprises 1000 trajectories, each generated from a different initial radius $r$. Each trajectory consists of 101 snapshots over the time interval $t \in [0, 1]$. We use 900 trajectories for training, 50 trajectories for validation, and 50 trajectories for testing. We preprocess the data by subtracting 1 from the state everywhere to center it, and corrupt by adding i.i.d. Gaussian noise with a standard deviation of 0.01. For our multiscale models, the macroscale state is defined on a coarse $8 \times 8$ spatial grid ($n_\zeta = 64$).

## H.6 Neural network architectures

This section details the neural network architectures for the smoothing operator, encoder, decoder, and drift functions. All hidden layers in these architectures use a LeakyReLU activation function.

### H.6.1 Encoder architecture

The probabilistic encoder models the conditional distribution of the latent state $z$ given the fully-resolved state $y$ as $p_\theta(z \mid y) = \mathcal{N}(z \mid \mu_\theta^z(y), \Sigma_\theta^z)$. For all test cases, we use a diagonal parametrization of the covariance matrix $\Sigma_\theta^z \in \mathbb{R}^{n_z \times n_z}$. The mean of the encoder is given by

$$\mu_\theta^z(y) = \mathbb{E}[z \mid y] = \begin{bmatrix} \mathcal{E}_\theta^\zeta(\mathcal{S}_\theta(y)) \\ \mathcal{E}_\theta^\eta(S_\theta^\perp(y)) \end{bmatrix},$$

where $\mathcal{E}_\theta^\zeta$ and $\mathcal{E}_\theta^\eta$ are the macroscale and microscale encoders, respectively, $\mathcal{S}_\theta$ is the smoothing operator, and $S_\theta^\perp(y) = y - \mathcal{S}_\theta(y)$ is the residual component containing small-scale features. The architectures for these components are as follows:

1. **Smoothing operator $\mathcal{S}_\theta$:** We parametrize the smoothing operator for each field variable using a single-channel convolutional layer with a stride of 1. Let $s = (n_f/n_c)^{1/d}$ denote the downsampling factor between the fine grid with $n_f$ points and the coarse grid with $n_c$ points, where $d$ is the spatial domain dimension. We set the kernel size to $6s + 1$ for advecting wave and $4s + 1$ for KdV, while for the 2D test cases the kernel size is set to $(4s + 1) \times (4s + 1)$. We use circular padding of length $3s$ for advecting wave, and $2s$ for all other test cases.

2. **Macroscale encoder $\mathcal{E}_\theta^\zeta$:** The macroscale encoder maps the smoothed fully-resolved observation, $\mathcal{S}_\theta(y)$, from the fine grid with $n_f$ points to the coarse grid with $n_c$ points. We implement this as a strided convolution that shares the same learnable kernel as the smoothing operator but uses a stride of $s$, the downsampling factor defined above.

3. **Microscale encoder $\mathcal{E}_\theta^\eta$:** The input to the microscale encoder is the residual $\widetilde{y} = y - \mathcal{S}_\theta(y)$. The architecture of the microscale encoder consists of a series of convolutional layers that

progressively downsample the input (using a stride of 2), followed by a final linear layer to produce the $\eta$ vector. The details of the architecture used for each test case are provided in Table 4. For the 1D test cases, we use a kernel size of 25 and circular padding of length 12. For the 2D test cases, we use a kernel size of $9 \times 9$ and circular padding of length 4.

Table 4: Architecture of the microscale state encoder for each test case.

| Wave 1D | | KdV 1D | | Burgers 2D | | Cylinder 2D | | SWE 2D | |
|---|---|---|---|---|---|---|---|---|---|
| Conv layers | | Conv layers | | Conv layers | | Conv layers | | Conv layers | |
| # | # filters | # | # filters | # | # filters | # | # filters | # | # filters |
| 1 | 1 | 1 | 1 | 1 | 1 | 1 | 2 | 1 | 1 |
| 2 | 4 | 2 | 4 | 2 | 64 | 2 | 64 | 2 | 64 |
| 3 | 16 | 3 | 16 | 3 | 32 | 3 | 32 | 3 | 32 |
| 4 | 64 | 4 | 64 | 4 | 16 | 4 | 16 | 4 | 16 |
| Linear layers | | Linear layers | | Linear layers | | Linear layers | | Linear layers | |
| # | In | Out | # | In | Out | # | In | Out | # | In | Out | # | In | Out |
| 5 | 8000 | $n_\eta$ | 5 | 8000 | $n_\eta$ | 5 | 4096 | $n_\eta$ | 5 | 6400 | $n_\eta$ | 5 | 4096 | $n_\eta$ |

#### H.6.2 Decoder architecture

The decoder maps the latent state $z = (\zeta, \eta)$ to the parameters of the conditional distribution $p_\theta(y \mid z)$. As shown in (32), the conditional mean is a weighted sum of the microscale and macroscale contributions, i.e., $\mathbb{E}[y \mid z] = \mathcal{D}_\theta^\zeta(\zeta) + \Sigma_y S_\theta^\eta \mathcal{D}_\theta^\eta(\eta)$, while the conditional covariance is given by $\mathrm{Cov}[y \mid z] = (S_\theta^\zeta + S_\theta^\eta)^{-1}$. We use a diagonal parametrization for the precision matrices $S_\theta^\zeta$ and $S_\theta^\eta$. The macroscale and microscale decoder architectures are described below:

1. The macroscale decoder, $\mathcal{D}_\theta^\zeta$ is a single convolutional transpose layer with a stride equal to the downsampling factor $s$. Its weights are tied to the macroscale encoder, serving as its approximate inverse to prolong the macroscale state to the fine spatial grid.

2. The architecture of the microscale decoder mirrors that of the encoder. We use a linear layer to project the microscale state to a small spatial feature map, followed by a series of convolutional transpose layers to progressively upsample the microscale features back to the fine grid resolution. These convolutional transpose layers use the same kernel size, stride, and padding settings as their counterparts in the microscale encoder. The specific architecture for each test case is detailed in Table 5.

#### H.6.3 Macroscale drift

As described in Appendix A, the macroscale drift at each grid point is parameterized by a feedforward neural network, $\widehat{f}_\theta$, whose weights are shared across all spatial locations to enforce translation equivariance. This network takes as input the state values from a local stencil, leading to the input dimension $n_{\mathrm{in}} = d_u (2q + 1)^d + n_\eta^*$, where $d_u$ is the number of field variables and $d$ is the spatial dimension. For all cases, we set the stencil half-width $q = 2$ and use an identity coupling function $\phi$ leading to $n_\eta^* = n_\eta$ microscale features used as input. The specific MLP architecture for each test case is given in Table 6.

For the Wave, KdV, Burgers, and SWE test cases, the physical systems are defined on simple domains and are spatially homogeneous. The translation-equivariant MLP is therefore a well-suited architecture. The cylinder flow problem, however, is spatially non-homogeneous due to the presence of the cylinder obstacle. To allow the shared-weights network to learn position-dependent dynamics, we augment its input with four spatial positional encoding features. The input to the elementwise drift $\widehat{f}_\theta$ is augmented with $\{\cos(2\pi x_1/L_1), \sin(2\pi x_1/L_1), \cos(2\pi x_2/L_2), \sin(2\pi x_2/L_2)\}$, where

Table 5: Architecture of the microscale state decoder for each test case.

| | Wave 1D | | | KdV 1D | | | Burgers 2D | | | Cylinder 2D | | | SWE 2D | |
|---|---|---|---|---|---|---|---|---|---|---|---|---|---|---|
| Linear layers | | | Linear layers | | | Linear layers | | | Linear layers | | | Linear layers | | |
| # | In | Out | # | In | Out | # | In | Out | # | In | Out | # | In | Out |
| 1 | $n_\eta$ | 8000 | 1 | $n_\eta$ | 8000 | 1 | $n_\eta$ | 4096 | 1 | $n_\eta$ | 6400 | 1 | $n_\eta$ | 4096 |
| ConvT layers | | | ConvT layers | | | ConvT layers | | | ConvT layers | | | ConvT layers | | |
| # | # filters | | # | # filters | | # | # filters | | # | # filters | | # | # filters | |
| 2 | 64 | | 2 | 64 | | 2 | 16 | | 2 | 16 | | 2 | 16 | |
| 3 | 16 | | 3 | 16 | | 3 | 32 | | 3 | 32 | | 3 | 32 | |
| 4 | 4 | | 4 | 4 | | 4 | 64 | | 4 | 64 | | 4 | 64 | |
| 5 | 1 | | 5 | 1 | | 5 | 1 | | 5 | 2 | | 5 | 1 | |

Table 6: Architecture of the macroscale drift function for each test case.

| | Wave 1D | | | KdV 1D | | | Burgers 2D | | | Cylinder 2D | | | SWE 2D | |
|---|---|---|---|---|---|---|---|---|---|---|---|---|---|---|
| Linear layers | | | Linear layers | | | Linear layers | | | Linear layers | | | Linear layers | | |
| # | In | Out | # | In | Out | # | In | Out | # | In | Out | # | In | Out |
| 1 | $n_{in}$ | 128 | 1 | $n_{in}$ | 128 | 1 | $n_{in}$ | 128 | 1 | $n_{in}+4$ | 128 | 1 | $n_{in}$ | 128 |
| 2 | 128 | 128 | 2 | 128 | 128 | 2 | 128 | 256 | 2 | 128 | 256 | 2 | 128 | 256 |
| 3 | 128 | 128 | 3 | 128 | 128 | 3 | 256 | 128 | 3 | 256 | 128 | 3 | 256 | 128 |
| 4 | 128 | 1 | 4 | 128 | 1 | 4 | 128 | 1 | 4 | 128 | 2 | 4 | 128 | 1 |

$L_1$ and $L_2$ denote the length of the spatial domain along the coordinates $x_1$ and $x_2$, respectively. This results in an input dimension of $n_{in} + 4$ for the cylinder flow case.

Table 7: Architecture of the microscale drift function for each test case.

| | Wave 1D | | | KdV 1D | | | Burgers 2D | | | Cylinder 2D | | | SWE 2D | |
|---|---|---|---|---|---|---|---|---|---|---|---|---|---|---|
| Linear layers | | | Linear layers | | | Linear layers | | | Linear layers | | | Linear layers | | |
| # | In | Out | # | In | Out | # | In | Out | # | In | Out | # | In | Out |
| 1 | $n_{in}$ | 128 | 1 | $n_{in}$ | 128 | 1 | $n_{in}$ | 128 | 1 | $n_{in}$ | 128 | 1 | $n_{in}$ | 128 |
| 2 | 128 | 512 | 2 | 128 | 512 | 2 | 128 | 256 | 2 | 128 | 256 | 2 | 128 | 256 |
| 3 | 512 | 128 | 3 | 512 | 128 | 3 | 256 | 128 | 3 | 256 | 128 | 3 | 256 | 128 |
| 4 | 128 | $n_\eta$ | 4 | 128 | $n_\eta$ | 4 | 128 | $n_\eta$ | 4 | 128 | $n_\eta$ | 4 | 128 | $n_\eta$ |

### H.6.4 Microscale drift

In our numerical studies, the microscale state is not chosen to have a spatial grid structure. We therefore parametrize the microscale drift function using a fully-connected neural network. We use a non-autonomous model for the drift leading to the input dimension $n_{in} = n_\eta + n_\zeta^* + 1$. For the 1D cases, the coupling term $\psi$ is identity ($n_\zeta^* = n_\zeta$). For the 2D cases, $\psi$ is a learned linear map that projects the $n_\zeta$-dimensional macroscale state to $\mathbb{R}^{n_\eta}$ ($n_\zeta^* = n_\eta$). The architecture for each case is shown in Table 7.

## H.7 Training

Each model is trained with the Adam optimizer[10] and the loss function is the negative of the ELBO defined in (8). The batch size is 64. The expectation terms in the ELBO are estimated via a single Monte Carlo sample from the variational distributions. To approximate the temporal integral term in the ELBO, we use 64 quadrature points. The advecting wave models were trained for 50 epochs, KdV for 100 epochs, Burgers for 100 epochs, cylinder flow for 1000 epochs, and shallow water for 20 epochs.

Each multiscale model is trained in a hierarchical manner. We begin by training an implicit-scale model with an initial learning rate of $10^{-3}$. We apply an exponential scheduler that decays the learning rate by 10% every 2000 optimization steps. We then initialize the multiscale model with $n_\eta = 1$ using the trained parameters from the implicit-scale model; we apply Xavier normal initialization to all new weights and zero initialization to all new biases. The multiscale model is then trained with a reduced initial learning rate of $10^{-4}$, as its loss landscape is more sensitive to large step sizes than that of the implicit-scale model. For a multiscale model with $n_\eta = 2$, we repeat this process using the parameters from the trained model with $n_\eta = 1$, and so on for larger $n_\eta$.

Although training a model for large $n_\eta$ directly is possible, we found that the highly non-convex loss landscape can lead to suboptimal local minima. The hierarchical initialization strategy acts as a form of curriculum learning, guiding the optimization through a more stable path. This procedure was crucial for achieving the near-monotonic performance improvements with increasing $n_\eta$ reported in Table 1.

# I   Additional plots for results

This section contains expanded plots of the results presented in Section 5. Many of these figures expand on plots presented in the body of the paper and earlier in the appendix. They are summarized in Table 8.

Table 8: Additional result plots in Appendix I

| Figure in Appendix I | Relevant earlier figure | Description |
|---|---|---|
| | | Scale separation for... |
| 12 | 3 | Wave 1D case |
| 13 | 3 | KdV 1D case |
| 14 | 11 | Burgers 2D case |
| 15,16 | 5 | Cylinder 2D case |
| 17 | 7 | SWE 2D case |
| | | Multiscale predictions for... |
| 18 | N/A | Wave 1D case |
| 19 | N/A | KdV 1D case |
| 20 | 10 | Burgers 2D case |
| 21,22 | 4 | Cylinder 2D case |
| 23 | 6 | SWE 2D case |
| | | Baseline comparisons for... |
| 24 | N/A | Wave 1D case |
| 25 | N/A | KdV 1D case |
| 26 | 10 | Burgers 2D case |
| 27,28 | 4 | Cylinder 2D case |
| 29 | 6 | SWE 2D case |

---

[10]   Diederik P. Kingma and Jimmy Ba. Adam: A method for stochastic optimization. *3rd International Conference on Learning Representations*, 2015

## I.1 Scale separation plots

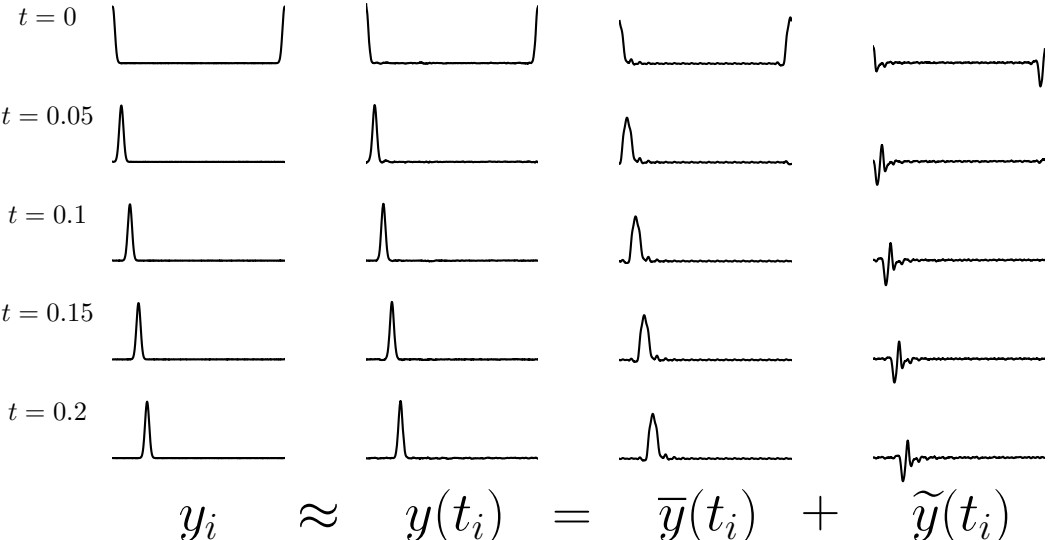

$$y_i \quad \approx \quad y(t_i) \quad = \quad \overline{y}(t_i) \quad + \quad \widetilde{y}(t_i)$$

Figure 12: Advecting wave: visualization of scale separation with $n_\zeta = 20$ and $n_\eta = 5$. The columns are $y_i$: observation from dataset, $y(t_i)$: reconstructed full state, $\overline{y}(t_i)$: prolonged macroscale state, $\widetilde{y}(t_i)$: decoded microscale state.

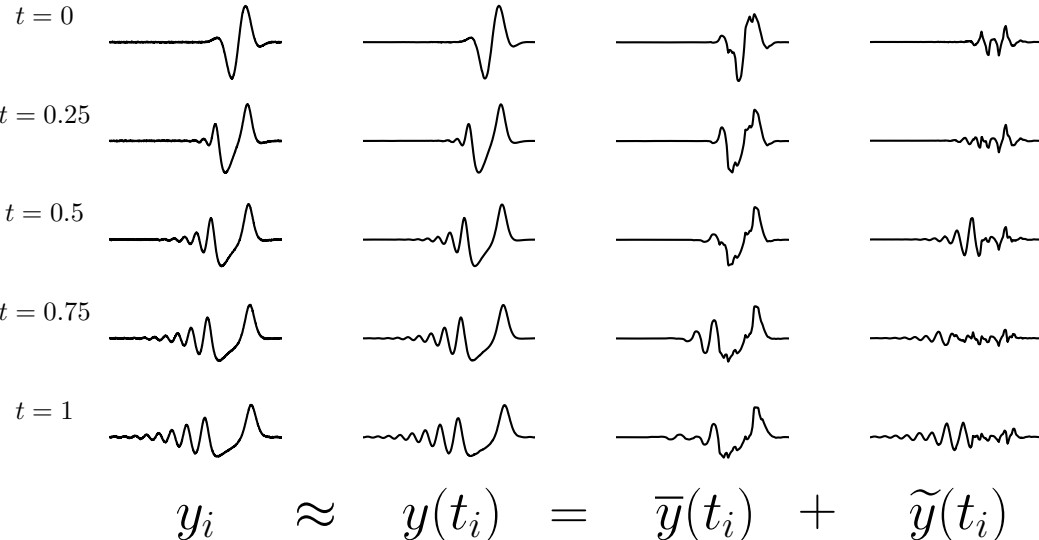

$$y_i \quad \approx \quad y(t_i) \quad = \quad \overline{y}(t_i) \quad + \quad \widetilde{y}(t_i)$$

Figure 13: KdV equation: visualization of scale separation with $n_\zeta = 20$ and $n_\eta = 5$. All plots on common $x$ and $y$-axis scales. The columns are $y_i$: observation from dataset, $y(t_i)$: reconstructed full state, $\overline{y}(t_i)$: prolonged macroscale state, $\widetilde{y}(t_i)$: decoded microscale state.

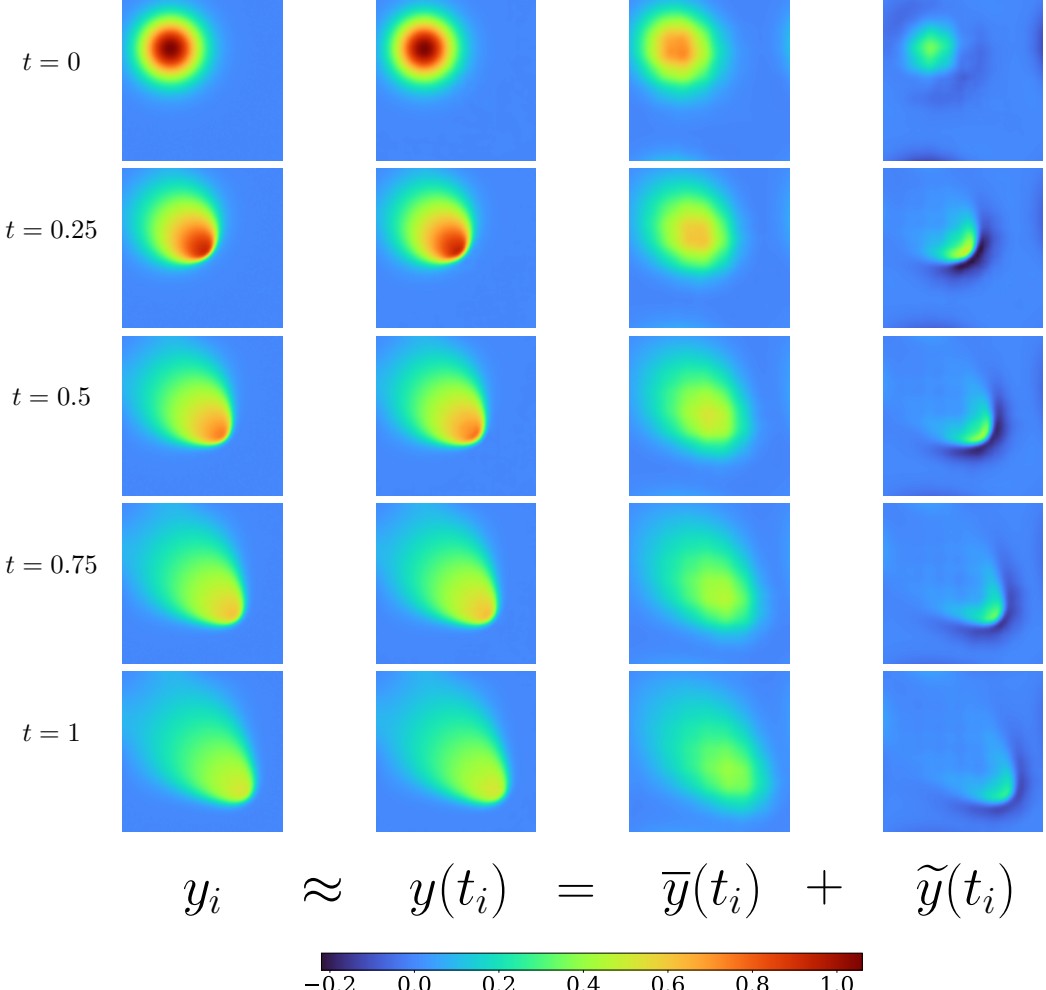

Figure 14: Burgers' equation in 2D: visualization of scale separation with $n_\zeta = 8 \times 8 = 64$ and $n_\eta = 5$. The columns are $y_i$: observation from dataset, $y(t_i)$: reconstructed full state, $\overline{y}(t_i)$: prolonged macroscale state, $\widetilde{y}(t_i)$: decoded microscale state.

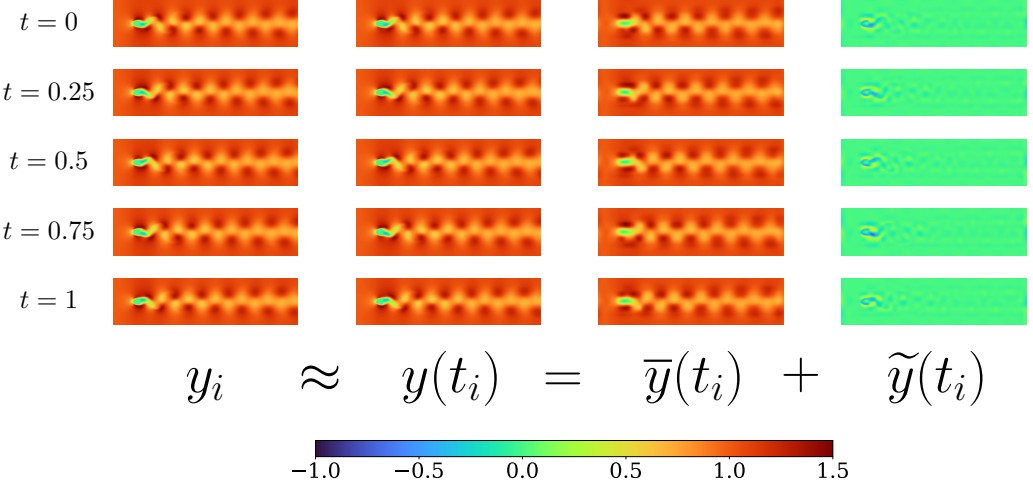

$$y_i \quad \approx \quad y(t_i) \quad = \quad \overline{y}(t_i) \quad + \quad \widetilde{y}(t_i)$$

Figure 15: Cylinder flow in 2D: visualization of scale separation on velocity $x$-component with $n_\zeta = 2 \times 32 \times 8 = 512$ and $n_\eta = 5$. The columns are $y_i$: observation from dataset, $y(t_i)$: reconstructed full state, $\overline{y}(t_i)$: prolonged macroscale state, $\widetilde{y}(t_i)$: decoded microscale state.

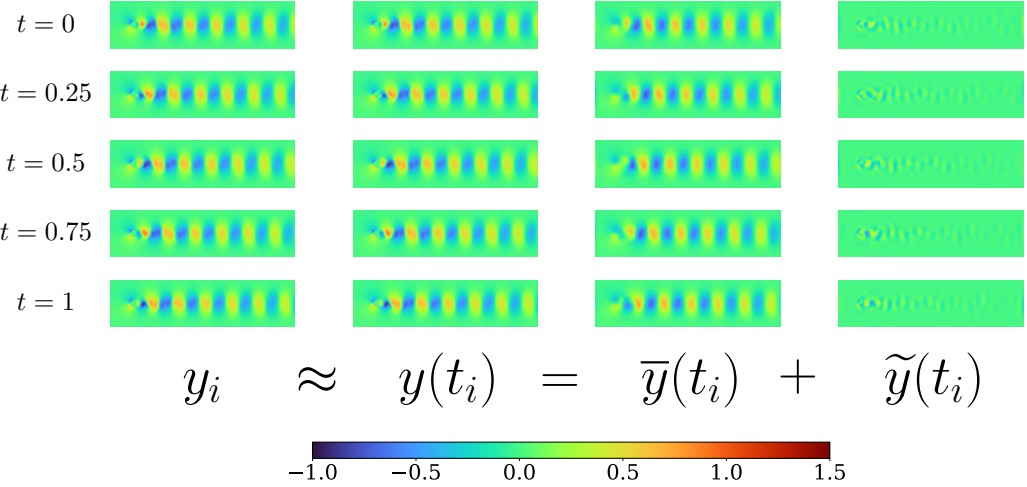

$$y_i \quad \approx \quad y(t_i) \quad = \quad \overline{y}(t_i) \quad + \quad \widetilde{y}(t_i)$$

Figure 16: Cylinder flow in 2D: visualization of scale separation on velocity $y$-component with $n_\zeta = 2 \times 32 \times 8 = 512$ and $n_\eta = 5$. The columns are $y_i$: observation from dataset, $y(t_i)$: reconstructed full state, $\overline{y}(t_i)$: prolonged macroscale state, $\widetilde{y}(t_i)$: decoded microscale state.

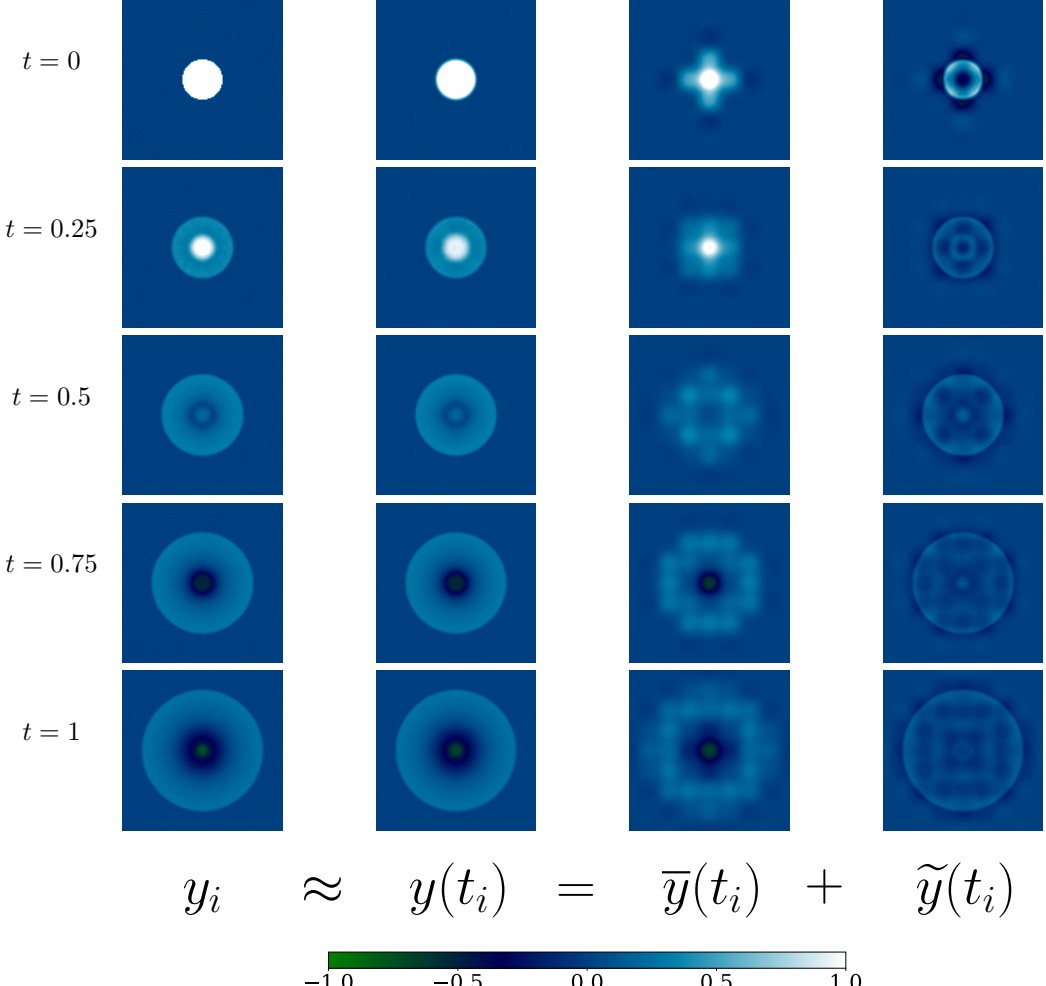

Figure 17: Shallow water equations in 2D: visualization of scale separation with $n_\zeta = 8 \times 8 = 64$ and $n_\eta = 5$. The columns are $y_i$: observation from dataset, $y(t_i)$: reconstructed full state, $\overline{y}(t_i)$: prolonged macroscale state, $\widetilde{y}(t_i)$: decoded microscale state.

## I.2 Multiscale model prediction plots

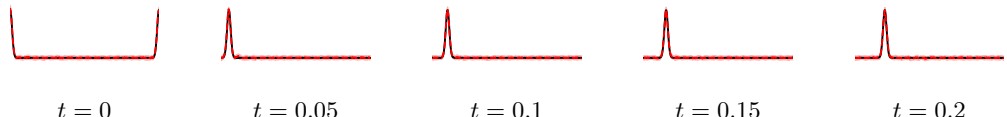

| $t = 0$ | $t = 0.05$ | $t = 0.1$ | $t = 0.15$ | $t = 0.2$ |

Figure 18: Advecting wave: multiscale model prediction on trajectory from test set with $n_\zeta = 20$ and $n_\eta = 5$. Observations are represented with black solid curves and model predictions with red dashed curves.

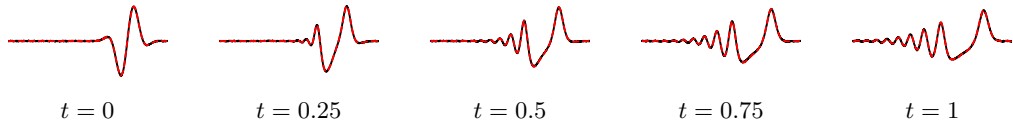

| $t = 0$ | $t = 0.25$ | $t = 0.5$ | $t = 0.75$ | $t = 1$ |

Figure 19: KdV equation: multiscale model prediction on trajectory from test set with $n_\zeta = 20$ and $n_\eta = 5$. Observations are represented with black solid curves and model predictions with red dashed curves.

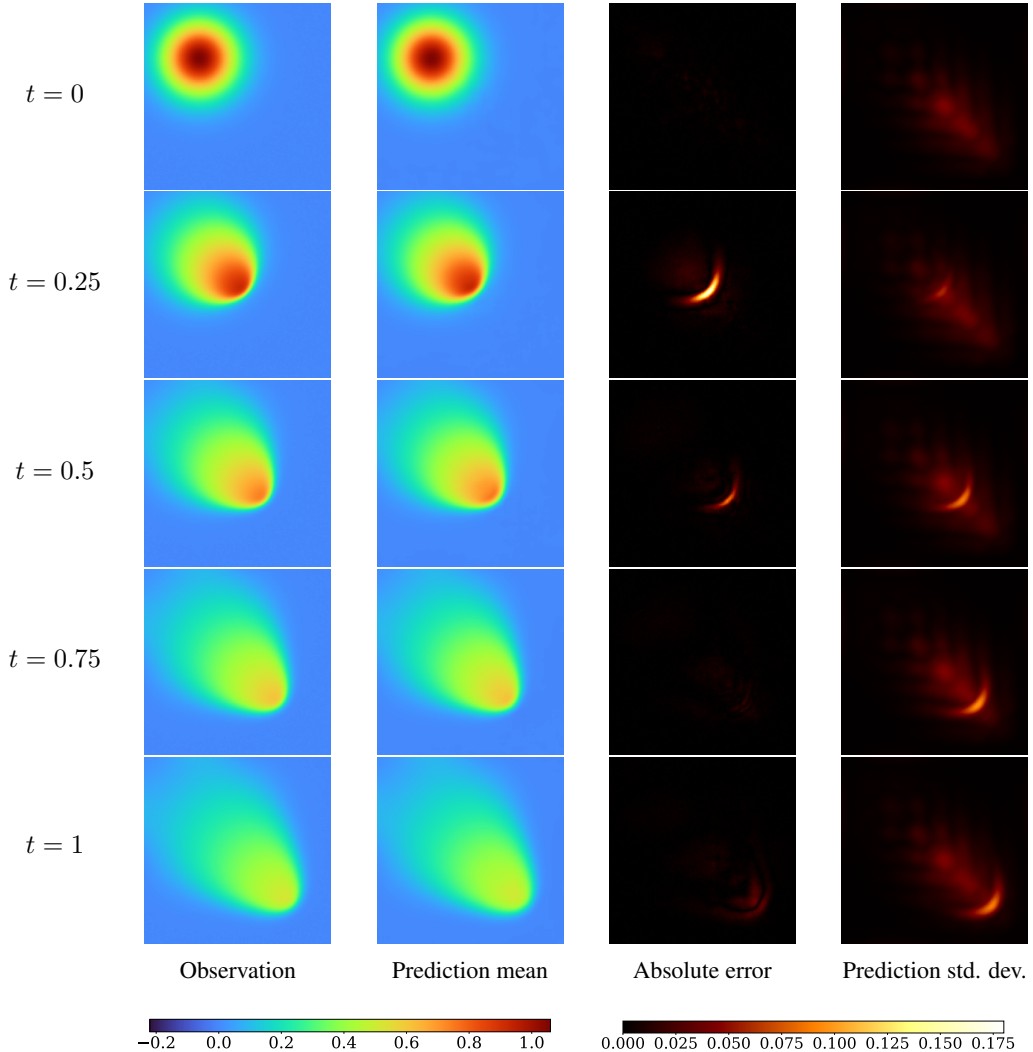

Figure 20: Burgers' equation in 2D: multiscale model prediction on trajectory from test set with $n_\zeta = 8 \times 8 = 64$ and $n_\eta = 5$.

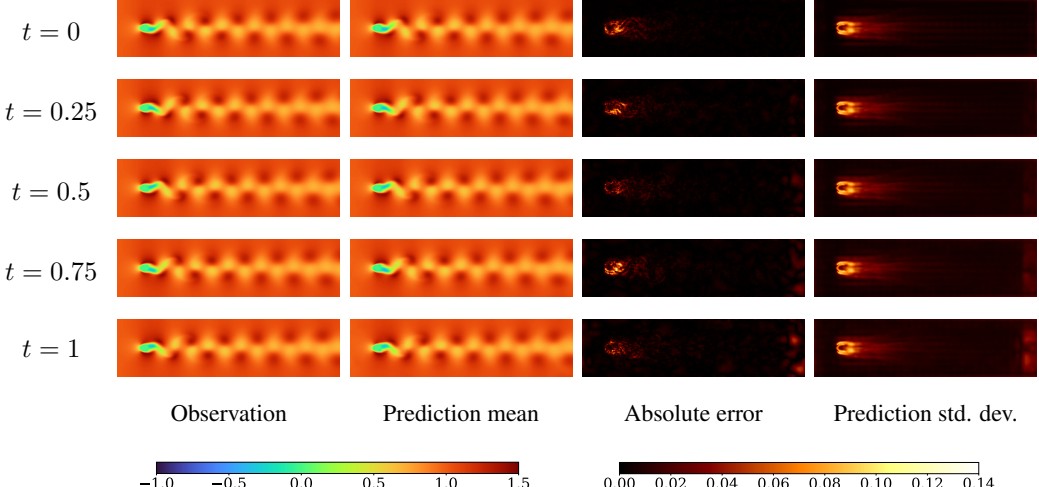

Figure 21: Cylinder flow in 2D: multiscale model prediction of $x$-component of velocity on test interval with $n_\zeta = 2 \times 32 \times 8 = 512$ and $n_\eta = 5$.

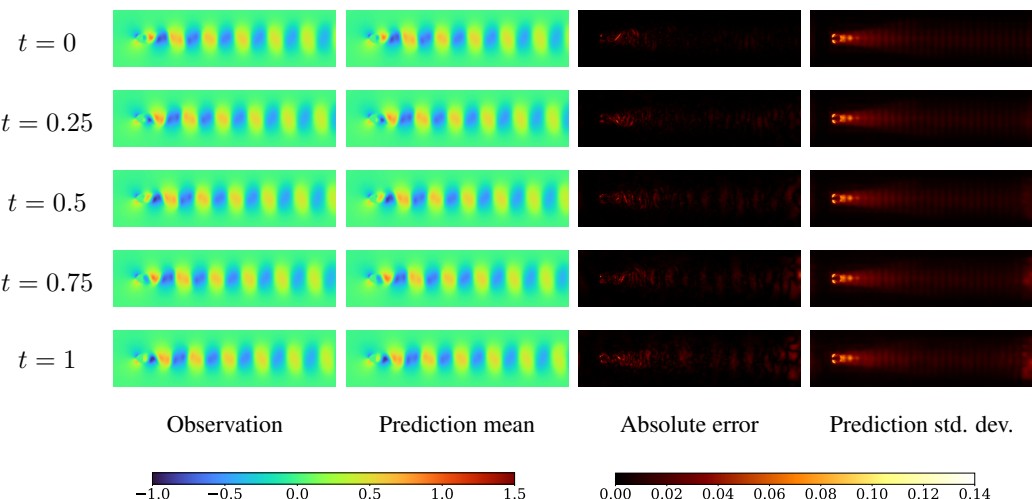

Figure 22: Cylinder flow in 2D: multiscale model prediction on test interval with $n_\zeta = 2 \times 32 \times 8 = 512$ and $n_\eta = 5$. Velocity $y$-component is shown.

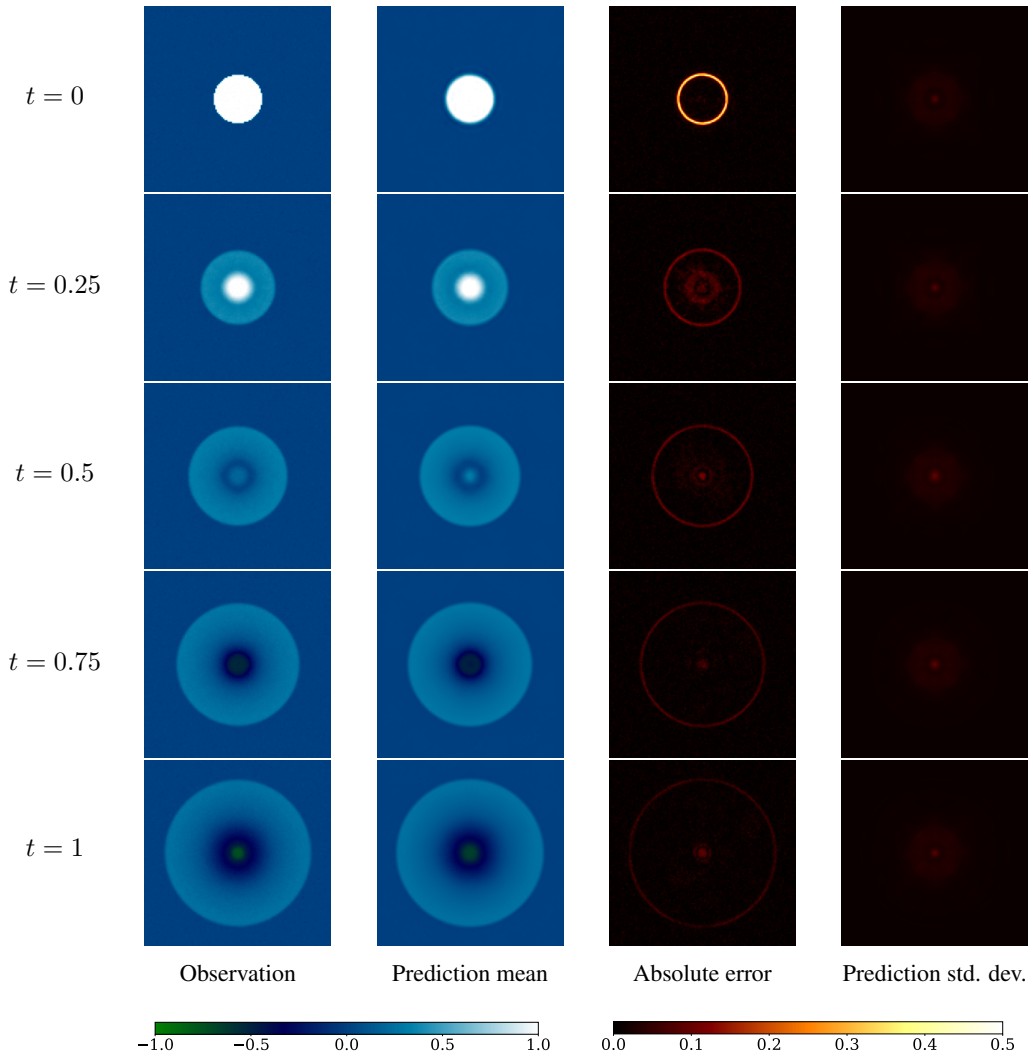

Figure 23: Shallow water equations in 2D: multiscale model prediction on trajectory from test set with $n_\zeta = 8 \times 8 = 64$ and $n_\eta = 5$.

## I.3 Comparison of predictions

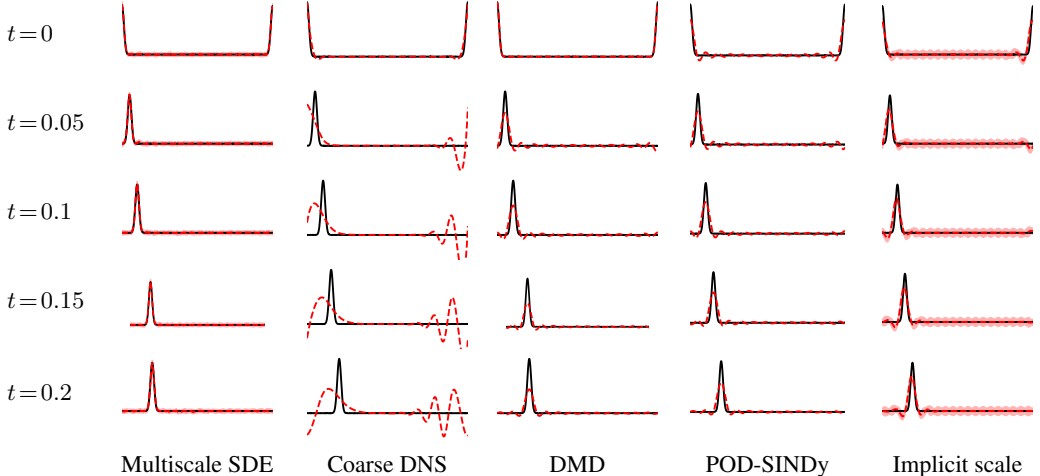

Figure 24: Advecting wave: Comparison of predictions on trajectory from test set. Observations are represented with black solid curves and model predictions with red dashed curves. For the multiscale model, the predictions corresponding to $n_\zeta = 20$ and $n_\eta = 5$ are shown.

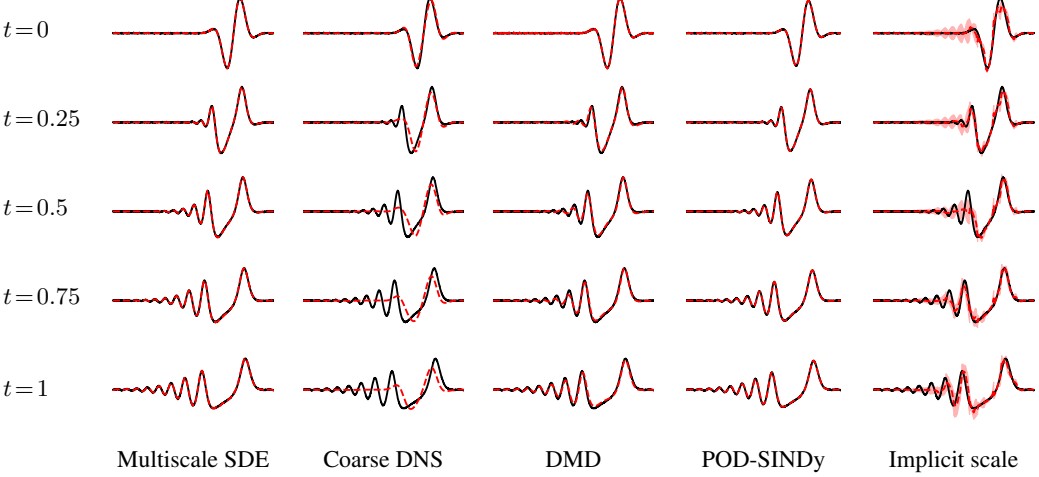

Figure 25: KdV equation: comparison of predictions on trajectory from test set. Observations are represented with black solid curves and model predictions with red dashed curves. For the multiscale model, the predictions corresponding to $n_\zeta = 20$ and $n_\eta = 5$ are shown.

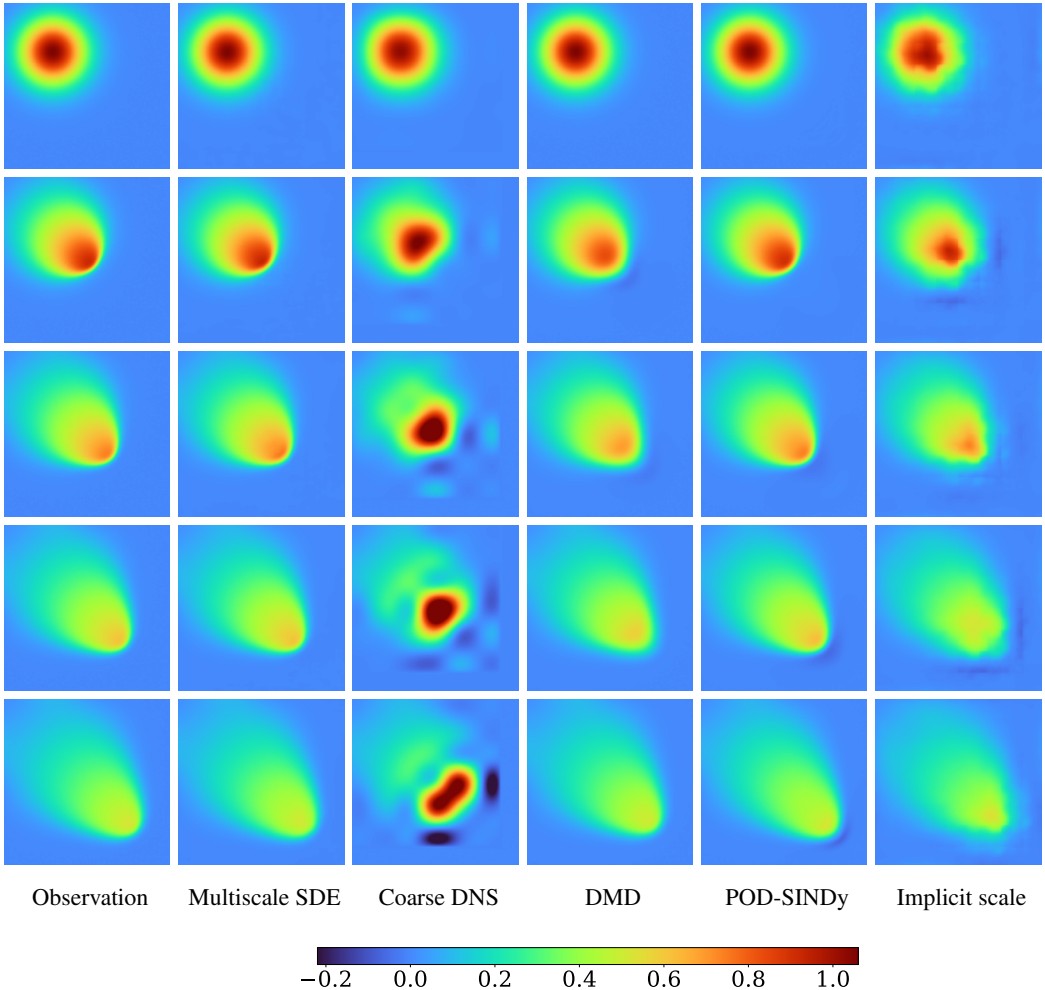

Observation  Multiscale SDE  Coarse DNS  DMD  POD-SINDy  Implicit scale

Figure 26: Burgers' equation in 2D: Comparison of predictions on trajectory from test set. The rows correspond to time instances $t = 0, 0.25, 0.5, 0.75, 1$ from top to bottom. For the multiscale model, the mean predictions corresponding to $n_\zeta = 64$ and $n_\eta = 5$ are shown.

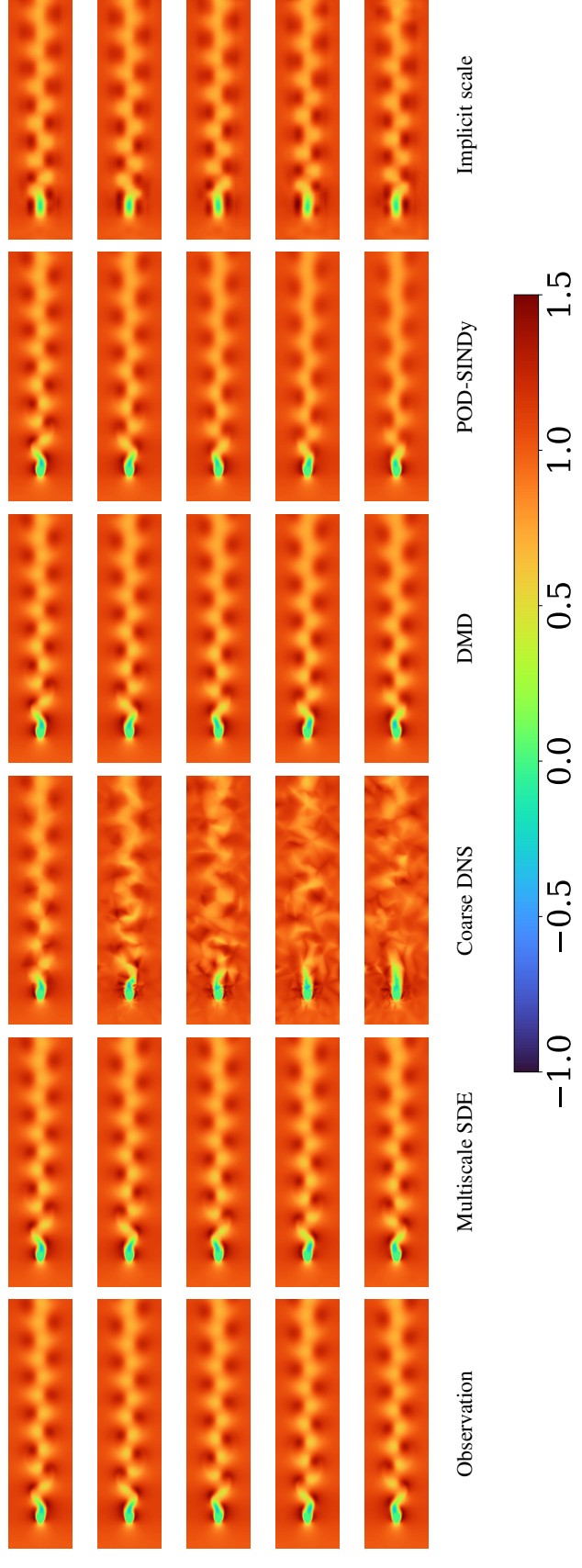

Figure 27: Cylinder flow in 2D: comparison of predictions of $x$-component of velocity on trajectory from test set. The rows correspond to time instances $t = 0, 0.25, 0.5, 0.75, 1$ from top to bottom. For the multiscale model, the mean predictions corresponding to $n_\zeta = 512$ and $n_\eta = 5$ are shown.

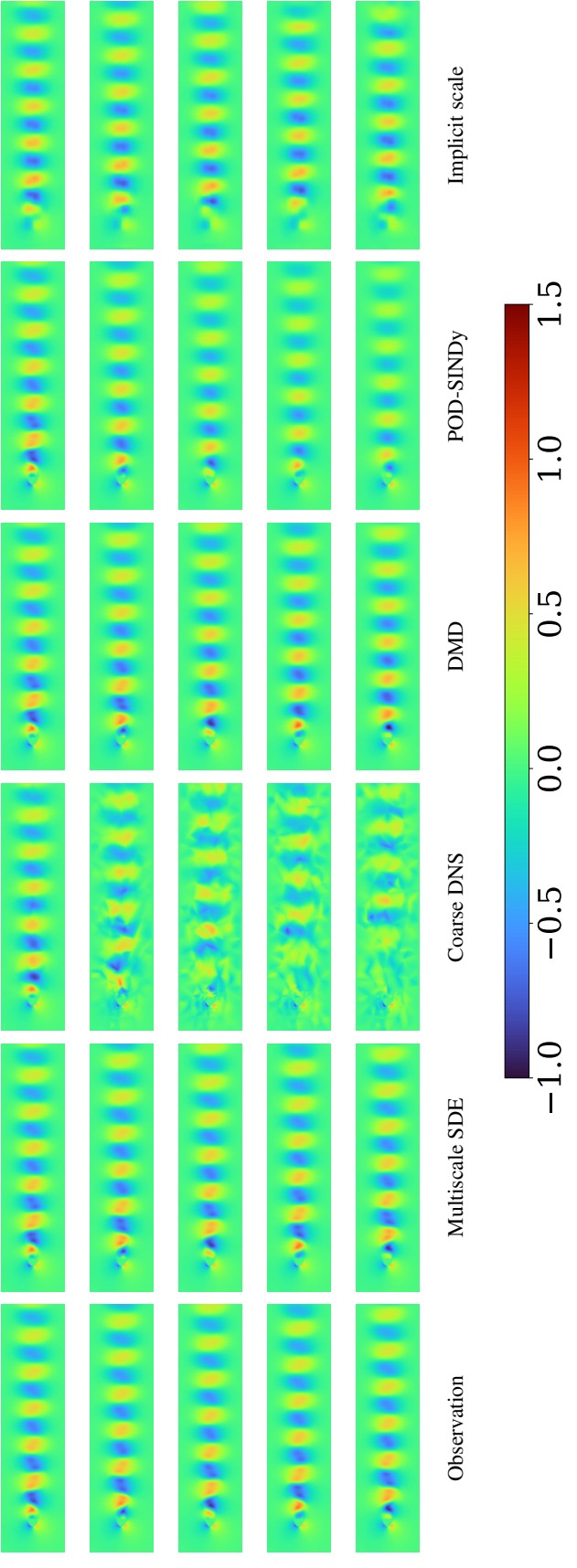

Figure 28: Cylinder flow in 2D: comparison of predictions of $y$-component of velocity on trajectory from test set. The rows correspond to time instances $t = 0, 0.25, 0.5, 0.75, 1$ from top to bottom. For the multiscale model, the mean predictions corresponding to $n_\zeta = 512$ and $n_\eta = 5$ are shown.

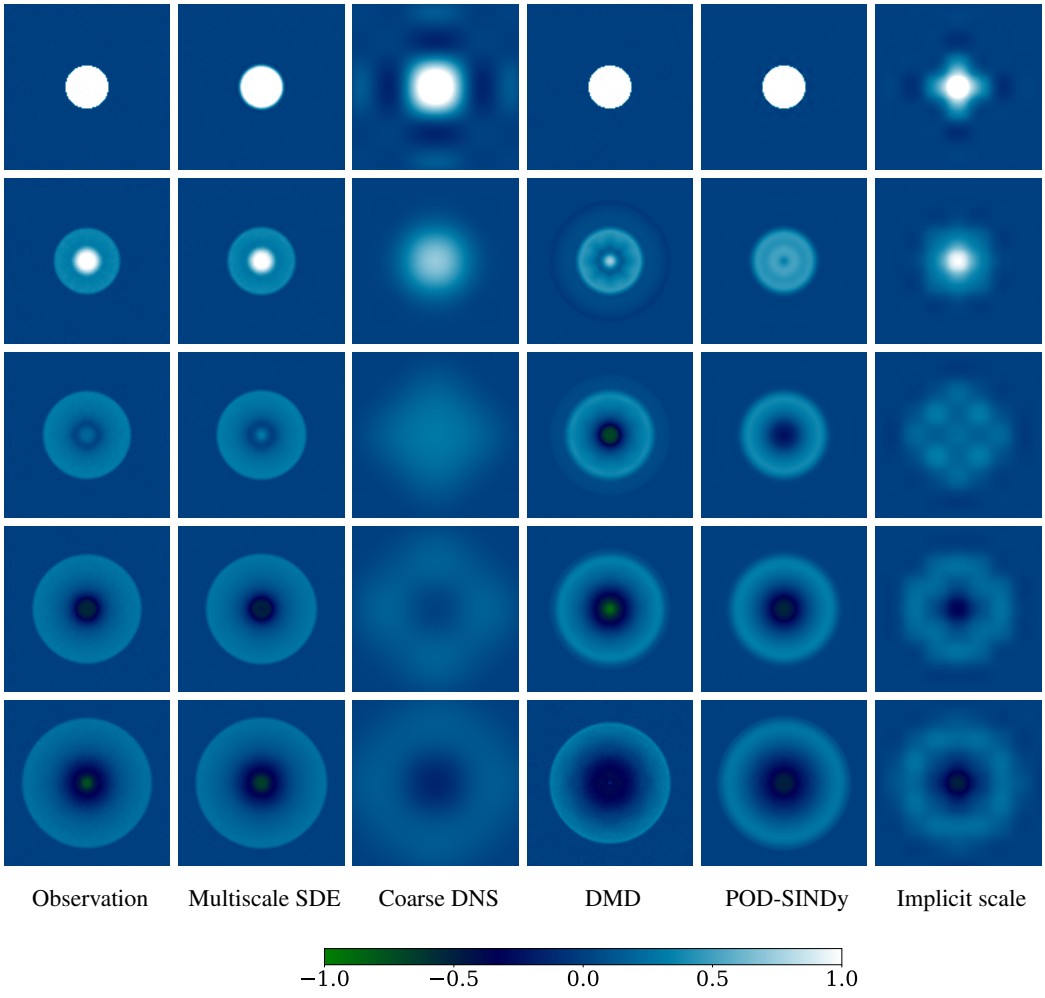

Figure 29: Shallow water equations in 2D: Comparison of predictions on trajectory from test set. The rows correspond to time instances $t = 0, 0.25, 0.5, 0.75, 1$ from top to bottom. For the multiscale model, the mean predictions corresponding to $n_\zeta = 64$ and $n_\eta = 5$ are shown.

# J   Learning smoothing kernels

In this work, we choose to use a convolution to obtain the macroscale state from the full state. With an appropriate kernel size and stride, the convolution smooths the full state and sparsely samples it on the coarse macroscale mesh. Prolonging the macroscale state onto the full mesh is accomplished by transposing the convolution used to obtain it. Because this operation is lossy, the prolonged macroscale state loses sub-grid-scale features of the full state.

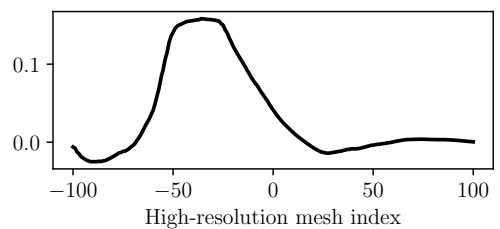

Figure 30: The macroscale smoothing kernel learned for the multiscale model ($n_\zeta = 20$, $n_\eta = 5$) trained on the KdV dataset. Note that adjacent gridpoints on the macroscale mesh are 50 gridpoints apart on the high-resolution mesh.

If the kernel of the convolution was simply chosen to be Gaussian, then this operation would amount to a low-pass filter. We choose however to leave the kernel parametrized and to infer it from data. The optimal kernel for a particular dataset may turn out to be non-Gaussian, and indeed this is what we observed in our numerical studies. The kernel for the multiscale model ($n_\zeta = 20$, $n_\eta = 5$) trained on the KdV dataset is shown in Figure 30. Although the training process did not converge on a Gaussian kernel, it did converge on a smooth curve.

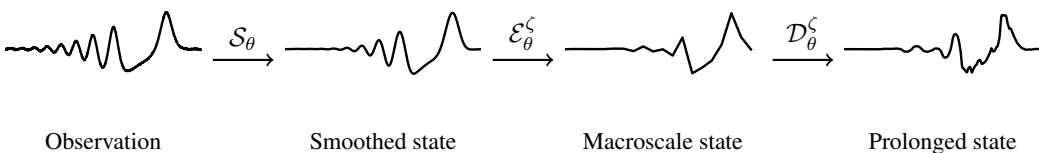

Figure 31: The macroscale smoothing, restriction, and prolongation operators visualized on a snapshot from the KdV dataset.

Figure 31 illustrates the macroscale smoothing, restriction, and prolongation. As our example state, we use the snapshot at $t = 1$ from a test trajectory of the KdV dataset. We see how the smoothed state and especially the macroscale state on its coarsened mesh lose the small-scale features present in the original snapshot. The learned kernel however reintroduces some small-scale features in the prolongation, as determined by the training process to maximize the likelihood defined in Section 4.1.

Figure 32 shows a spectral plot of the example snapshot from Figure 31 compared to the prolonged macroscale state and a multiscale reconstruction. As expected, the decoded macroscale state accurately follows the true spectrum above the spatial Nyquist frequency of the macroscale mesh. Below this frequency, however, the small-scale features introduced by the deconvolution approximately follow the true spectrum, although they cannot be directly resolved by the macroscale mesh. This sug-

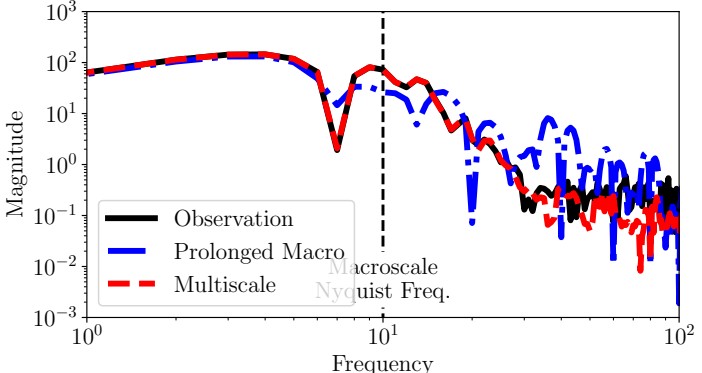

Figure 32: Spectral plot of observation, multiscale reconstruction ($n_\zeta = 20$, $n_\eta = 5$), and prolonged macroscale state using a snapshot from the KdV dataset.

gests that the training process selects a non-Gaussian kernel not just to smooth, but also to encode information about the sub-grid scales that can be partially reconstructed during decoding.

