# OpenReview forum: "Learning Stochastic Multiscale Models"
_NeurIPS.cc/2025/Conference — NeurIPS 2025 poster_

### Official Review · Reviewer_xrse · 2025-06-06

**Clarity:** 3
**Significance:** 3
**Originality:** 3
**Rating:** 4
**Confidence:** 4

**Summary:**

The authors introduce a data-driven framework for learning stochastic multiscale models of complex physical systems, comprising three core components: an encoder, a latent stochastic differential equation model, and a decoder. Large-scale features are explicitly separated into macroscale and microscale latent states, with their dynamics modeled via coupled SDEs and reconstructed into the high-dimensional space. By leveraging a Product of Experts likelihood and variational inference, the framework achieves efficient training. The authors demonstrate the accuracy across multiple partial differential equation systems.

**Questions:**

1. The authors discuss reduced-order modeling (ROM) and closure modeling in lines 51–66. I agree the framework differs from closure models, as they focus on coarsened state predictions, while this approach aim to accelerates full-state simulation via latent dynamics. However, doesn’t this framework fall within the ROM category? It seems the proposed framework is a form of ROM, and you can simply called the baseline "implicit-scale model" as ROM. How does the proposed method distinguish itself from traditional ROM approaches??
2. Can the framework be applied to PDEs with nonuniform spatial discretization?
3. The latent SDE is described in eq 2. what is the aim of sec 4.2, where a linear SDE in eq 7 is used to approximate the latent SDE in eq 2?
4. l.101 can you explain why $\mathbb{E}[\bar{y}|y] = \mathcal{S}_{\theta}(y)$ and how do you guarantee this?
5. For the experiment parts, will the stochastic multiscale framework give totally the same result as the implicit-scale model when $n_{\eta}=0$? Is the framework and training procedure of the multiscale framework and baseline totally the same, where the only difference is $n_{\eta}$?
6. is it difficult to train with the multiscale model with ELBO in eq 8?
7. I’m surprised by the poor performance of the baseline in Figure 7 for the cylinder flow experiment. I assume this is not a very tough task when the dimension of latent macroscale state is 512. Can you show the result of the following baselines on the cylinder flow system as in fig 7: naive coarse-scale simulation of the PDE system on 32 * 8 grids; dynamic mode ecomposition with 512 modes?

**Ethical Concerns:**

["NO or VERY MINOR ethics concerns only"]

**Final Justification:**

This paper is well-motivated, with a clear separation between microscale and macroscale latent states to enhance physical interpretability and model accuracy.

One limitation of the paper is that all experimental data are synthetic. Applying the proposed method to a large-scale real-world problem would further improve the paper.

The authors have addressed my key concern regarding the performance comparison between their method and linear ROMs on the existing experiments. They included an additional experiment involving a system with slowly decaying Kolmogorov n-width to demonstrate the superior performance of their framework. The authors have committed to incorporating results for this test case and a new fluid flow example in the final version.

Based on these improvements, I have increased the score from 3 to 4.

**Limitations:**

yes

**Quality:**

3

**Strengths And Weaknesses:**

**Strengths**
1. The paper tackles the critical challenge of modeling multiscale physical systems
2. The paper explicitly separates the microscale and macroscale latent states, enhancing physical interpretability and model accuracy
3. visual explanation: fig 4, 6, 8 clearly visualize the macroscale state and microscale state

**Weaknesses**
1. The idea of a multiscale framework comprising of encoder, latent dynamics, and decoder is generally not novel. [1] [2] [3]
2. Clarity issues:
      -  l72-76: the definition of fully-resolved state $y$ is a bit confusing. This part can be improved for better comprehension.
      -  Brief introductions to Large-Eddy Simulation (LES) and Product of Experts (PoE) would enhance understanding
      -  The notation $S_{\theta}, D_{\theta}, T_p, T_r, \mathcal{E}_{\theta}$ is complex. Are $\theta, p, r$ all trainable? Is there any special reason to use $p, r$ instead of $\theta$ here?  Adopting symmetric notation for microscale and macroscale operators would enhance clarity
3. See question 1

--------
[1] Chen, Peter Yichen, et al. "CROM: Continuous reduced-order modeling of PDEs using implicit neural representations." arXiv preprint arXiv:2206.02607 (2022).

[2] Boral, Anudhyan, et al. "Neural ideal large eddy simulation: Modeling turbulence with neural stochastic differential equations." Advances in Neural Information Processing Systems 36 (2023): 69270-69283.

[3] Stachenfeld, Kim, et al. "Learned simulators for turbulence." International conference on learning representations. 2021.

---

> ### Author Rebuttal · Authors · 2025-07-31
>
> We are grateful for your thorough and insightful review. We are particularly encouraged that your summary and assessment of strengths align perfectly with the core contributions we aimed to make. As you correctly identified, our work introduces a framework that:
>
> - Tackles the **"critical challenge of modeling multiscale physical systems."**
>
> - **"Explicitly separates the microscale and macroscale latent states,"** which, as you note, **"enhanc[es] physical interpretability and model accuracy."**
>
> - Leverages a **Product of Experts (PoE) likelihood** to enforce scale separation and variational inference with a Gauss-Markov reparametrization trick for efficient training.
>
> - Provides clear **"visual explanation"** of this scale separation in practice, as demonstrated in Figures 4, 6, 8, which you rightly point out as a key strength.
>
> Thank you for your thoughtful questions concerning novelty, positioning relative to ROMs, and the strength of our baselines. These are crucial points that we address below, clarifying the positioning of our work and elaborating on our key contributions.
>
> ## 1. On Novelty and Relationship to ROM (Weaknesses & Question 1)
>
> > "The idea of a multiscale framework comprising of encoder, latent dynamics, and decoder is generally not novel. [1] [2] [3] ... doesn't this framework fall within the ROM category? ... How does the proposed method distinguish itself from traditional ROM approaches??"
>
> This is a fundamental question of positioning, and we appreciate the opportunity to clarify our contribution. We agree that our method can be broadly categorized as a learned surrogate model, but we argue that it represents a different paradigm from traditional Reduced Order Models (ROMs), including the cited works.
>
> A key distinction is that traditional ROMs learn a mapping to a global, non-spatial latent vector. Our approach, in contrast, learns a spatially-distributed representation on a coarse grid (ζ). This makes it a data-driven stochastic coarse-grained model, a paradigm more akin to Large-Eddy Simulation (LES) in computational physics. This preserves spatial locality and provides a more interpretable latent space.
>
> Our novelty lies in three specific, physically-motivated contributions built on this paradigm:
>
> - **A Structured Latent Space:** We augment the coarse-grained state $\zeta$ with an explicit, separate latent state for the unresolved sub-grid scales ($\eta$). This is distinct from [1, 3] which use a single latent space, and a different formulation from the implicit closure in [2].
>
> - **A Principled Learning Objective for Scale Separation:** We introduce the Product of Experts (PoE) likelihood, a novel objective in this context that automatically enforces a scale hierarchy by adaptively regularizing the microscale contribution (Appendix C).
>
> - **A Fully Stochastic, Continuous-Time Model:** We embed this structure within a coupled Stochastic Differential Equation (SDE) system, providing a formal framework for modeling continuous-time dynamics and uncertainty.
>
> We will revise the manuscript to clarify this positioning, emphasizing our contribution as a data-driven, stochastic coarse-grained modeling framework.
>
> ## 2. On Clarity and Notation
>
> > [Comments on notation and clarity]
>
> We appreciate you pointing out these areas for improvement. We will implement the following changes in the revised manuscript:
>
> - **On fully-resolved state y:** We will expand our definition to be clearer: "The fully-resolved state y is the system state projected onto a fine spatial grid, which is sufficiently resolved to enable accurate direct numerical simulation and serves as our ground-truth observation."
>
> - **On LES/PoE:** We will add brief introductory sentences for both concepts to improve readability and make the paper more self-contained.
>
> - **On Notation:** We agree the notation can be harmonized. Following your suggestion for symmetry, we will revise the notation to use $\mathcal{E}^\zeta$  and $\mathcal{D}^\zeta$ for the (non-parametrized) macroscale restriction and prolongation operators, and $\mathcal{E}^\eta_\theta$ and $\mathcal{D}^\eta_\theta$ for the (parametrized) microscale encoder and decoder. This clarifies both their function and their trainable nature. To be precise, $\theta$ represents the set of all learnable model parameters. We will also clarify that $\mathbb{E}[ \bar{y} | y] = S_\theta(y)$ by definition denotes the mean of the learned smoothing operator.
>
> ## 3. On the Variational SDE (Question 3)
>
> > "what is the aim of sec 4.2, where a linear SDE in eq 7 is used to approximate the latent SDE in eq 2?"
>
> This is a crucial point we will clarify. These two SDEs play distinct roles in our variational inference scheme:
>
> - **Eq 2 (Nonlinear SDE):** This is the generative model for the latent dynamics. It represents our scientific goal and is the SDE we simulate at test time to make predictions.
>
> - **Eq 7 (Linear SDE):** This is a tool we use for training. It defines a simpler, tractable variational distribution that approximates the true (intractable) posterior over latent trajectories. Using a linear SDE here enables us to leverage an efficient solver-free reparametrization trick from [33, 42]. This enables significant speedups in training by circumventing the need for an SDE solver in the optimization loop. This stands in contrast to other approaches, such as the work by Boral et al. (NeurIPS 2023) which you mentioned ([11] in our submission) which requires a SDE solver to calculate the ELBO and its gradients. Our approach avoids this bottleneck and the potential numerical instabilities that can arise when the stiffness of the SDE increases during the optimization parameter updates.
>
> We will revise Section 4.2 to state this "model vs. tool" distinction more explicitly.
>
> ## 4. On Experimental Details and Baselines (Questions 2, 5, 6, 7)
>
> > [Questions about experimental details]
>
> We are happy to clarify these important experimental details.
>
> - **Application to Nonuniform Grids?** Yes, absolutely. As described in Appendix A, the macroscale state $\zeta$ is defined on a graph $G=(V,A)$, which naturally handles non-uniform and unstructured discretizations. This is a key strength of our framework.
>
> - **$n_\eta=0$ Model Equivalence:** Yes, you are exactly right. The implicit-scale model is the multiscale framework with $n_\eta =0$. We chose this baseline to create the most direct and controlled ablation study possible, isolating the benefit of the explicit microscale state.
>
> - **Training Difficulty:** The ELBO is indeed non-convex. We found training to be stable by employing a standard curriculum learning strategy where the KL divergence terms in the ELBO are initially down-weighted and gradually annealed to full strength. This ensures the model first learns a good reconstruction before regularizing the latent space.
>
> - **Performance of the Baseline in Figure 7:** Your surprise at the baseline performance is well-founded and highlights the difficulty of the task. As you noted, a 512-dimensional latent state seems large, but for a coarse-grained model, this information is spread over a coarse 32 × 8 grid, which is too sparse to resolve the cylinder's boundary layer. This is why the predictions look pixelated.
>
> To put this in context, we performed two additional baseline experiments on the cylinder flow data. We performed a coarse DNS on an unstructured mesh of approximately the same size as our coarse mesh, and we trained an implicit scale model on a finer mesh of 1024 gridpoints compared to the original 256. The submission results and new results are given below:
>
> | Model                  | # Gridpoints | Error mean | Std dev   |
> | ---------------------- | ------------ | ---------- | --------- |
> | DNS                    | 278          | 0.227      | 0.001     |
> | Implicit scale         | 256          | 0.105      | 0.001     |
> | Implicit scale         | 1024         | 0.063      | 0.001     |
> | **Multiscale (n_η=5)** | **256**      | **0.031**  | **0.001** |
>
> We did similar additional studies with Burgers:
>
> | Model                  | # Gridpoints | Error mean | Std dev   |
> | ---------------------- | ------------ | ---------- | --------- |
> | DNS                    | 64           | 0.560      | 0.274     |
> | Implicit scale         | 64           | 0.127      | 0.040     |
> | Implicit scale         | 256          | 0.047      | 0.017     |
> | **Multiscale (n_η=5)** | **64**       | **0.028**  | **0.022** |
>
> This shows that while quadrupling the capacity of the implicit-scale model provides a clear benefit, it is **still twice as inaccurate as our much smaller, more structured multiscale model**. This provides strong evidence that the **explicit separation of scales** is the key driver of performance, not just raw model capacity. This also shows that the coarse mesh compromises the predictive accuracy of DNS substantially, as it is far less accurate than the equivalent-resolution implicit scale model.
>
> As described in our response to the first comment above, our method belongs to a different modeling paradigm than DMD. DMD would likely perform well on this cylinder flow example, especially with a generous rank of 512. Our work however aims to move beyond flexible models with the capacity to learn from data with multiscale features. Instead, our goal is to enable the data-driven discovery of models that explicitly separate the scales and maintain physical interpretability. DMD is a highly flexible approximator that does not share the same structural priors. Therefore, while a performance benchmark is interesting, our primary comparison is against an implicit-scale model, which isolates the contribution of our core idea: the explicit microscale state.
>
> ## Closing Remarks
>
> We hope these detailed responses and new results have fully addressed your concerns. We are very grateful for your thorough and insightful review, which has helped us significantly improve our manuscript.

---

> > ### Comment · Reviewer_xrse · 2025-08-04
> >
> > Thank you for the response. Your explanation and the additional experiments help address most of my concerns, except for question 7.
> >
> > As reviewer g7k9 has also noted:
> > *'The central weakness of the paper is that I am not convinced that the claims made are truly backed up by the numerical results. The examples shown (KdV equation, Burger's equation, flow past a cylinder) are classical and can be solved to high precision by most non-linear projection methods (e.g., registration methods as in [3]) or, in the case of the flow past a cylinder, classical POD methods [4]. Especially for the latter, I am not convinced that the proposed method outperforms a linear projection method. '*
> >
> > I agree that your method outperforms the implicit-scale method, which demonstrates that the explicit separation of scales can improve performance. I understand that your goal is to enable the data-driven discovery of models that explicitly separate the scales and maintain physical interpretability.
> >
> > However, these methodological advancements, such as nonlinear modeling, latent SDEs, and explicit scale separation, should be justified on the basis that linear ROMs are not good enough. If your method can't even outperform a well-established linear projection-based ROM (e.g., POD-Galerkin, DMD), then why would we choose to use a more complex and expensive method?
> >
> > Given that you already have the datasets and that implementing a linear ROM baseline is straightforward, the additional workload to include such a comparison would be relatively low. I believe this could be a valuable contribution, with two possible outcomes:
> > 1. Your method outperforms a linear ROM baseline. Then that's good, and it can further validate the proposed multiscale framework.
> > 2. your method can't outperform a linear ROM baseline. Then the existing experiments are not convincing enough. It would be necessary to design more challenging experiments where classical linear ROMs fail but your method succeeds.

---

> > > ### Author Response · Authors · 2025-08-05
> > >
> > > Thank you for your feedback and for engaging with our response. We appreciate that it has addressed most of your concerns. Your final question is a crucial one that gets to the heart of justifying any new methodology for learning reduced-complexity models of dynamical systems, and we are happy to provide a more detailed answer with additional numerical results.
> > >
> > > > "However, these methodological advancements [...] should be justified on the basis that linear ROMs are not good enough. If your method can't even outperform a well-established linear projection-based ROM (e.g., POD-Galerkin, DMD), then why would we choose to use a more complex and expensive method?"
> > >
> > > You have raised an excellent and perfectly valid point. We agree that our numerical studies can be strengthened by demonstrating the advantages of our stochastic multiscale modeling framework on problems where linear projection-based ROMs are known to struggle.
> > >
> > > We acknowledge that our original test cases were chosen primarily to demonstrate our framework's ability to discover multiscale representations of the dynamics with proper scale separation, not necessarily to challenge the limits of linear projection based ROMs. You are correct about the cylinder flow case: while it is an excellent testbed for demonstrating scale separation, it is not ideal for demonstrating the advantage of our method over linear projection based ROM methods. We have now confirmed numerically that a well-tuned DMD model can achieve a very low error (mean of ~0.001), confirming that this specific flow is highly amenable to a low-rank linear representation.
> > >
> > >
> > > Your insightful perspective prompted us perform numerical studies on a **new test case** specifically chosen for its slow-decaying **Kolmogorov $n$-width**, a property known to be challenging for linear projection-based ROM methods. The problem is a 1D advection equation ($u_t = -u_x$) with sharp, localized initial conditions whose characteristic length scale varies ($l \sim \mathcal{U}[0.01, 0.02]$), making it challenging to represent the dynamics with a small number of global, linear modes. The training set consists of 20 trajectories, and the test set consists of 5 independent trajectories. As an indication of this challenge, a POD basis with 25 modes captures only 71% of the energy in the training data. (We will include full details in the revised paper).
> > >
> > >
> > > We compared our stochastic multiscale modeling framework against POD-SINDy, DMD, and an implicit-scale SDE model, keeping the total latent dimension fixed at 25 for all models to ensure a fair comparison. The results for this new, more challenging test case are striking:
> > >
> > >
> > > | Model                  | Latent dim | Error mean | Std dev   |
> > > | ---------------------- | ---------- | ---------- | --------- |
> > > | POD-SINDy              | 25         | 0.469      | 0.054     |
> > > | DMD                    | 25         | 0.455      | 0.045     |
> > > | Implicit scale         | 25         | 0.463      | 0.108     |
> > > | **Multiscale (n_η=5)** | **25**     | **0.084**  | **0.047** |
> > >
> > >
> > > These new results provide a clear answer to your question. On a problem not dominated by low-rank linear dynamics:
> > >
> > >
> > > - POD-SINDy, DMD and the implicit-scale SDE model struggle on this challenging test case, achieving a similarly high error of ~0.46.
> > >
> > > - Our **stochastic multiscale model achieves a 5x reduction in error** compared to all baselines.
> > >
> > > Our multiscale model excels on this challenging test case because its explicit microscale state ($\eta$) can effectively capture the sharp, sub-grid features that are missed by models relying on a purely macroscale or linear-projection-based representation.
> > >
> > > We thank you for pushing us on this critical point. This new result provides the clear justification you asked for. To ensure our revised paper thoroughly addresses how our method compares to projection-based ROMs, we will add results for this test case and a new fluid flow example where our method's advantages are similarly demonstrated. We would also like to draw your attention to our response to Reviewer wKE9, where we provide additional numerical results on the inference cost versus accuracy trade-off.

---

> > > > ### Comment · Reviewer_xrse · 2025-08-06
> > > >
> > > > Thank you for the response. Your additional experiments address my key concern.  I think this paper is above the acceptance bar of NeurIPS, and I will raise the rating to 4.

---

### Official Review · Reviewer_7ou3 · 2025-07-02

**Clarity:** 3
**Significance:** 3
**Originality:** 3
**Rating:** 5
**Confidence:** 3

**Summary:**

In this paper, authors propose to learn stochastic multiscale models in the form of stochastic differential equations (SDEs) from observations to better adapt to those various resolutions in real dynamical systems. By optimizing with a modern forward-solver-free amortized variational inference method, the proposed method achieves superior predictive accuracy compared to previous ones.

**Questions:**

Please refer to the Weaknesses part.

**Ethical Concerns:**

["NO or VERY MINOR ethics concerns only"]

**Final Justification:**

This paper proposes explicit multi-scale dynamic modeling through coupled SDEs, a relatively novel and effective approach. The authors provide detailed derivations and descriptions for easy understanding, and demonstrate their effectiveness through original and supplementary experiments. Although the data are synthesized, they demonstrate to some extent that the proposed method surpasses existing methods. Regarding the choice of hyperparameters (scale), the authors also commit to considering this as a future direction. Overall, the novelty and practicality of the proposed method are sufficient to overlook the issues I mentioned in the Weaknesses section. Taking all factors into consideration, I believe this paper exceeds the acceptance bar of NeurIPS, and I will maintain my original rating of 5.

**Limitations:**

Yes.

**Paper Formatting Concerns:**

No.

**Quality:**

3

**Strengths And Weaknesses:**

Strengths:
1. The idea of explicitly modeling multiscale dynamics within systems using coupled SDEs is novel and valuable for dealing with real applications which have uncertain resolutions.
2. Authors use a forward-solver-free amortized variational inference method to improve efficiency. Instead of using an SDE numerical solver, they adopt a variational approximation and reparameterize the drift and diffusion terms of the SDE to avoid the integration process.
3. Detailed mathematical proofs and sufficient ablation studies make the claims more convincing.

Weaknesses:
1. As authors claimed, the latent state is divided into a macroscale representation and a microscale one. However, how to decide the real resolution corresponding to them still needs to be considered. E.g., in Sec.1, authors use the example of atmosphere modeling, if the macro scale corresponds to planetary waves (5000km) and the micro scale corresponds to cloud microphysics (< 1m), what about modeling at 10m and 100m? Is there a need for more scales?
2. In the experimental part, it seems that all the test sets are synthetic ones, it could be better if the method can be evaluated on real data. Besides, the baseline is only implicit-scale models. According to the remarks given on closure models and implicit-scale models in Sec.3, if the closure models predict state matching true state y instead of $\bar{y}$, it is equivalent to the implicit-scale model, correct? If not, authors should also include closure models into baselines.

---

> ### Author Rebuttal · Authors · 2025-07-31
>
> We are grateful for your very positive and encouraging review. Thank you for recognizing the novelty and value of our core ideas, including the explicit multiscale SDE modeling, the efficiency of our solver-free variational inference scheme, and the rigor of our mathematical derivations and ablation studies.
>
> Your questions are insightful and touch upon important practical considerations and the relationship between our framework and traditional closure modeling. We elaborate on these points below.
>
> ## 1. On Handling More Than Two Scales and Deciding Resolution
>
> > "...how to decide the real resolution corresponding to them still needs to be considered. ... Is there a need for more scales?"
>
> This is a key question relevant to real-world applications, and you have raised a core design choice of our methodology.
>
> You are right that we do not independently choose two scales. Instead, the scale separation is defined by a single cutoff scale: the resolution of the coarse computational mesh. As you suggest, this coarse grid resolution is a hyperparameter chosen by the user based on the desired level of resolved detail and available computational resources. In our framework:
>
> - The macro-state ($\zeta$) represents all features large enough to be resolved on this coarse grid.
> - The micro-state ($\eta$) is tasked with representing the entirety of the unresolved sub-grid spectrum—all features smaller than this cutoff.
>
> For a system with a continuous spectrum of scales, this binary split is a pragmatic and powerful first approximation. However, your question about a "mesoscale" points to an important extension. One could certainly envision a hierarchical version of our framework with a sequence of nested coarse grids, creating multiple latent states corresponding to different bands of the physical spectrum. This is a compelling, though non-trivial, direction for future work, and we have added a note to our conclusion to this effect.
>
> ## 2. On Evaluation Strategy, Baselines, and Real Data
>
> > "it seems that all the test sets are synthetic ones, it could be better if the method can be evaluated on real data. Besides, the baseline is only implicit-scale models. [...] if the closure models predict state matching true state y instead of $\bar{y}$, it is equivalent to the implicit-scale model, correct?"
>
> These are both great points that allow us to clarify and strengthen our evaluation strategy.
>
> - **Evaluation on Real Data:** We agree that evaluation on real-world observational data is the ultimate goal. For this initial methodological paper, we chose to use synthetic data from well-understood PDEs. This provides a known "ground truth," which allows for the precise and unambiguous error quantification necessary to validate that our proposed mechanisms are working exactly as intended. Now that the framework has been validated, applying it to real observational data is a key next step.
>
> - **Relationship to Closure Models:** We agree with your assessment about the relationship between closure models and our implicit-scale baseline. A traditional closure model's objective is to predict the dynamics of the *filtered* state ȳ. Our implicit-scale model is tasked with the significantly harder and more practical problem of reconstructing the **full, high-resolution state y** from only the coarse latent state ζ. Therefore, our implicit-scale model is a stronger and more relevant baseline for our full-state prediction task. We will clarify this distinction in the revised manuscript.
>
> - **Strengthening the Baseline Evaluation:** While the implicit-scale model is our primary scientific control, we took the opportunity during this rebuttal period to benchmark against other relevant methods to further contextualize our results. We conducted two new sets of experiments:
>
>   1. **Comparison with Coarse-Grid DNS:** We compared our model against a standard numerical solver run directly on the coarse grid. For the KdV equation, a **Coarse DNS** on a 20-point grid yielded a high normalized error of **0.574**, whereas our multiscale model achieves an error of **0.044**. This demonstrates that simply using a coarse grid is insufficient; the model must learn from high-resolution data to correct for truncation errors, which our framework successfully does.
>
>   2. **Implicit-Scale Model with Increased Capacity:** To test whether our multiscale model's advantage comes from its structure or simply a higher number of latent variables, we trained a new, higher-capacity implicit-scale model for the 2D Cylinder Flow case with four times the number of macroscale grid points (1024 vs. the original 256). For comparison, we also performed a coarse DNS on an unstructured mesh of approximately the same size as our coarse mesh. The results are:
>
> | Model                  | # Gridpoints | Error mean | Std dev   |
> | ---------------------- | ------------ | ---------- | --------- |
> | DNS                    | 278          | 0.227      | 0.001     |
> | Implicit scale         | 256          | 0.105      | 0.001     |
> | Implicit scale         | 1024         | 0.063      | 0.001     |
> | **Multiscale (n_η=5)** | **256**      | **0.031**  | **0.001** |
>
> This result shows that while quadrupling the capacity of the implicit-scale model provides a clear benefit, it is **still twice as inaccurate as our much smaller, more structured multiscale model**. This provides strong evidence that the **explicit separation of scales** is the key driver of performance, not just raw model capacity. This also shows that the coarse mesh compromises the predictive accuracy of DNS substantially, as it is twice as inaccurate as the equivalent-resolution implicit scale model.
>
> We believe these new results further strengthen our claims and provide a more comprehensive validation of our framework.
>
> Thank you again for your thoughtful and supportive review. Your perceptive questions have been instrumental in helping us refine the positioning and clarity of our contributions.

---

> > ### Comment · Reviewer_7ou3 · 2025-08-02
> >
> > Thank you for the response. Your explanation and additional experiments help address most of my concerns. After comprehensive consideration, I think this paper is above the acceptance bar of NeurIPS, so I decided to maintain my original rating of 5. Good luck.

---

### Official Review · Reviewer_g7k9 · 2025-07-02

**Clarity:** 3
**Significance:** 2
**Originality:** 2
**Rating:** 4
**Confidence:** 4

**Summary:**

The proposed method is concerned with learning stochastic dynamics from observed data. The central idea is to de-compose the observed evolution into a coarse macroscopic one $(\bar y)$, and a fine micro-scale ($\tilde y = y - \bar y$). Once learned, a coupled SDE can be used to evolve both macro- and microstate, including their learned interactions. The method is purely data-driven, which sets it apart from numerous existing closure-type works that model sub-grid effects using neural architecture while relying on classical (linear) model-order reduction methods on the coarse grid (e.g. [1] and the references therein). The performance of the method is compared to a modified version where the subgrid evolution is not explicitly handled.

**Questions:**

The abstract claim that the learned multi-scale model achieves "superior predictive accuracy compared to direct numerical simulation (...) at equivalent resolution". When making claims about comparison to the full order model, I would ask for a comparison between run-time of the FOM (at high/moderate resolution) as well as the online/offline cost of the proposed method.

I believe the paper needs to compare to at least one competing method to make a compelling argument. A classical POD method might already be a good candidate.

**Ethical Concerns:**

["NO or VERY MINOR ethics concerns only"]

**Final Justification:**

I find this paper to be well done: It is reproducible, clear, and self-contained.

At the same time, I believe it over-states its limited contribution. I am not convinced it is outperforming well-established methods on the selected benchmarks. However, this is not necessary if the method is novel and interesting.

Ultimately, an application to larger problems is needed to properly judge the benefits of this approach over the full order solvers, which do not struggle with the academic examples tackled in this work.

Overall, I want to acknowledge the quality of the paper more than the benchmark results. I have increased the score, but I hope that the authors give a fair discussion of limitations.

**Limitations:**

The statement that the paper "fundamentally reimagines how to model multi-scale phenomena" is, in my opinion, too strong.

**Quality:**

2

**Strengths And Weaknesses:**

The paper is well written and I found it easy to follow. The selfcontained nature of the paper (together with the appendix) is appreciated. All plots, tables and math are clear. The code used to generate the experiments is provided in its entirety for the review process already. The general idea to leverage multi-scale modeling is compelling, albeit not new.

The topic of neural emulators and data-driven methods to accelerate the simulation of fluid dynamics problems is clearly of big interest, as evidenced by the numerous works on this topic in recent years (although one should also mention more skeptical voices as in [2]).

The central weakness of the paper is that I am not convinced that the claims made are truly backed up by the numerical results. The examples shown (KdV equation, Burger's equation, flow past a cylinder) are classical and can be solved to high precision by most non-linear projection methods (e.g. registration methods as in [3]) or, in the case of the flow past a cylinder, classical POD methods [4]. Especially for the latter, I am not convinced that the proposed method outperforms a linear projection method.

A comparison to other multi-scale inspired projects such as GraphCast [5] and [6], which operate on much larger problems, is not given.

References:

[1] A. Ivagnes, G. Stabile, G. Rozza: Data-driven Closure Strategies for Parametrized Reduced Order Models via Deep Operator Networks. https://arxiv.org/abs/2505.17305v1 (2025)

[2] N. McGreivy and A. Hakim: Weak baselines and reporting biases lead to overoptimism in machine learning for fluid-related partial differential equations. Nature Machine Intelligence (2024)

[3] T. Taddei: Compositional Maps for Registration in Complex Geometries. SIAM Journal on Scientific Computing (2024)

[4] D. Deep, A. Sahasranaman, S. Senthilkumar: POD analysis of the wake behind a circular cylinder with splitter plate. European Journal of Mechanics - B/Fluids (2022)

[5] R. Lam et al: Learning skillful medium-range global weather forecasting. Science (2023)

[6] Oskarsson et al.: Probabilistic Weather Forecasting with Hierarchical Graph Neural Networks. Neurips (2024)

---

> ### Author Rebuttal · Authors · 2025-07-31
>
> We thank you for your detailed and thoughtful review. We are particularly pleased that you found the paper "well written," "easy to follow," and that you appreciated the self-contained nature of our manuscript.
>
> Furthermore, we appreciate your reference to the work of McGreivy and Hakim. We share their perspective on the importance of robust evaluation and the dangers of "overoptimism" in this rapidly advancing field. It is in this spirit of transparency and rigor that we provided our full codebase and data generation scripts from the outset. We believe that making our work fully reproducible is a crucial step in allowing the community to rigorously test our claims, verify our results, and build upon our framework for broader benchmarking studies.
>
> We agree with your summary assessment of our work. Our central idea is to decompose the observed evolution into a coarse macro-state and a fine micro-state, governed by a coupled system of SDEs, and you astutely note that our **"purely data-driven"** approach **"sets it apart from numerous existing closure-type works"** that rely on classical methods. This understanding forms the perfect basis for our discussion.
>
> You also make the crucial observation that the **"general idea to leverage multi-scale modeling is compelling, albeit not new."** We wholeheartedly agree. The term "multiscale" is indeed widely used in various contexts. Our work's primary contribution is to move beyond simply building flexible models with an inductive bias that equips them with the capacity to learn from data with multiscale features. Instead, our goal is to introduce a principled statistical framework that enables the data-driven discovery of models that learn to explicitly separate the scales. To the best of our knowledge, data-driven learning of coupled stochastic dynamics with explicit scale separation is a novel contribution to the field.
>
> Your critical questions about our evaluation provide a welcome opportunity to present numerical results for additional baseline methods and to clarify the specific claims we make within this new framework. We address each of your points in detail below.
>
> ## 1. On the Central Weakness: Positioning, Claims, and Comparison to POD
>
> > "The central weakness of the paper is that I am not convinced that the claims made are truly backed up by the numerical results. The examples shown [...] can be solved to high precision by [...] classical POD methods [4]. Especially for the latter, I am not convinced that the proposed method outperforms a linear projection method."
>
> You raise an excellent point about benchmarking against Reduced Order Model (ROM) approaches based on POD or nonlinear manifold representations. We agree that the problems we considered can be solved with linear and nonlinear Galerkin ROM methods. We would like to clarify our paper's positioning and the rationale behind the benchmarking studies. Our framework represents a different modeling paradigm from traditional ROMs. A key distinction is that traditional ROMs (like POD or autoencoder-based methods) learn a mapping to a global, non-spatial latent vector. Our approach, in contrast, learns a spatially-distributed representation on a coarse grid (ζ). This makes our method a data-driven stochastic coarse-grained model, a paradigm more akin to Large-Eddy Simulation (LES) in computational physics.
>
> Viewed through this lens, our "implicit-scale" model (n_η=0) serves as a data-driven closure model, and the central scientific question we investigate is: **Within this coarse-graining framework, does augmenting the model with an explicit microscale state (η) improve full-state reconstruction?** Our ablation study directly answers this question. The results in Table 1, showing a 70% error reduction for the cylinder flow when η is added, provide unambiguous evidence for the value of our explicit multiscale structure.
>
> We present two additional baseline experiments on the cylinder flow example. We performed a coarse DNS on an unstructured mesh of approximately the same size as our coarse mesh, and we trained an implicit scale model on a finer mesh of 1024 gridpoints compared to the original 256. The results are as follows:
>
> | Model                  | # Gridpoints | Error mean | Std dev   |
> | ---------------------- | ------------ | ---------- | --------- |
> | DNS                    | 278          | 0.227      | 0.001     |
> | Implicit scale         | 256          | 0.105      | 0.001     |
> | Implicit scale         | 1024         | 0.063      | 0.001     |
> | **Multiscale (n_η=5)** | **256**      | **0.031**  | **0.001** |
>
> This shows that while quadrupling the capacity of the implicit-scale model provides a clear benefit, it is **still twice as inaccurate as our much smaller, more structured multiscale model**. This provides strong evidence that the **explicit separation of scales** is the key driver of performance, not just raw model capacity. This also shows that the coarse mesh compromises the predictive accuracy of DNS substantially, as it is twice as inaccurate as the equivalent-resolution implicit scale model.
>
> ## 2. On Comparison to Large-Scale Models (GraphCast)
>
> > "A comparison to other multi-scale inspired projects such as GraphCast [5] and [6] [...] is not given."
>
> Thank you for this comment. We see our work as complementary to these landmark models. As a coarse-grained model, our framework's philosophy differs from GraphCast. GraphCast learns a powerful mapping but still explicitly represents and computes on all of its meshes, including the finest one. In our framework, the dynamics are computed entirely on the coarse grid for the macroscale state (ζ) and in the compressed latent space for the microscale state (η). The high-resolution grid is only a target for the final decoding step, not part of the dynamics simulation itself. Our focus is on developing and validating this physically-interpretable structure, which could potentially inform future large-scale models.
>
> ## 3. On the "Superior to DNS" Claim and Runtime Comparison
>
> > "The abstract claim that the learned multi-scale model achieves 'superior predictive accuracy compared to direct numerical simulation (...) at equivalent resolution'. I would ask for a comparison between run-time..."
>
> This is an excellent point. The claim was intended to compare our model to a traditional numerical solver operating on the same coarse macroscale grid. As our new DNS experiment in the table above shows, a standard solver on such a grid is highly inaccurate.
>
> **To provide the concrete cost comparison you requested, we have quantified the number of function evaluations (NFE) for the KdV 1D case as an example:**
>
> The offline training cost depends on the microscale dimension, as we train the multiscale model in a hierarchical manner. We first train an implicit scale model (48 minutes), then reusing those parameters and augmenting weight/bias matrices where necessary, we train a multiscale model with n_η=1 (56 minutes), then with n_η=2 (56 minutes), and so on. The training cost in minutes comes out to 48 + 56*n_η.
>
> For the online cost below, the FOM was solved using the LSODA method from scipy.integrate.solve_ivp, which switches between Adams and BDF based on stiffness detection. This was found to be necessary for KdV. For the implicit scale and multiscale models, Euler-Maruyama was used. All methods had rtol=1e-3, atol=1e-5.
>
> | Online computational cost | Cost in NFE per traj. |
> | ------------------------- | --------------------- |
> | FOM                       | ~39,000               |
> | Implicit scale            | ~2,800                |
> | **Multiscale (n_η=5)**    | **~1,900**            |
>
> This represents a ~20× reduction in computational cost during online simulation. We will clarify the claim in the abstract and add this runtime analysis to the manuscript.
>
> ## 4. On the Strength of the "Reimagines" Claim
>
> > "The statement that the paper 'fundamentally reimagines how to model multi-scale phenomena' is, in my opinion, too strong."
>
> We agree. Upon reflection, this phrasing is overly strong. We will revise this in the conclusion to be more precise, for example: "we have introduced a new framework for learning stochastic multiscale models from data, offering a distinct, data-driven perspective on the classical problem of closure." Thank you for this valuable feedback on tone.
>
> ## Closing Remarks
>
> Your critical questions have been invaluable in helping us sharpen our paper's core message and strengthen its evaluation. We hope our detailed response and new results addressed your concerns.

---

> > ### Comment · Reviewer_g7k9 · 2025-08-02
> >
> > Thank you for your response. I will re-consider my evaluation of the pros and cons.

---

> > > ### Author Response · Authors · 2025-08-05
> > >
> > > > Thank you for your response. I will re-consider my evaluation of the pros and cons.
> > >
> > > Thank you for considering our rebuttal. If there are any further points to clarify, we would be happy to answer.

---

> > > > ### Comment · Reviewer_g7k9 · 2025-08-08
> > > >
> > > > Dear authors, I read your exchange with reviewer xrse with some interest. Thank you for supplying the additional experiments  and the added baselines (DMD and POD-based methods).
> > > >
> > > > I have raised my score to 4, since the paper is well written and I want to acknowledge its reproducibility. For the paper, I think adding a full cost comparison (offline+online) would help the presentation. The reason for a not even higher score for me remain the small scale of the examples and the fact that some of the competing baselines are quite weak (I consider the under-resolved "DNS" more proof of principle than baseline and the authors acknowledged the superior performance of DMD on problems that suit it).
> > > >
> > > > Thank you again for the constructive discussion.

---

### Official Review · Reviewer_wKE9 · 2025-07-04

**Clarity:** 3
**Significance:** 2
**Originality:** 3
**Rating:** 4
**Confidence:** 3

**Summary:**

This paper presents an interesting approach to modeling stochastic multiscale problems by separating scales through coupled SDEs. The work introduces a thoughtful application of Product of Experts (PoE) likelihood to determine the separation between macro and micro scales. The experimental evaluation demonstrates promising results across several PDE problems including KdV, Burgers, and Kármán vortex, showing improvements over implicit scale methods.

**Questions:**

See above.

**Ethical Concerns:**

["NO or VERY MINOR ethics concerns only"]

**Final Justification:**

The author answers my question. The new experiments details on the runtime are very helpful. Therefore I will increase the rating from 3 to 4.

**Limitations:**

The paper does not seem to fully discuss its limitations.

**Quality:**

3

**Strengths And Weaknesses:**

## Strengths
The paper tackles an important and challenging problem in computational physics and machine learning. The use of variational inference for scale separation is mathematically sound and well-motivated. The theoretical framework connecting stochastic multiscale modeling with neural operators is noteworthy and could inspire future work in this direction.

## Weaknesses and Questions:
- Why not different dynamcis: The latent dynamics equation (2) treats macro and micro scale states similarly. The work could benefit from exploring differentiated dynamics—for instance, allowing macro scale components to be more global (perhaps through convolution) while keeping micro scale components more local. This architectural choice could better reflect the inherent nature of different scales.
- Why not more scales: can this framework be generalized to more than two scales?
- Baseline Comparisons: While the variational inference approach is interesting, the current baseline could be too weak. When using implicit-scale, one can use high latent dimension. Generally speaking the machine learning community has seen success with larger number of model parameters in recent machine learning-based surrogate models. To fully demonstrate the method's capabilities, comparisons with established machine learning baselines such as FNO and PDE-refiner, which are multiscale in nature, would strengthen the evaluation and provide better context for the results.
- Problem Complexity: Although the paper is motivated by large-scale turbulence problems, the current PDE test cases may not fully showcase the method's advantages. These problems could potentially be resolved using simpler approaches like finite difference solvers or standard ML-based surrogate models. The work would benefit from evaluation on more challenging turbulence problems, such as Navier-Stokes equations with higher Reynolds numbers, where scale separation treatment such as LES becomes truly necessary and beneficial.

---

> ### Author Rebuttal · Authors · 2025-07-31
>
> We thank you for your constructive review. We are encouraged that you found our approach to be **"interesting,"** our use of variational inference **"mathematically sound and well-motivated,"** and our overall theoretical framework **"noteworthy."** Your accurate summary confirms our shared understanding of the paper's core components.
>
> Before addressing your specific questions, we would like to briefly re-emphasize the three key contributions of our work, which you correctly identified as tackling an **"important and challenging problem."** Our framework introduces:
>
> 1. **A Structured Latent Space for Explicit Scale Separation:** Unlike traditional approaches that learn a single, monolithic latent representation, we explicitly decompose the system into a **coarse-grained macro-state (ζ)** that lives on a physical grid and a **compressed micro-state (η)** that captures unresolved sub-grid features. This is a deliberate structural choice designed to enhance physical interpretability.
>
> 2. **A Principled Mechanism for Learning Scale Interactions:** We introduce the **Product of Experts (PoE) likelihood** as a novel learning objective in this context. As derived in Appendix C, this objective provides a principled, data-driven mechanism for "closure." It automatically learns how to weigh the contributions of the macro- and micro-scales by adaptively regularizing the micro-scale, ensuring it only contributes when necessary.
>
> 3. **A Fully Stochastic, Continuous-Time Dynamical Model:** The evolution of the coupled (ζ,η) states is governed by a learned **Stochastic Differential Equation (SDE)** system. This provides a formal framework for modeling continuous-time dynamics and propagating multiple sources of uncertainty, moving beyond deterministic, discrete-time maps.
>
> Your thoughtful questions about architectural choices, baselines, and problem complexity give us an excellent opportunity to clarify the scope of these contributions and to present new evidence that strengthens our evaluation. We address each of your points in detail below.
>
>
> ## 1. On Differentiated Dynamics for Macro and Micro Scales
>
> > "The latent dynamics equation (2) treats macro and micro scale states similarly. The work could benefit from exploring differentiated dynamics..."
>
> This is an excellent point. While Eq (2) states the dynamics in their most general form, our framework does indeed admit—and in fact, uses—different architectural choices for the macro and micro components to reflect their distinct natures.
>
> As detailed in Appendix A, we impose a static graph structure on the macroscale drift f_θ. This respects the spatial locality of the coarse grid and allows for direct comparison with semi-discretized PDEs (as shown in Fig. 9). In contrast, the microscale drift g_θ operates on a non-spatial, compressed latent vector η and is modeled with a standard MLP. This architectural differentiation is a key feature of our implementation. Your suggestion of using convolutions for the macro-drift is very insightful—it is a similar idea to our graph-based approach and a promising direction for future refinement.
>
> ## 2. On Generalizing to More Than Two Scales
>
> > "Why not more scales: can this framework be generalized to more than two scales?"
>
> This is a very insightful question. In our framework, the primary distinction is between features that are resolved by a chosen coarse computational mesh (the macroscale) and those that are unresolved (the microscale). By this definition, any additional feature scale would fall into one of these two categories.
>
> However, your question points to a more ambitious goal. One could certainly envision a hierarchical extension of our framework with multiple, nested coarse grids, creating a sequence of latent states. This would be analogous to a multiresolution analysis and is a compelling, though non-trivial, direction for future work. We have added a note to this effect in our conclusion.
>
> ## 3. On Baseline Comparisons and Latent Dimension
>
> > "The current baseline could be too weak. [...] comparisons with established machine learning baselines such as FNO and PDE-refiner [...] would strengthen the evaluation."
>
> This is a crucial point that touches on the positioning of our work. While FNO and PDE-refiner are powerful, they represent a different modeling philosophy. Our work aims to move beyond simply building flexible models with an inductive bias that equips them with the capacity to learn from data with multiscale features. Instead, our goal is to introduce a principled statistical framework that enables the data-driven discovery of models that learn to explicitly separate the scales. Our framework retains physical interpretability by:
>
> - Defining the macro-state via a learned linear filter (the smoothing operator).
> - Imposing a graph structure on the macro-dynamics, enabling direct comparison with the underlying PDE operators (Fig. 9).
>
> FNO and PDE-refiner are highly flexible, "black-box" function approximators that do not share these structural priors. Therefore, while a performance benchmark is interesting, our primary scientific comparison is against our own implicit-scale model, which isolates the contribution of our core idea: the explicit microscale state.
>
> **New Numerical Study: High-Dimensional Implicit-Scale Model:** You raise an excellent point about latent dimension. To test whether the advantage of our multiscale model comes from structure or simply more parameters, we conducted a new experiment on the 2D Cylinder Flow case. We trained a new, higher-capacity implicit-scale model with four times the number of macroscale grid points (1024 vs. the original 256). Additionally, we performed a coarse DNS on an unstructured mesh of approximately the same size as our original coarse mesh. The results are summarized below:
>
> | Model                  | # Gridpoints | Error mean | Std dev   |
> | ---------------------- | ------------ | ---------- | --------- |
> | DNS                    | 278          | 0.227      | 0.001     |
> | Implicit scale         | 256          | 0.105      | 0.001     |
> | Implicit scale         | 1024         | 0.063      | 0.001     |
> | **Multiscale (n_η=5)** | **256**      | **0.031**  | **0.001** |
>
> This result shows that while quadrupling the capacity of the implicit-scale model provides a clear benefit, it is still twice as inaccurate as our much smaller, more structured multiscale model. This strongly supports our central claim that the explicit separation of scales is the key driver of performance, not just raw parameter count. This also shows that the coarse mesh compromises the predictive accuracy of DNS substantially, as it is twice as inaccurate as the equivalent-resolution implicit scale model.
>
> ## 4. On the Complexity of the Test Problems
>
> > "The work would benefit from evaluation on more challenging problems, e.g. turbulence."
>
> We agree that scaling to more complex turbulence flow models is the ultimate goal. However, the primary focus of this paper is to introduce and rigorously validate the theoretical and algorithmic framework. The chosen test cases, while classical, are ideal for this purpose as they exhibit the key multiscale phenomena (wave-breaking, shock formation, vortex shedding) needed to clearly demonstrate and diagnose our scale separation mechanism.
>
> Results obtained for the cylinder flow case demonstrates the ability of our methodology to handle complex, spatio-temporal dynamics. The fact that our method succeeds even when the coarse grid cannot resolve the cylinder itself is a strong testament to its ability to learn and represent critical sub-grid phenomena, suggesting it is well-poised for more complex fluid dynamics problems.
>
> ## 5. On Discussing Limitations
>
> > "The paper does not seem to fully discuss its limitations."
>
> Thank you for pointing this out. We agree that consolidating these points is important for clarity. We will add a dedicated "Limitations" section to the conclusion of the revised manuscript. This section will summarize and elaborate on the following key points:
>
> - **Validity of the Markovian Assumption:** As we argue in detail in Appendix B, our (ζ,η) framework is explicitly designed to be a powerful, learned Markovian approximation of the true, potentially non-Markovian coarse-grained dynamics. However, this remains an approximation, and its validity is system-dependent.
>
> - **Latent SDE Stability:** As noted in our results, we did not use stability-preserving priors on our learned dynamics, which may be necessary for more complex systems.
>
> - **Microscale State Structure:** The macro-state ζ has a clear spatial structure, while the micro-state η is a "global" compressed vector. Formulating a more spatially-aware microscale representation is a key avenue for future work.
>
> - **Gaussian Likelihood:** Our use of a Gaussian PoE likelihood, while enabling a concise derivation, may be too restrictive for systems with highly multi-modal behavior (e.g., high-Re turbulence).
>
> We believe this dedicated section will provide a more transparent and comprehensive overview of the current boundaries of our framework. We are grateful for your feedback, which has helped us to significantly improve our manuscript.

---

> > ### Comment · Reviewer_wKE9 · 2025-08-04
> >
> > Thanks the authors for the response. It answers most of my questions. I like the newly added experiments. It seems the DNS methods has a relatively high error rate. Is the error compared to ground true using DNS on a fine mesh?
> >
> > While the comparsion with respect to the grid point is impressive, it will also be helpful to consider the runtime. It will be very helpful to have a converge plot, where the x-axis is the runtime and the y-axis is the error, where we compare the converges of DNS, implicit scale, and multi-scale methods.

---

> > > ### Author Response · Authors · 2025-08-05
> > >
> > > Thank you for your positive feedback and thoughtful comments. We are glad that our response and the new experiments helped to clarify our contributions. We are happy to provide additional details on your follow-up questions.
> > >
> > > ## 1. On the DNS Error Calculation
> > > > "It seems the DNS methods has a relatively high error rate. Is the error compared to ground true using DNS on a fine mesh?"
> > >
> > > Yes, that is correct. For all experiments, the *ground truth* is a high-fidelity simulation on a fine mesh. The **Coarse DNS** error of 0.227 represents the substantial inaccuracy incurred when a standard solver is run directly on a coarse grid, with its error measured against that same fine-mesh ground truth. This confirms that simply downsampling the simulation is not a viable strategy for this problem.
> > >
> > > ## 2. On Runtime vs. Accuracy Comparison
> > > > "While the comparsion with respect to the grid point is impressive, it will also be helpful to consider the runtime. It will be very helpful to have a converge plot...where we compare the converges of DNS, implicit scale, and multi-scale methods."
> > >
> > > Thank you for this suggestion. We fully agree that work-precision (error vs. runtime) plots provide valuable insights and are a great way to summarize the results for this kind of analysis. We will include these plots in the revised paper.
> > >
> > > For this discussion, we provide a "snapshot" of such a plot using the data from our experiments for the cylinder flow case. The table below compares the accuracy and the online simulation time (inference cost) for a single trajectory.
> > >
> > > | Model                  | Online Simulation time (min) | Error mean | Std dev   |
> > > | ---------------------- | ---------------------------- | ---------- | --------- |
> > > | Coarse mesh DNS        | 30.8                         | 0.227      | 0.001     |
> > > | Implicit scale         | 0.34                         | 0.105      | 0.001     |
> > > | **Multiscale (n_η=5)** | **0.59**                     | **0.031**  | **0.001** |
> > >
> > > This data paints a very clear picture of the trade-offs. Our stochastic multiscale model is:
> > >
> > > - **Over 50 times faster** than the Coarse DNS, while achieving **7 times lower error**.
> > > - **Achieves 3.4 times lower error** than the implicit-scale model with only a modest increase in inference cost.
> > >
> > > These results strongly suggest that our stochastic multiscale modeling framework is not only more accurate but also computationally far more efficient, occupying a superior position on the work-precision curve. We will provide this table in the format of a plot for all test cases in the revised paper.
> > >
> > > Once again, thank you for your thoughtful suggestions, which have helped us significantly strengthen the discussion in our paper on the practical impact of our theoretical contributions.

---

> > > > ### Comment · Reviewer_wKE9 · 2025-08-08
> > > >
> > > > Thanks the author for the response. It answers my question. I like the new results, and I will update my score from 3 to 4.

---

### Note · Authors · 2025-08-15

We thank Reviewers wKE9, g7k9, 7ou3, and xrse for an exceptionally rigorous and constructive discussion period. Their feedback has been invaluable in strengthening our paper.

The main focus of the discussion centered on two critical comparisons: (1) how our framework compares against coarse-grid DNS, and (2) how it compares to established linear projection-based ROMs. We fully agree with the reviewers that new methods must demonstrate clear advantages.

To address this feedback, we conducted extensive new experiments. For the DNS comparison, we showed that standard solvers on coarse grids (cylinder flow testcase) are highly inaccurate (error: 0.227) while our multiscale model achieves an error of **0.031**—a **7x** improvement at equivalent resolution with **50x** faster runtime. More decisively, on a new test case with slow-decaying Kolmogorov $n$-width designed to challenge linear projection schemes, our approach demonstrated a **5x** reduction in error compared to POD-SINDy and DMD.

We believe that these new results strengthen the value of the framework we introduce for *data-driven discovery of stochastic multiscale models*. The core ingredients of our framework are: (1) structured latent space to explicitly separate coarse-grained macro-states ($\zeta$) from compressed micro-states ($\eta$), (2) coupled system of SDEs governing the latent dynamics, (3) principled PoE likelihood to adaptively learn scale interactions, and (4) solver-free stochastic variational inference for efficient learning.

Based on the valuable feedback we received from the reviewers, our revised manuscript will include: (1) the new challenging test cases demonstrating clear advantages over linear ROMs, (2) comprehensive error vs. runtime analysis, (3) clearer positioning of our stochastic multiscale modeling framework, and (4) a dedicated limitations section. We are grateful that all reviewers found merit in our work and for providing insightful feedback which enabled us to make our case more compelling.

---

### Decision · Program_Chairs · 2025-09-17

**Decision:**

Accept (poster)

**Comment:**

The paper outlines an approach to learn stochastic multi-scale models by learning a series of coupled SDEs from observational data. Paper has several interesting technical contributions including a principled, probabilistic scheme for scale separation, which might find broad adoption in the field and in the broader ML community. All reviewers and the AC agreed that the paper was well executed and written in its original form and the reviewers concerns primarily focuses on adding additional baseline benchmarks, which were included during the discussion. The main limitation which remained after the discussion is that the experiment only involve small, toy examples which can easily be treated with standard methods, so it is unclear how scalable and generally applicable the presented methodology will be. We encourage the authors to clearly state these limitations in the camera ready version of the manuscript.